# A stochastic neuronal model predicts random search behaviors at multiple spatial scales in *C. elegans*

William M Roberts[1], Steven B Augustine[2†], Kristy J Lawton[3†], Theodore H Lindsay[4†], Tod R Thiele[5†], Eduardo J Izquierdo[6], Serge Faumont[1], Rebecca A Lindsay[7], Matthew Cale Britton[8], Navin Pokala[9], Cornelia I Bargmann[10], Shawn R Lockery[1*]

[1]Institute of Neuroscience, University of Oregon, Eugene, United States; [2]School of Nursing, University of Pennsylvania, Philadelphia, United States; [3]Biology Department, Reed College, Portland, United States; [4]Division of Biology and Biological Engineering, California Institute of Technology, Pasadena, United States; [5]Department of Biological Sciences, University of Toronto, Toronto, Canada; [6]Cognitive Science Program, Indiana University, Bloomington, United States; [7]Department of Ophthalmology, The Vision Center, Children's Hospital Los Angeles, Los Angeles, United States; [8]Department of Neurology, University of Minnesota, Minneapolis, United States; [9]Department of Life Sciences, New York Institute of Technology, Old Westbury, United States; [10]Howard Hughes Medical Institute, Rockefeller University, New York, United States

*For correspondence:
shawn@uoregon.edu

†These authors contributed equally to this work

**Abstract** Random search is a behavioral strategy used by organisms from bacteria to humans to locate food that is randomly distributed and undetectable at a distance. We investigated this behavior in the nematode *Caenorhabditis elegans*, an organism with a small, well-described nervous system. Here we formulate a mathematical model of random search abstracted from the *C. elegans* connectome and fit to a large-scale kinematic analysis of *C. elegans* behavior at submicron resolution. The model predicts behavioral effects of neuronal ablations and genetic perturbations, as well as unexpected aspects of wild type behavior. The predictive success of the model indicates that random search in *C. elegans* can be understood in terms of a neuronal flip-flop circuit involving reciprocal inhibition between two populations of stochastic neurons. Our findings establish a unified theoretical framework for understanding *C. elegans* locomotion and a testable neuronal model of random search that can be applied to other organisms.
DOI: https://doi.org/10.7554/eLife.12572.001

## Introduction

Random search is an evolutionarily ancient set of foraging strategies that evolved as an adaptation to environments in which prey items are undetectable at a distance and occur at unpredictable locations. Rather than attempting to exhaustively search a region of interest, the organism samples the environment at randomly selected points. This is achieved by executing a series of straight-line movements, called 'runs,' terminated at random intervals by sampling episodes during which the organism may or may not find prey. Sampling ends in a reorientation event, called a 'turn,' such that the next run is usually in a different direction from the preceding one. In optimal random foraging strategies the probability distribution of run length is matched to the statistical distribution of isolated food patches or prey items (*Viswanathan, 2011*), with power law distributions predominating

**eLife digest** An animal's ability to rapidly and efficiently locate new sources of food in its environment can mean the difference between life and death. As a result, animals have evolved foraging strategies that are adapted to the distribution and detectability of food sources. Organisms ranging from bacteria to humans use one such strategy, called random search, to locate food that cannot be detected at a distance and that is randomly distributed in their surroundings. The biological mechanisms that underpin random search are relatively well understood in single-cell organisms such as bacteria, but this information tells us little about the mechanisms that are used by animals, which use their nervous system to control their foraging behavior.

Roberts et al. have now investigated the biological basis for random search behavior in a tiny roundworm called *Caenorhabditis elegans*. This worm forages for pockets of bacteria in decaying plant matter and has a simple and well-understood nervous system. Roberts et al. used information on how the cells in this worm's nervous system connect together into so-called "neural circuits" to generate a mathematical model of random searching.

The model revealed that the worm's neural circuitry for random searching can be understood in terms of two groups of neuron-like components that switch randomly between "ON" and "OFF" states. While one group promotes forward movement, the other promotes backward movement, which is associated with a change in search direction. These two groups inhibit each other so that only one group usually is active at a given time. By adjusting this model to reproduce the behavioral records of real worms searching for food, Roberts et al. could predict the key neuronal connections involved. These predictions were then confirmed by taking electrical recordings from neurons. The model could also account for the unexpected behavioral effects that are seen when a neuron in one of these groups was destroyed or altered by a genetic mutation.

These findings thus reveal a biological mechanism for random search behavior in worms that might operate in other animals as well. The findings might also provide future insight into the neural circuits involved in sleep and wakefulness in mammals, which is organized in a similar way.

DOI: https://doi.org/10.7554/eLife.12572.002

when resources are sparsely distributed and exponential distributions predominating when resources are densely distributed (*Humphries et al., 2010*; *Sims et al., 2012*; *Humphries et al., 2012*).

Random search has been documented in a wide range of species including microorganisms, nematodes, insects, mollusks, fish, birds, and mammals including humans (*Viswanathan, 2011*; *Berg and Brown, 1972*; *Pierce-Shimomura et al., 1999*). In humans this strategy is observed in diverse contexts, from traditional hunter-gatherer societies (*Brown et al., 2007*; *Humphries and Sims, 2014*) to technologically enhanced fishing industries (*Bertrand et al., 2007*). The formal similarities between random search across widely diverse phyla and spatial scales (Viswanathan, 2011) may point to a common mechanism, even in organisms that are highly cognitive. Despite the universality of random search, little is known about its neuronal basis, in part because of the difficulty of recording and manipulating activity in the brain of an unrestrained animal while it explores a large region of space.

The relatively small spatial scale of random search behavior in *C. elegans*, coupled with the simplicity of its nervous system, provides a unique opportunity to identify the neuronal basis of random search in this species. To the unaided eye, *C. elegans* search behavior consists of forward runs, each terminated after a variable interval by a briefer period of reverse locomotion, which is also variable in duration (*Pierce-Shimomura et al., 1999*; *Zhao et al., 2003*; *Wakabayashi et al., 2004*), with apparently stochastic switching between these two behavioral states. Reversals are followed by resumption of forward movement that frequently begins with a deep body bend. These bends are highly variable in amplitude and lead to movement in a new direction. Thus, the sequence reverse–forward–deep bend, called a 'pirouette' (*Pierce-Shimomura et al., 1999*) is the fundamental turning event in *C. elegans* random search, with functional analogies to tumbles in bacterial chemotaxis (*Berg and Brown, 1972*). Careful inspection reveals a third state, called "pause," in which locomotion ceases for a fraction of a second or more (*Croll, 1975*; *Shingai, 2000*; *Stephens et al., 2008*;

*Rakowski et al., 2013*; *Salvador et al., 2014*). Thus, *C. elegans* locomotion consists of three main behavioral states – forward, reverse, and pause – together with the transitions between them.

*C. elegans* subsists on a diet of bacteria that it finds mainly in rotting plant material (*Frézal and Félix, 2015*). In the laboratory, search behavior is studied in worms foraging on agar plates containing one or more dense bacterial lawns, analogous to food patches in the ethological literature. Like many other organisms, *C. elegans* can tune the spatial scale of random search to its physiological state, the availability of food (*Wakabayashi et al., 2004*; *Gray et al., 2005*), and prior knowledge of its distribution (*Calhoun et al., 2014*). The lowest values of search scale are observed during "cropping," (*Jander, 1975*) the exploitation of a dense food patch. In *C. elegans*, two substates of cropping have been described: "dwelling," characterized by especially low crawling speed and frequent (presumably short) reversals, and "roaming," characterized by somewhat higher speeds and less frequent reversals. Transitions between dwelling and roaming, like the transitions between forward and reverse locomotion, are stochastic (*Ben Arous et al., 2009*; *Fujiwara et al., 2002*; *Flavell et al., 2013*). Intermediate values of search scale are observed during "local search" (*Wakabayashi et al., 2004*; *Hills et al., 2004*) when, for example, the animal is suddenly transferred from a bacterial lawn to a foodless region of the plate. The highest values of search scale are observed during "ranging," when food is exhausted, starvation sets in, and the need to find a new food patch becomes urgent (*Wakabayashi et al., 2004*; *Gray et al., 2005*). Worms sometimes spontaneously leave a food patch well before it is exhausted, with leaving rate inversely related to food quality and food density (*Shtonda and Avery, 2006*; *Harvey, 2009*), which may reflect a trade-off between exploitation and exploration (*Bendesky et al., 2011*).

At the heart of the *C. elegans* locomotion circuit are five pairs of premotor 'command' interneurons organized into two functional groups that promote forward and reverse locomotion, respectively (*Chalfie et al., 1985*; *Zheng et al., 1999*; *Stirman et al., 2010*; *Schmitt et al., 2012*). The two groups are reciprocally connected, and make output synapses onto distinct, non-overlapping sets of motor neurons that control body-wall muscle. The locomotory state (forward or reverse) is believed to be determined mainly by whichever set of motor neurons is more highly activated by input from the command neurons (*Kawano et al., 2011*; *Xie et al., 2013*; *Gao et al., 2015*; *Liu et al., 2014*). Command neuron activation depends upon influences that are both external and intrinsic to the command neuron network, and appears to have a strong stochastic component that underlies switching between forward and reverse locomotion. Some command neurons are tightly linked both functionally and synaptically to upstream interneurons that also switch state stochastically in concert and counterpoint to them (*Gordus et al., 2015*), providing a potential additional source of the stochasticity on which random search depends. At least nine classes of chemosensory neurons and twelve classes of upstream interneurons are required for normal regulation of the duration of forward locomotion (*Viswanathan, 2011*; *Gray et al., 2005*; *Tsalik and Hobert, 2003*; *Fang-Yen et al., 2015*). Input from these neurons onto the command neuron network modulates the mean run length and, thereby, the spatial scale of random search. Search scale also appears to be modulated by neurons that release biogenic amines (serotonin, dopamine, and tyramine) (*Flavell et al., 2013*; *Hills et al., 2004*; *Bendesky et al., 2011*) or peptides (*Ben Arous et al., 2009*; *Flavell et al., 2013*; *Gloria-Soria and Azevedo, 2008*; *Styer et al., 2008*; *Reddy et al., 2009*; *Bhattacharya et al., 2014*). These diverse signaling pathways may provide the means by which the worm optimizes its search strategy in response to feeding history (*Gray et al., 2005*), the quality, density and spatial distribution of food (*Shtonda and Avery, 2006*; *Calhoun et al., 2015*), and other factors that constrain survival and reproduction (*Gloria-Soria and Azevedo, 2008*; *Pujol et al., 2001*; *Pradel et al., 2007*; *Lipton et al., 2004*).

Although the neural circuitry for local search has been described in considerable detail, our understanding of the system remains limited, partly for lack of key physiological data, but also for lack of a model in which to interpret the data. Common sense suggests that the forward and reverse command neurons should inhibit each other to minimize simultaneous occurrences of neuronal states for incompatible behaviors (*Zheng et al., 1999*). A plausible anatomical substrate for such reciprocal inhibitory connections between command neurons exists in the *C. elegans* connectome (*White et al., 1986*), but anatomical data do not specify the signs or strengths of synaptic connections. A quantitative model that incorporates physiological properties of the command neurons and their synaptic connections is needed to interpret experimental results, such as the unexpected observation that silencing some of the reverse command neurons causes a reduction in forward

dwell time, and conversely for forward command neurons (*Rakowski et al., 2013*; *Zheng et al., 1999*). It is also needed to explain complex patterns of changes in dwell times observed across the three locomotory states caused by introducing or eliminating tonic membrane conductances in the command neurons, and to answer basic mechanistic questions about the control of *C. elegans* locomotion.

At present, the experimental data are insufficient for creating a neuron-by-neuron model of the command network that incorporates biophysical details such as synaptic and membrane conductances without introducing a heavy load of unconstrained parameters (*Rakowski et al., 2013*). Nor would such a mechanistically detailed model necessarily provide the appropriate level of abstraction in which to intuitively understand *C. elegans* search behaviors, including their strong stochastic component. Instead, we have kept the level of biological detail to the minimum needed to predict the statistical distributions of dwell times in forward, reverse and pause states, and other fundamental aspects of the behavior. Each of the model's three main assumptions remains within the bounds of widely accepted experimental results; our mathematical analysis simply shows what follows necessarily from these assumptions.

To provide an empirical basis for the model we quantified *C. elegans* search behavior in terms of tangential velocity, defined as the speed and direction of worm's movement along its sinuous trajectory, which we recorded at higher resolution than previously possible. Behavioral data were then fit to a four-state hidden Markov model in which each state corresponds to a unique pattern of activation across the command neurons. Importantly, rate constants governing probabilistic transitions between states in the Markov model are expressed in terms of synaptic weights in an analytically tractable version of the model. We were therefore able to validate the model by showing that it correctly predicts phenomena on which it was not fit, such as reciprocal inhibition between forward and reverse command neurons in the biological network and the behavioral effects of perturbations introduced by laser ablations and genetic mutations. Although the model is inherently probabilistic, we found that it also makes accurate predictions concerning deterministic behaviors in *C. elegans*, indicating a potentially high level of generality. The present findings thus establish a simple theory of *C. elegans* locomotory control and provide a testable model of random search that can be applied to other organisms.

## Results

A neuronal model of random search in *C. elegans* is a theory of the relationship between activation states of the command neurons and foraging behavior. Methods presently available for observing neuronal activity in freely behaving *C. elegans* utilize calcium-sensitive probes that have insufficient temporal resolution to observe the changes in neuronal activity associated with the rapidly changing behavioral states, especially the frequent brief pauses that are an integral part of the behavior. Therefore, as a proxy for command neuron state, we used the worm's tangential velocity, defined as the speed and direction of worm's movement along its sinuous trajectory. We focused on tangential velocity because in sinusoidal locomotion the net reactive forces produced by body-wall muscle contractions acting against the substrate are tangential to the body surface (*Gray et al., 2005*). Tangential velocity therefore provides the most direct readout of which group of motor neurons and command neurons (forward or reverse) is more active (*Qi et al., 2013*). Alternative measures of the rate of translation such as centroid velocity (*Pierce-Shimomura et al., 1999*) or postural phase velocity (*Stephens et al., 2011*) have a less direct relationship to command neuron state because these measures either depend in complex ways on the shape of the worm, or rely on a representation of posture that ignores some of the thrust-generating components of the worm's shape that come into play unless the worm is moving along a fairly linear trajectory. To monitor tangential velocity as directly as possible, we painted a microscopic black spot on the worm and used a motorized stage controlled by a computer to keep the spot in the field of view (*Figure 1A*). The most common alternative method for measuring tangential velocity, tracking virtual points obtained by segmenting the worm's centerline, is subject to segmentation errors introduced by low contrast images of the worm's head and tail (see *Cronin et al., 2005*) which changes the distance between virtual points. This method can also be compromised by dropped frames when the worm's centerline crosses itself during tight turns.

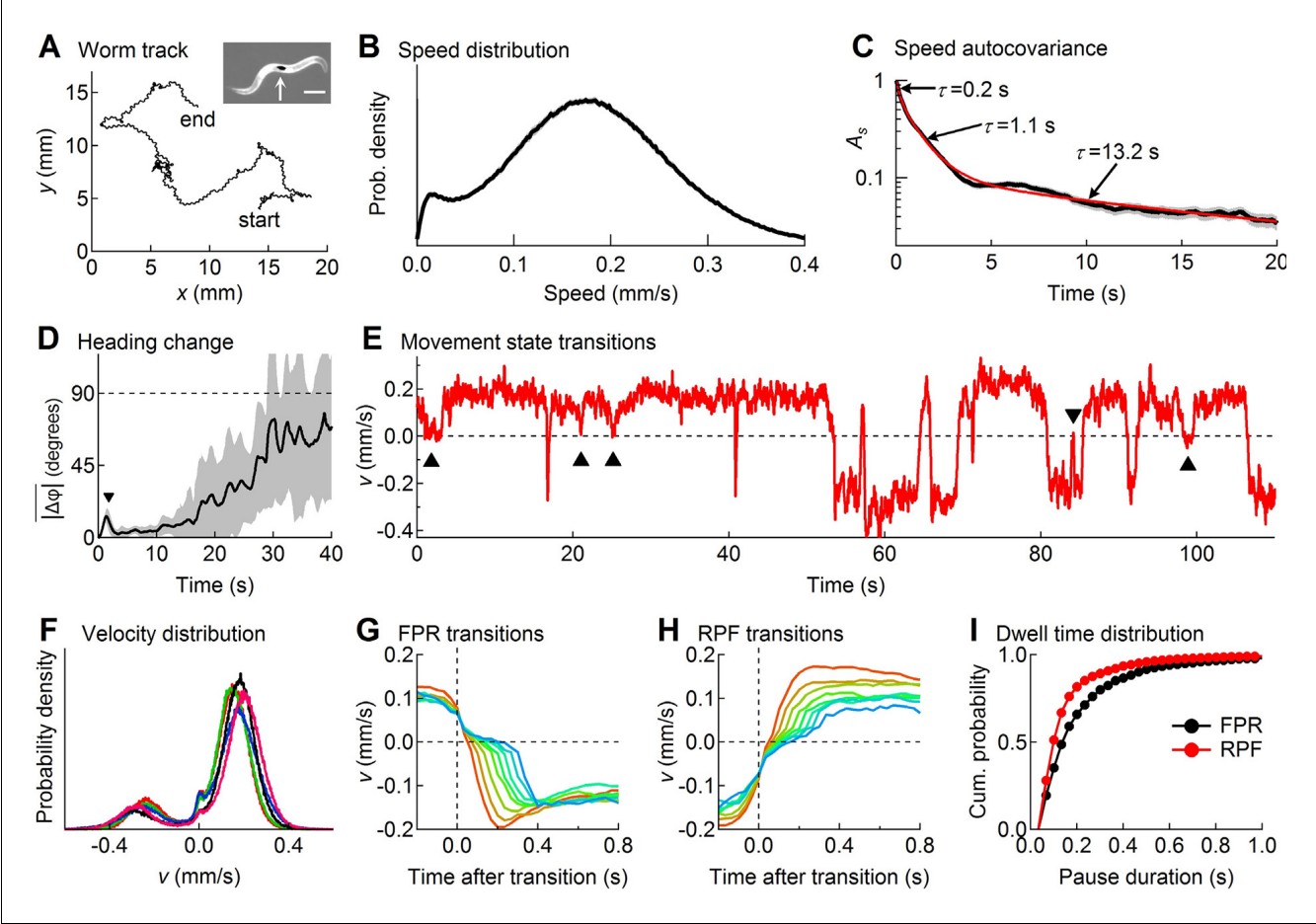

**Figure 1.** Descriptive statistics of wild type worm tracks. (A) $(x, y)$-coordinates of a worm during 10 min of foraging. *Inset*: Image of a worm showing the black spot (arrow) used for optical tracking (scale bar = 200 μm). (B) The speed distribution computed from the distance moved between successive video frames had a peak at 180 μm/s, which includes both forward and reverse locomotion. A second peak at 14 μm/s corresponds to pauses. The decreased probability of observing speeds <14 μm/s (<0.47 μm/frame) is due to noise in the position measurement. (C) At least three time constants were required to fit (red) the speed autocovariance function (black; grey shading shows ± 1 sem). (D) The worm's heading remained nearly constant for ~10 s except for a transient peak at 1.4 s (▼), which corresponds to the period of one half cycle of undulation during sinusoidal locomotion. The dashed line shows random reorientation; shading shows ± 1 sem. (E) Example of $v(t)$ showing periods of forward locomotion, reverse locomotion and pauses of various durations. Upward triangles (▲) mark forward-pause-forward (FPF) events; the downward triangle (▼) marks a reverse-pause-reverse (RPR) event. (F) Velocity distributions for the 5 wild type cohorts (5 colors) analyzed in this study. (G) Ensemble-averaged velocity during FPR transitions. All FPR transitions in all wild type cohorts were aligned at the end of forward movement, grouped according to the duration of the pause (2–9 frames), and averaged. Such transitions were defined using a threshold criterion of $|v| < 50$ μm/s to identify state P (*Rakowski et al., 2013*). Pauses lasting ≤ 1 frame are not shown because of ambiguity in state identification; pauses lasting ≥ 10 frames are omitted for clarity. (H) Identical to G except RPF transitions are shown. (I) Cumulative probability distributions for dwell time in the pause state defined as in G and H for all FPR and RPF transitions of duration >1 frame in wild type worms.

DOI: https://doi.org/10.7554/eLife.12572.003

The following figure supplements are available for figure 1:

**Figure supplement 1.** Optical tracking error.
DOI: https://doi.org/10.7554/eLife.12572.004

**Figure supplement 2.** The worm's search behavior closely resembles a Brownian random walk on time scales longer than 10 s, but not on shorter time scales.
DOI: https://doi.org/10.7554/eLife.12572.005

At the start of a 10 min observation period an individual worm was transferred from a food-laden culture plate to a bare agar surface devoid of overt sensory cues, thereby inducing a period of intensive local search behavior (*Jander, 1975*; *Hills et al., 2004*). The $(x, y)$-coordinates of the centroid of the spot were recorded with a temporal resolution of 33 ms (i.e., frame rate = 30 Hz) and a spatial

resolution of 0.5 μm that was limited mainly by the precision of the stage position encoder; the optical tracking error was much smaller (*Figure 1—figure supplement 1*). A spatial resolution of approximately 0.5 μm amounts to an approximately 10-fold improvement over previously published tracking systems (*Kocabas et al., 2012*); thus worm speed (*Figure 1B*) could be extracted with unprecedented accuracy. For statistical analysis, worms were grouped into cohorts having the same genotype or neurons ablated (17–31 worms per cohort), which had been reared together and tested in parallel as young adults within the same 2–3 day period. This approach yielded a comprehensive data set containing a total of 8.3 million position measurements from 501 individuals in 20 cohorts.

## Model-independent identification of locomotory states

*Figure 1A– D* describes important features of search behavior obtained by regarding the worm as a point moving in an external reference frame (allocentric coordinates) without regard to the orientation of the body axis. The speed distribution was bimodal (*Figure 1B*) with a broad peak around 180 μm/s that includes both forward and reverse motion, and a narrower peak near zero that corresponds to pauses. The speed autocovariance function had multiple exponential components (*Figure 1C*), suggesting at least three locomotory states. The average change in heading angle ($\overline{|\Delta\varphi|}$), plotted as a function of the intervening time interval (*Figure 1D*), showed that worms maintained a nearly constant heading for up to 10 s (*Stephens et al., 2010*; *Peliti et al., 2013*), but reoriented randomly within ~30 s, establishing the shortest time scale over which the behavior can be considered a Brownian random walk (*Figure 1—figure supplement 2*), the simplest form of random search. On shorter time scales the path takes on the character of a truncated Lévy flight (*Mantegna and Stanley, 1994*).

For more detailed analysis we distinguished forward from reverse movement by visual inspection of the recorded videos, and defined velocity, $V(t)$ , to be a signed scalar value that denotes the speed of movement along the worm's track in the direction of the head (+) or tail (-) (*Figure 1E*; see Materials and methods). The probability distribution of $V(t)$ (*Figure 1F*) showed two broad peaks that correspond to forward and reverse movement, and a narrow third peak centered at zero that corresponds to pauses. For the initial analysis we defined pauses using a fixed speed threshold of 0.05 mm/sec (*Rakowski et al., 2013*). Pauses occurred most frequently as transient interruptions of forward locomotion, causing the worm to stutter as it moves (*Figure 1E*; *Video 1*); stuttering also occurred, albeit less frequently, during reverse locomotion (*Figure 1E*; *Video 2*). Distinct pauses were also observed during transitions from forward to reverse (*Figure 1G*; *Video 3*) and from reverse to forward (*Figure 1H*; *Video 4*). Most pauses lasted longer than one video frame, indicating the presence of a locomotory state having a detectable dwell time; thus pauses were not merely zero crossings in plots of velocity versus time. We found that pauses during forward to reverse transitions were on average longer in duration than pauses during reverse to forward transitions (*Figure 1I*; $p<10^{-5}$ ; Mann-Whitney U-test). These findings are consistent with the predictions of the model presented below, which uses a probabilistic criterion rather than a fixed velocity threshold to identify pauses.

## The stochastic switch model

Based on the results presented in *Figure 1* and previous studies noted below, we propose a minimal model for the control of random search behavior that involves two opposing neuron-like "units" that can exist in four distinct states corresponding to forward locomotion, reverse locomotion, and two pause states. This model differs from a previous model that represents the worm as a point in "shape space" (*Stephens et al., 2008*) in that here velocity is measured directly by observing the motion of a point on the body surface relative to the substrate, rather than indirectly by the temporal progression of shape changes. It also differs from previous models (*Rakowski et al., 2013*; *Zheng et al., 1999*; *Wicks et al., 1996*; *Kunert et al., 2014*) by representing changes in locomotory state as probabilistic transitions in a Markov process.

Ablation of individual premotor interneurons (*Chalfie et al., 1985*) has led to the hypothesis that the direction of locomotion is controlled by a network comprising five pairs of premotor command interneurons organized into two functional groups that promote forward and reverse locomotion, respectively. Although the anatomical pattern of synaptic connectivity among these interneurons has been established (*White et al., 1986*) (*Figure 2A*), this information does not yield an intuitive

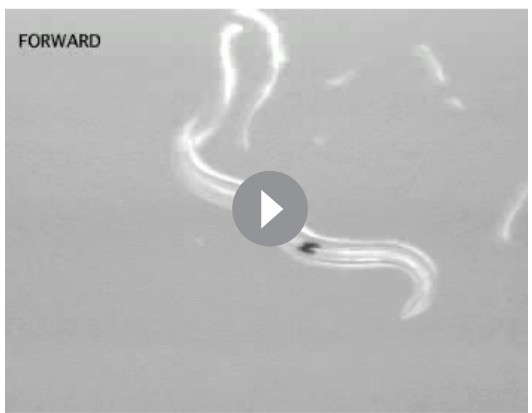

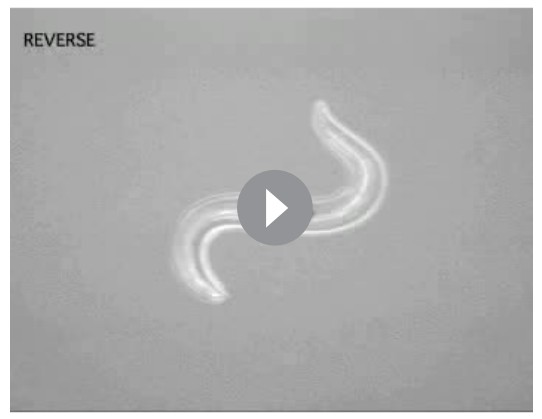

**Video 1.** Forward-Pause-Forward transition. The worm is crawling on a foodless agar plate. The microscope stage moves continuously to keep the tracking spot near the center of the frame. Stage movement can be assessed by monitoring the white streaks in the background, which are segments of the worm's track at earlier times. Behavioral state is indicated in the upper left corner of the frame. The indicated behavioral transition is shown at normal speed, and slowed down by a factor of 5. The worm is paused when the tracking spot is stationary relative to the streaks.
DOI: https://doi.org/10.7554/eLife.12572.006

**Video 2.** Reverse-Pause-Reverse transition.
DOI: https://doi.org/10.7554/eLife.12572.007

explanation of how the direction of locomotion is regulated. Nor, in our view, does the present state of the anatomical connectivity provide the basis for a neuron-by-neuron simulation of the network (but see *Rakowski et al., 2013*), as neither signs nor physiological strengths (weights) of synapses in *C. elegans* can be inferred reliably from anatomical structure or neurotransmitter type, and almost nothing is known about the intrinsic membrane currents of these neurons or how they shape the input-output function of individual command neurons.

To establish a mathematically tractable framework for understanding how the command network functions during search behavior, we created a minimal model based on three simplifying assumptions, each of which was biologically motivated. (i) Command neurons act like binary units (*Hopfield, 1982*). This assumption was based on voltage recordings from command neurons in which we regularly observed two stable membrane potentials with rapid transitions between them (*Figure 2B*; also see *Kato et al., 2015*). It is also supported by the observation of a bimodal distribution of calcium activity in AVA neurons and their upstream partners AIB and RIM (*Gordus et al., 2015*), and the report of distinct up and down states in voltage recordings from motor neruons (*Liu et al.,*

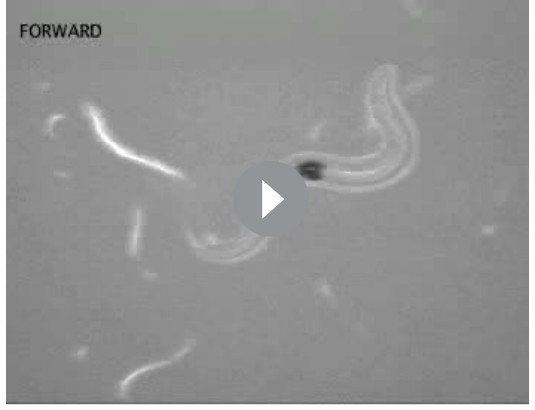

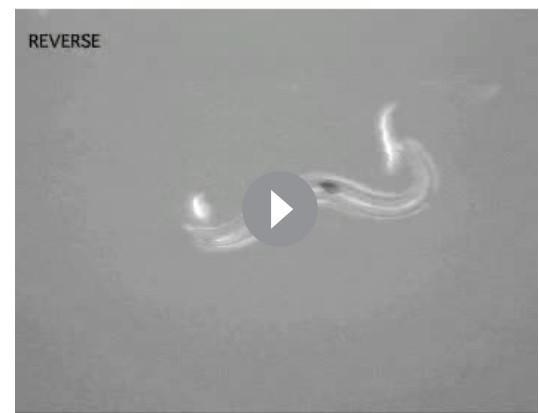

**Video 3.** Forward-Pause-Reverse transition.
DOI: https://doi.org/10.7554/eLife.12572.008

**Video 4.** Reverse-Pause-Forward transition.
DOI: https://doi.org/10.7554/eLife.12572.009

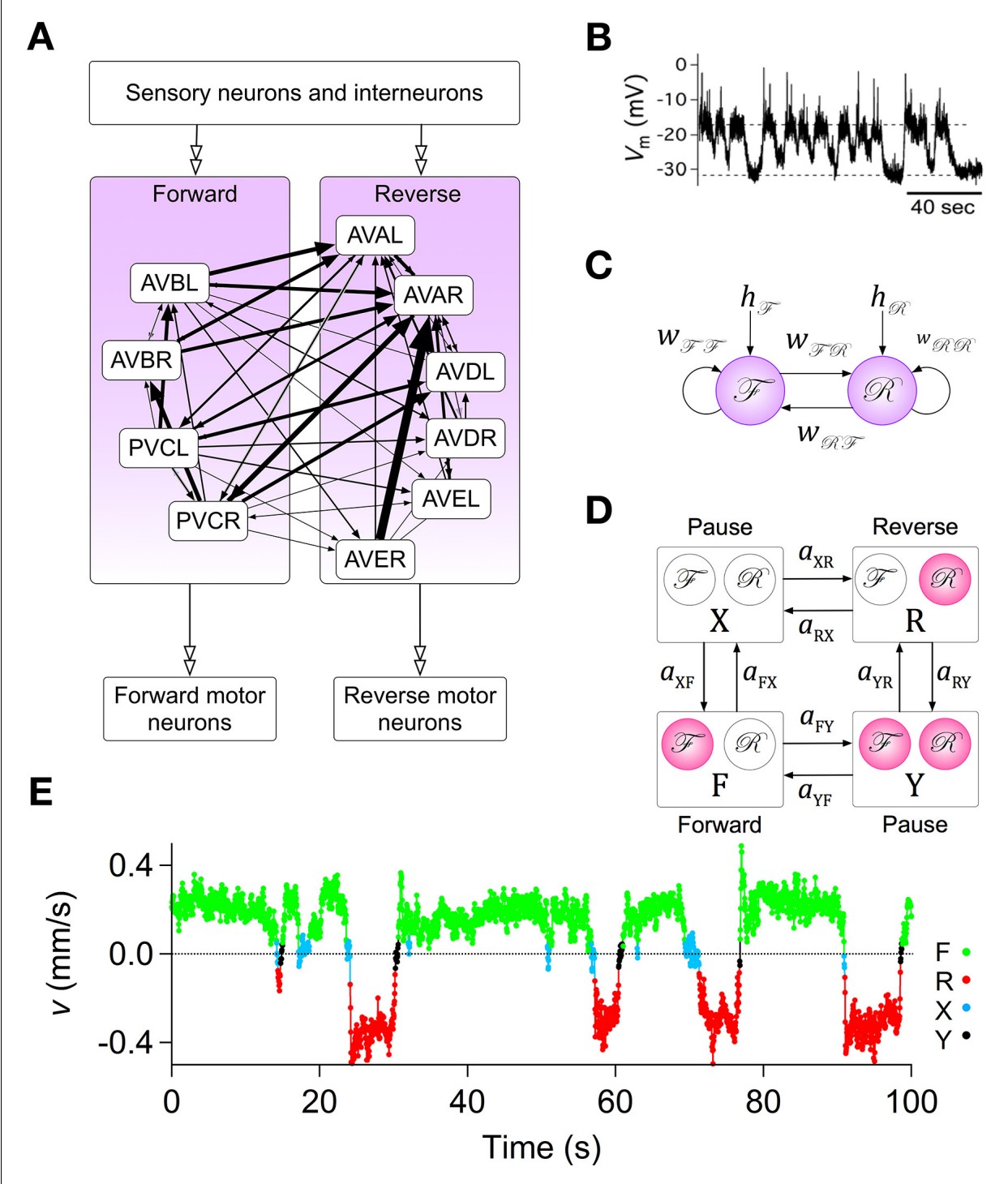

**Figure 2.** Assumptions with supporting data for the Stochastic Switch Model. (A) Connectivity of forward and reverse command neurons. Arrows with single heads are monosynaptic connections inferred from the *C. elegans* connectome (*White et al., 1986*; *Varshney et al., 2011*) line thickness is proportional to the number of presynaptic specializations seen in the reconstruction of each pairwise connection. Open, double-headed arrows indicate synaptic pathways from or to the indicated pool of neurons outside the network. (B) Voltage recording from the command neuron AVA in the absence of injected current. In this neuron, quasi-stable membrane potentials are seen at -17 and -32 mV. These results differ from previously published AVA recordings, which were made in the presence of hyperpolarizing current (5–10 pA) that kept the membrane potential near -55 mV (*Lindsay et al., 2011*). (C) Neuronal representation of the Stochastic Switch Model. Forward and reverse command neurons are represented as single binary neuron-like units $\mathcal{F}$ and $\mathcal{R}$, respectively. Arrows depicting cross connections ($W_{\mathcal{FR}}$, $W_{\mathcal{RF}}$) represent functional (net mono- and polysynaptic) connections

*Figure 2 continued on next page*

*Figure 2 continued*

between forward and reverse units. Self-connections ($W_{\mathcal{FF}}$, $W_{\mathcal{RR}}$) represent synaptic connections between neurons comprising a given unit, voltage dependent currents in these neurons, and polysynaptic recurrent pathways involving non-command neurons. Downward arrows ($h_{\mathcal{F}}$, $h_{\mathcal{R}}$) represent the combined effects of input from presynaptic neurons, including sensory neurons, and neuromodulation. (D) Markov model representation of the command neuron network. The color of a unit indicates its state of activation (red on, white off). In addition to the forward state F and the reverse state R, there are two pause states, X and Y. Arrows, with their associated rate constants, indicate transitions in which a single unit changes state. Transitions in which two units change state simultaneously have probability zero because single-unit transitions are assumed to be statistically independent. (E) The most likely sequence of states in the hidden Markov model (computed using the Viterbi algorithm) for a representative data segment.

DOI: https://doi.org/10.7554/eLife.12572.010

The following figure supplements are available for figure 2:

**Figure supplement 1.** Effects of synaptic input on transition rates in stochastic units $\mathcal{F}$ and $\mathcal{R}$.

DOI: https://doi.org/10.7554/eLife.12572.011

**Figure supplement 2.** Velocity distributions in forward, reverse and pause states.

DOI: https://doi.org/10.7554/eLife.12572.012

**Figure supplement 3.** Cumulative dwell time distributions in states F, R, X and Y.

DOI: https://doi.org/10.7554/eLife.12572.013

*2014*). (ii) Command neurons switch state stochastically. This assumption was based on the observation that *C. elegans* locomotory behavior has a strong stochastic component, with exponentially distributed dwell times in forward and reverse states (*Pierce-Shimomura et al., 1999*; *Zhao et al., 2003*; *Flavell et al., 2013*; *Gordus et al., 2015*; *Stephens et al., 2011*). (iii) Command neurons within the forward pool are co-active, as are command neurons in the reverse pool. This assumption is based on simultaneous calcium imaging data from multiple command neurons in freely moving animals which suggest that the activity of neurons within the reversal pool is tightly correlated (*Schrödel et al., 2013*; *Pokala et al., 2014*). Additionally, neurons in opposing groups are likely to be reciprocally active, as indicated by simultaneous calcium imaging from AVA and AVB (*Pokala et al., 2014*; *Faumont et al., 2011*), as well as AVE and AVB (*Kawano et al., 2011*). A fourth assumption, concerning the relationship between neuronal states and behavioral states, is introduced below.

The three simplifying assumptions, together with the anatomical data (*White et al., 1986*), lead to a model that has two binary stochastic elements, *F* and *R*, and six synaptic weights (*Figure 2C*). Each type of weight has a specific interpretation. The cross-connections ($W_{\mathcal{FR}}$, $W_{\mathcal{RF}}$) represent mono- and polysynaptic connections between command neurons in different groups. The self-connections ($W_{\mathcal{FF}}$, $W_{\mathcal{RR}}$) represent connections between command neurons in the same group, including recurrent polysynaptic pathways involving neurons outside the command network. Self-connections also represent possible intrinsic voltage dependent currents within the command neurons, such as *C. elegans* plateau currents (*Mellem et al., 2008*). The pair of connections originating from an *F* or *R* unit can have either the same sign or different signs. Allowing a single unit to have opposing effects on different postsynaptic targets is justified by the fact that synaptic weights in the model represent polysynaptic pathways, the effects of which can be excitatory or inhibitory, and by the observation that some *C. elegans* neurons can monosynaptically excite some postsynaptic neurons while inhibiting others (*Chalasani et al., 2007*). Two additional weights, $h_{\mathcal{F}}$ and $h_{\mathcal{R}}$, represent inputs from sensory neurons, interneurons, neural modulators, and any other sources outside the network (*Gray et al., 2005*; *Fry et al., 2014*), plus intrinsic membrane conductances that produce sustained effects on membrane potential (*Zheng et al., 1999*; *Gao et al., 2015*). The summed synaptic inputs onto *F* and *R* are, respectively, $S_{\mathcal{F}}(t) = h_{\mathcal{F}} + W_{\mathcal{FF}} b_{\mathcal{F}}(t) + W_{\mathcal{RF}} b_{\mathcal{R}}(t) h_{\mathcal{F}}$ and $S_{\mathcal{R}}(t) = h_{\mathcal{R}} + W_{\mathcal{RR}} b_{\mathcal{R}}(t) + W_{\mathcal{FR}} b_{\mathcal{F}}(t)$, where $b_{\mathcal{F}}(t)$ and $b_{\mathcal{R}}(t)$ are the states of *F* and *R* at time *t* (1 = ON, 0 = OFF). The quantities $h_{\mathcal{F}}$ and $h_{\mathcal{R}}$ were assumed to be constant during the 10 min observation period of local search behavior on a bare agar surface.

State transitions of *F* and *R* were modeled as independent non-homogeneous Poisson processes in which the transition rates are exponential functions of the summed synaptic input to the units, as shown in *Figure 2—figure supplement 1*. Changes of the state of *F* and *R* can be regarded as thermally-driven transitions across energy barriers of height proportional to $S_{\mathcal{F}}(t)$ and $S_{\mathcal{R}}(t)$, respectively. Inhibitory synaptic input increased the height of the barrier for OFF→ON transitions while

decreasing the height of the barrier for ON→OFF transitions by the same amount; excitatory synaptic inputs had the opposite effect. The variable $A$ (Materials and methods, *Equations 26, 27*) represents the fundamental timescale of the system, defined as the rate at which units $F$ and $R$ change state when the summed synaptic input is zero. The present model is distinct from deterministic models of the command neuron network (*Rakowski et al., 2013*; *Zheng et al., 1999*; *Wicks et al., 1996*; *Kunert et al., 2014*) in that it is inherently stochastic, like the behavior it is meant to predict. In particular, the synaptic input to a unit does not immediately determine its state, but instead modifies the transition rates between ON and OFF states.

The two binary units of the model can exist in four states (F, R, X, Y; *Figure 2D*), and provide the basis for a hidden Markov model having eight transitions in which a single unit changes state. The model was further constrained by the synaptic model, which allows the eight transition rate constants to be specified by only six synaptic weights as shown in *Equations 31–35* (Materials and methods). A Markov model was adopted to represent the biological system because dwell times in Markov states, like the observed dwell times in forward and reverse states (*Zhao et al., 2003*; *Viswanathan, 2011*), are exponentially distributed. A hidden Markov model was required because, as noted above, states of command neurons cannot be observed directly in freely moving animals, even using optical recording methods.

The fourth assumption is a particular mapping between the states of the two command units and behavioral states of the worm. The command units, $F$ and $R$, are intended to represent the two pools of forward and reverse command neurons, respectively, such that the worm moves forward when $F$ is ON and $R$ is OFF (state F), backwards when $R$ is ON and $F$ is OFF (state R), and pauses when both $F$ and $R$ are OFF (state X). These associations between states of the model and activation states of the command neurons are well supported by previous experimental evidence, including studies showing that genetic ablation or silencing of all command interneurons induces prolonged pauses (*Zheng et al., 1999*; *Kawano et al., 2011*), but they also assume the major simplification that all command neurons in a given pool act together as a unit.

The model also permits a fourth state, in which $F$ and $R$ are simultaneously ON (state Y). Whether the corresponding co-activation state of forward and reverse command neurons normally exists with any significant probability remains to be shown, but it has been observed that their downstream targets, the forward and reverse motor neurons, can be active simultaneously, causing the worm to pause (*Kawano et al., 2011*). Given the existence of gap junction synapses between the main forward and reverse command neurons and their respective sets of forward and reverse motor neurons, it is reasonable to suppose that forward and reverse command neurons are co-active when their motor neurons are co-active. Thus, there is some evidence to designate state Y as a second pause state, which we consider to be a working hypothesis. Together, states X and Y comprise the phenomenological pause state denoted P. In what follows, we explore the logical consequences of the model's four assumptions; it remains to be shown experimentally how closely the states of the model correspond to activity states of the command neurons.

We used a maximum likelihood method (*Colquhoun and Hawkes, 1995*) (Materials and methods) to estimate the set of transition rate constants that had the highest probability of generating the observed time series $v(t)$. Direct transitions between F and R, and between X and Y, were disallowed because the assumed statistical independence of the two command units implies that the probability of simultaneous transitions in $F$ and $R$ is vanishingly small. (Note, however, that the model does allow transitions between any two states during the interval between successive video frames by making two or more non-simultaneous transitions; see *Equation 21*). We first fit the velocity distribution for each cohort with three overlapping probability distributions corresponding to forward, reverse and pause states (*Figure 2—figure supplement 2*), then searched for the set of transition rate constants that maximized the likelihood of the observed $v(t)$ given the velocity distributions. The resulting rate constants were used to compute the most likely sequence of states via the Viterbi algorithm (*Rabiner, 1989*; *Viterbi, 1967*). The agreement between observed velocity data and the sequence of states shown in *Figure 2E* was typical of the entire data set.

## Wild type locomotion

The maximum likelihood rate constants for 5 wild-type cohorts, together with the predicted state probabilities and mean dwell times computed from them, are given in column A of *Table 1*. The model's predicted mean dwell time in the reverse state ($d_R = 1.94 \pm 0.04$ s) agreed with previously

reported values (*Zheng et al., 1999*; *Kawano et al., 2011*). In contrast, the predicted mean dwell time in the forward state ($d_F$ = 5.33 ± 0.25 s) was smaller than previously reported when dwell time was measured by eye (13–35 sec) (*Zhao et al., 2003*; *Zheng et al., 1999*; *Brockie et al., 2001*; *Ryu and Samuel, 2002*) or by velocity threshold crossings (9–16 sec) (*Rakowski et al., 2013*; *Stephens et al., 2011*). To determine whether this difference arose because we used a hidden Markov model rather than a fixed velocity threshold, we also identified states based on a fixed velocity threshold of 0.05 mm/s, and calculated the resulting mean dwell times: $d_{F,0.005}$ = 1.86 ± 0.03 s; $d_{R,0.005}$ = 1.23 ± 0.02 s; $d_{P,0.005}$ = 0.14 ± 0.001 s. We attribute the short mean dwell times in state F that we observed using either the hidden Markov model or a fixed velocity threshold to the fact that our tracking system is capable of revealing briefer visits to state P, which interrupt forward runs, than previous methods. Ignoring transient interruptions of forward locomotion (i.e., FPF transitions) and using the fixed velocity threshold of 0.05 mm/s yielded longer a mean forward dwell time of 9.13 ± 0.15 s, which matches the value obtained by others using the same threshold (8.98 ± 0.57 s) (*Rakowski et al., 2013*). Predicted mean dwell times in the two pause states differed substantially from each other ($d_X$ =0.44 ± 0.03 s, $d_Y$ = 0.21 ± 0.02). We assigned the long and short pause states to X and Y, respectively, based on the idea that the energetically expensive state in which both units are on should be relatively short-lived.

In previous work, transitions between locomotory states in *C. elegans* have been analyzed by choosing a speed threshold to distinguish pause states from the movement states (*Rakowski et al., 2013*; *Salvador et al., 2014*; *Stephens et al., 2011*). The choice of threshold is important because it affects the measured dwell times, yet is necessarily arbitrary because the velocity distributions of the states overlap (*Figure 1F*). The hidden Markov model used here replaces arbitrary thresholds with empirically determined state transition rates (i.e., the set of rates that maximizes the probability of the observed velocity time series), from which one can determine the sequence of states that is most likely to have generated the data (the Viterbi algorithm). The hidden Markov model thus offers two advantages: (1) it provides a statistical criterion for selecting the best parameter values and (2) it takes into account the uncertainties in identifying the state of the system from velocity data.

Under the assumptions of the hidden Markov model the state of the system cannot be observed directly because the velocity distributions overlap, making it impossible to test directly whether the predicted state probabilities agree with the observed velocity data. Nevertheless, an important test of the model can be obtained using the Viterbi algorithm to identify the most likely sequence of states given the observed velocity data, from which the histogram of dwell times in each state can be computed and compared to the exponential distribution predicted by the Markov model (*Figure 2—figure supplement 3*). The degree of agreement between the distributions shows that our model provides a good description of the system.

The initial rationale for including two pause states in the hidden Markov model came from our model-independent analysis of the tracking data (*Figure 1I*), which showed different dwell time distributions for pauses at FPR and RPF transitions. To test whether having two pause states yielded a statistically significant improvement in the ability of the model to fit the data, we eliminated one of the pause states and asked whether the resulting reduction in likelihood was greater than could be attributed to the reduction in the number of free parameters (see *Table 1*). For this comparison we constrained the transition rates into state Y to be extremely small ($a_{FY} = a_{RY} = 10^{-10}$ s$^{-1}$), effectively eliminating state Y and reducing the number of free parameters from six to four. The reduction in likelihood caused by eliminating one of the pause states was highly significant, and cannot be attributed simply to the elimination of two parameters ($p < 10^{-100}$; likelihood ratio test). Separately, we considered the most general one-pause state model, which allows direct transitions between states F and R and has no constraints on the 6 transition rates other than that they are all ≥0. The fit of this model (*Table 1* column C) converged to nearly the same set of transition rates as the one-state model described above (Model B). These comparisons show that our model with two pause states and six free parameters (the six synaptic weights) provides a much better fit to the data than models with only one pause state. We conclude that the tracking data contain a statistically significant signature of two distinct pause states. The model explains the observation that the pause dwell times during FPR transitions are longer than during RPF transitions (*Figure 1I*) in terms of the different dwell times in states X and Y ($d_X Y$), and the strong tendency to cycle clockwise through state space, exiting from state F to state X and from state R to state Y as shown by the fate diagram (*Figure 3*).

**Table 1.** Maximum likelihood fits of transition rates in wild type *C. elegans*.

Each cohort was fitted separately; values are expressed as mean ± sem ($n$ = 5 cohorts). Data from wild type cohorts were obtained on the same days as the experimental cohorts for which they served as controls (*Tables 3* and *4*), but experimental cohorts in this study were separated by weeks to months. All transition rates were constrained to be ≥0. Transition rates that were calculated using the synaptic constraints (*Equation 35*) are shaded orange; other constrained values are shaded grey. Mean dwell times and state probabilities were calculated from the transition rates. Column A shows fits using the standard model, which has 8 rate constants with two synaptic constraints, resulting in 6 free parameters that determine the 6 synaptic weights (*Figure 2C,D*; Materials and methods *Equations 31–35*). Column B shows fits to a model that has only one pause state (X); this model was derived from the standard model by imposing two more constraints: $a_{FY} = a_{RY} \cong 0$, yielding 4 free parameters. To allow comparison of models A and B by the likelihood ratio test, which requires tht model B be a special case of model A, $a_{RY}$ and $a_{FY}$ were set slightly >0 ($10^{-10}$ s$^{-1}$), thereby avoiding infinite values for $a_{YF}$ and $a_{YR}$ when applying the synaptic constraints, while maintaining a vanishingly small probability of being in state Y ($p_Y < 10^{-18}$). The $\log_e$ likelihood (summed over the 5 cohorts) for model B was 1854 less than for model A, with 30 degrees of freedom for model A (6 per cohort × 5 cohorts) and 20 degrees of freedom for model B (4 per cohort × 5 cohorts). Applying the likelihood ratio test, the difference was highly significant ($p < 10^{-100}$; $p$=Chi-squared($2L$, $df$), where $L$ = 1854 and $df$ = 30–20 =10. Model C is the most general 3-state (F, R, P) model, which allows all six transitions between the three states. The fitted transition rates for model C were nearly identical to model B. Likelihood values are relative to model A.

| | A<br>2 pause states<br>6 free parameters<br>0<br>30<br>mean ± sem ($n$ = 5) | B<br>1 pause state<br>4 free parameters<br>-1854<br>20<br>mean ± sem ($n$ = 5) | C<br>1 pause state<br>6 free parameters<br>-1836<br>30<br>mean ± sem ($n$ = 5) |
|---|---|---|---|
| Δ $\log_e$ likelihood<br>Degrees of freedom | | | |
| $a_{XR}$ (s$^{-1}$) | 1.201 ± 0.099 | 1.019 ± 0.085 | 1.008 ± 0.090 |
| $a_{XF}$ (s$^{-1}$) | 1.115 ± 0.087 | 1.915 ± 0.152 | 1.914 ± 0.152 |
| $a_{RX}$ (s$^{-1}$) | 0.025 ± 0.008 | 0.507 ± 0.013 | 0.507 ± 0.013 |
| $a_{RY}$ (s$^{-1}$) | 0.490 ± 0.015 | $10^{-10}$ | |
| $a_{FX}$ (s$^{-1}$) | 0.182 ± 0.007 | 0.198 ± 0.009 | 0.196 ± 0.008 |
| $a_{FY}$ (s$^{-1}$) | 0.007 ± 0.002 | $10^{-10}$ | |
| $a_{YR}$ (s$^{-1}$) | 0.411 ± 0.019 | >$10^9$ | |
| $a_{YF}$ (s$^{-1}$) | 4.575 ± 0.533 | >$10^9$ | |
| $a_{FR}$ (s$^{-1}$) | | | 0.001 ± 0.001 |
| $a_{RF}$ (s$^{-1}$) | | | 0.000 ± 0.000 |
| $d_F$ (s) | 5.329 ± 0.245 | 5.096 ± 0.235 | 5.135 ± 0.227 |
| $d_R$ (s) | 1.945 ± 0.043 | 1.975 ± 0.049 | 1.976 ± 0.049 |
| $d_X$ (s) | 0.441 ± 0.032 | 0.349 ± 0.026 | 0.351 ± 0.027 |
| $d_Y$ (s) | 0.208 ± 0.019 | <$10^{-9}$ | |
| $p_F$ | 0.762 ± 0.015 | 0.7641 ± 0.015 | 0.764 ± 0.014 |
| $p_R$ | 0.158 ± 0.007 | 0.158 ± 0.007 | 0.155 ± 0.007 |
| $p_X$ | 0.063 ± 0.006 | 0.081 ± 0.008 | 0.080 ± 0.008 |
| $p_Y$ | 0.017 ± 0.002 | <$10^{-18}$ | |

DOI: https://doi.org/10.7554/eLife.12572.014

It has been reported that pauses in *C. elegans* locomotion occur at specific points in "shape space" (*Stephens et al., 2011*), suggesting the worm pauses in preferred postures. To investigate this possibility, we analyzed worm tracks before and after pauses, inferring posture from the path of the tracking spot. This inference is justified by the fact that on an agar surface the worm moves without slipping, such that each segment of the body traces the trajectory of the one before it. Thus, the path of the tracking spot leading up to the pause reveals the worm's posture posterior to the spot during forward locomotion, and anterior to the spot during reverse locomotion (*Figure 4*).

Plotting mean curvature versus distance along the track (*Figure 4A*) reveals only a weak tendency to stop in a particular posture in state X (*r* = 0.14; *Figure 4B*). Nearly all of the transitions into state X were either stutters during forward locomotion (FXF transitions) or reversals (FXR transitions); when these were analyzed separately, similarly weak postural preferences were found at FXF transitions (*r* = 0.14) and FXR transitions (*r* = 0.14). A nearly identical result (*r* = 0.14 ) was obtained using a fixed velocity threshold of 0.05 mm/s rather than the hidden Markov model to determine state. For the latter case, in which there is only one pause state, we analyzed the posture at all FP transitions, which almost always correspond to FX transitions in the hidden Markov model because FY transitions are extremely rare (see *Figure 3* ). To test whether the failure to find a strong postural preference at FX transitions was due to including very short pauses in the analysis, we repeated the analysis after reclassifying all pauses shorter than a minimum duration as a continuation of the previous state, and obtained the same result; we found no strong postural preference at FX transitions for minimum pause durations up to 2 s (*r* = 0.16, 0.19, 0.23, 0.3 for X dwell times > 0.33 s, 0.67 s, 1 s, and 2 s, respectively); longer dwells in state X were too rare to analyze. Thus, FX transitions can occur at any locomotory phase and do not occur preferentially at a particular posture (*Figure 4D*); in the case of FXR transitions the worm generally retreats along the same track. In contrast, we found a strong tendency to stop in a particular posture in state Y (*Figure 4A,C,E*; *r* = 0.71). Almost all entries into state Y were RYF transitions and these were associated with a ventral bend in the middle of the worm (*Figure 4E*). These results suggest fundamental differences between the control of forward and reverse locomotion. In our model, forward locomotion terminates when forward command neurons turn OFF, and this can happen at any phase, whereas reverse locomotion terminates when forward neurons turn ON, and this is most likely to happen at a particular phase. The latter could be explained by phasic feedback from the locomotory pattern generator to the forward neurons (*Li et al., 2006*).

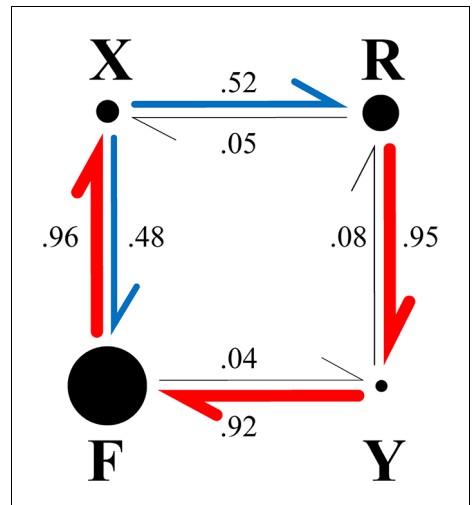

**Figure 3.** Fate diagram of the model. The system typically cycles clockwise through states F, X, R, Y, with state F frequently interrupted by FXF transitions, leading to state sequences of the form ...(FX)$_n$RY(FX)$_n$RY.... Nearly unidirectional transitions out of a given state are shown by red arrows; blue arrows indicate nearly equiprobable transitions. The width of the arrows and the numbers beside them show the probability that the transition out of the state at the tail of the arrow is into the state at the head. The area of each circle is proportional to the probability of the corresponding state (*Table 1*, column A).
DOI: https://doi.org/10.7554/eLife.12572.015

## Ablation of command neurons

To determine the contributions of individual command neurons to the overall function of the command network, we separately ablated the pair of neurons that comprises each command neuron class, then tracked ablated and sham operated animals during local search. Mean velocities in F and R, if significantly changed, were reduced (*Pokala et al., 2014*) (*Figure 5A*; ★★), as was the frequency of undulations during forward and reverse locomotion (*Table 3*). In many organisms, the frequency of rhythmic behaviors is regulated by the amplitude of tonic excitatory drive to the associated pattern generator (*Weeks and Kristan, 1978*; *Satterlie and Norekian, 2001*; *Böhm and Schildberger, 1992*; *Deliagina et al., 2000*; *Dembrow et al., 2003*; *Hedwig, 2000*; *Sirota et al., 2000*). To explain our results we propose that ablation of the locomotory command neurons reduces tonic drive to the presumptive locomotory pattern generator (*Xie et al., 2013*; *Gao et al., 2015*).

A previous study found that ablating a subset of the reverse command neurons (AVAL and AVAR) reduces dwell time in the reverse state but also paradoxically reduces dwell time in the forward state (*Zheng et al., 1999*). Similarly paradoxical effects have been reported following ablation of the reciprocally connected brain stem nuclei that regulate sleep and wakefulness (*Saper et al., 2010*). The stochastic switch

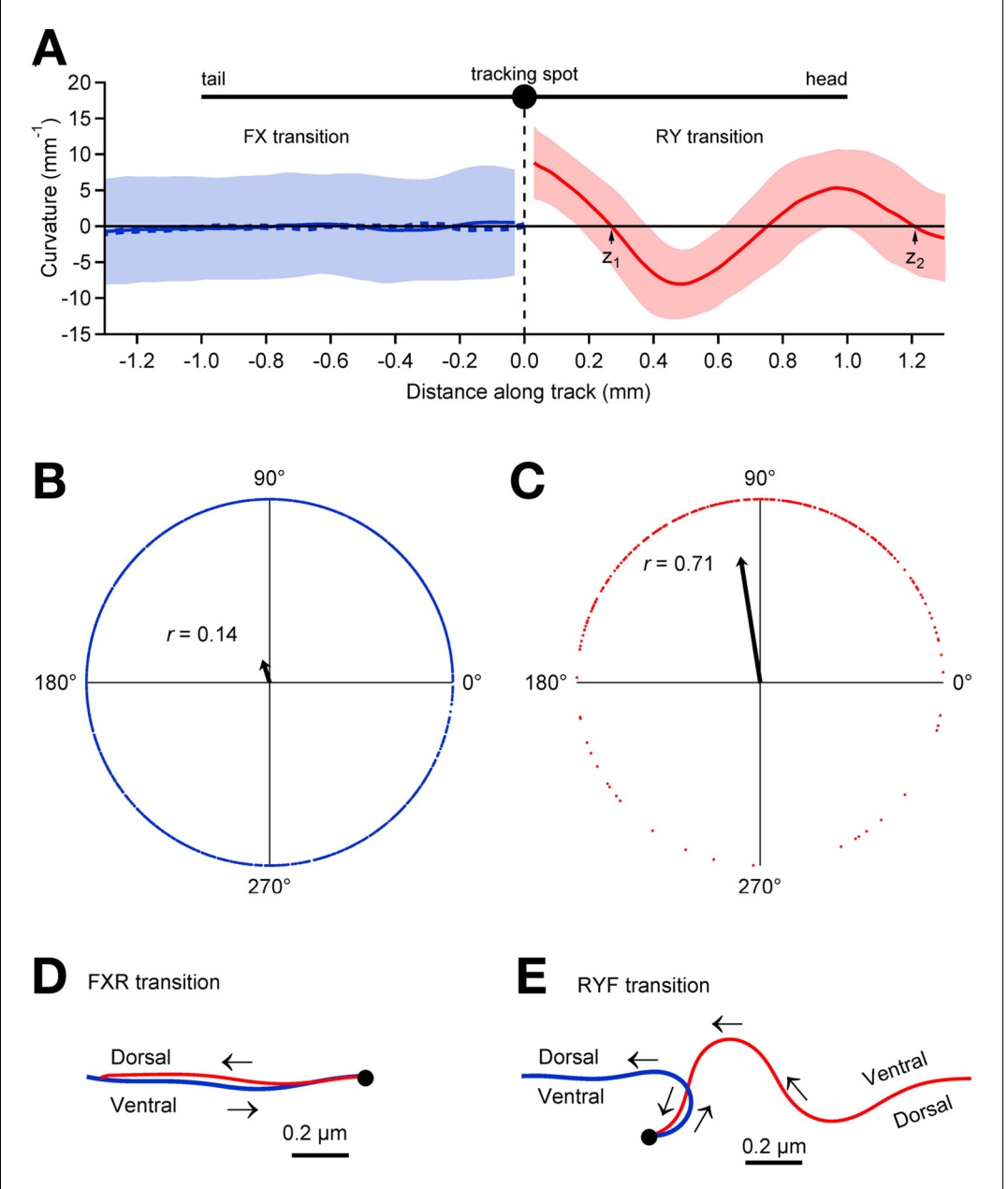

**Figure 4.** Relationship between pauses and posture. (A) Average track curvature upon entry in to the pause state in wild type worms. Prior to computing curvature, tracks of individual worms were mirror-imaged as needed such that positive curvature corresponds to a ventral bend. Tracks in the vicinity of pause events were aligned according to the location of the tracking spot in the pause state, converted to curvature, then averaged over all FX transitions (solid blue line; $n = 1907$), and all RY transitions (red; $n = 295$) for which the track length was >1.5 mm; shading shows ± 1 S.D. The trace depicts the curvature of the worm posterior to the tracking spot at the end of forward movement (FX transitions) and anterior to the tracking spot at the end of reverse movement (RY transitions). The dashed blue line shows the average curvature at FXR transitions (i.e., excluding FXF stutters). (B) Locomotory phases at which FX transitions occurred, plotted as blue dots on the unit circle. The phase at each FX transition was computed as $\varphi = 2\pi z1/(z2 - z1)$, where $z1$ and $z2$ are the positions of the two downward zero crossings of curvature preceding the pause as indicated in panel A, right. The uniform distribution of points around the circle, and therefore the small magnitude of the vector strength ($r = 0.14$; arrow), shows that there was only a small (but statistically significant) phase preference at the end of forward motion ($p < 10 - 16$; Rayleigh test). (C) Same as B, but for RY transitions. Vector strength is large ($r = 0.71$), indicating a strong tendency to end reverse runs at a particular phase ($p < 10 - 63$), with a ventral bend in the middle of the body. (D) Average posture

*Figure 4 continued on next page*

*Figure 4 continued*

at FXR transitions, calculated by integrating the average curvature, computed over all tracks that persisted for >1.5 mm in state F before the pause and >1 mm in state R after the pause. Arrows indicate direction of motion along the track (blue, forward; red, reverse). FXR transitions were typically a simple reversal along the same track. (**E**) Same as D but for RYF transitions that persisted for >1.5 mm in state R before the pause and >1 mm in state F after the pause. RYF transitions at the end of reverse runs that persisted for >1.5 mm were usually associated with a ventral bend that resulted in a ~180° change of direction as previously described (*Gray et al., 2005*).
DOI: https://doi.org/10.7554/eLife.12572.016

model predicts and explains such effects. In principle, the ablation of a subset of neurons in a pool of co-active neurons can have widespread effects on the pool's overall input and output connectivity. Widespread effects can be expected because ablation removes not only the outgoing synaptic connections from the ablated neurons, but also the targets of incoming synaptic connections. In the *C. elegans* command neuron network, ablating a reverse command neuron such as AVA potentially reduces four of the six weights in the network: $h_{\mathcal{R}}$, $W_{\mathcal{RR}}$, $W_{\mathcal{RF}}$, and $W_{\mathcal{FR}}$. Thus, a single ablation can move the system a considerable distance in weight space toward the uncoupled state in which all weights are zero. In the limiting case of a fully uncoupled network, all dwell times approach a value of $1/2A$, where $A$ is the intrinsic switching time of the stochastic units (see Materials and methods, *equations 31–34*); henceforth we will use $d_0$ to denote the uncoupled dwell time. Dwell times that in intact animals are greater than $d_0$ will be reduced by ablation, whereas dwell times that are less than $d_0$ will be increased. In particular, if $d_F$ and $d_R$ are both greater than $d_0$, ablation of a reverse command neuron is expected to reduce both dwell times; the same is true for ablation of a forward command neuron. Thus the observed paradoxical effects of ablations are to be expected if $d_0$ is below $d_F$ and $d_R$.

To determine the actual relationship between $d_0$ and dwell times in the forward and reverse state, we estimated the rate constants in ablated animals and sham operated controls, and computed the corresponding dwell times (*Figure 5B*; *Table 4*). Dwell times in F and R, if significantly altered by the ablation (★★), were reduced, indicating that $d_0$ is indeed below $d_F$ and $d_R$. Additionally, dwell times in the pause states $d_X$ and $d_Y$ were increased, with one exception ($d_Y$, AVB). Thus, the observed pattern of dwell time changes is consistent, overall, with a value of $d_0$ that is between the dwell times of the movement states and the dwell times of the pause states. This finding allowed us to place bounds on $d_0$. Specifically, $d_0$ must be less than or equal to the lowest post-ablation value of $d_R$, and greater than or equal to the largest post-ablation value of $d_X$; thus, $0.58 \leq d_0 \leq 1.24$ sec. Furthermore, because $A = 1/2d_0$, we can infer that $0.40$ Hz $\leq A \leq 0.86$ Hz. This inequality provides an estimate of the fundamental time scale of stochastic switching in *C. elegans* locomotion. For subsequent analysis, we defined $A_{\min} = 0.40$ Hz and $A_{\max} = 0.86$ Hz.

## Synaptic weights in the stochastic switch model

Having placed bounds on $A$, we were able to compute synaptic weights in the model (*Table 2*). This was done by deriving expressions for the weights in terms of the rate constants (Materials and methods, *Equations 36–38*) and substituting into these equations our estimates of rate constants together with the values of $A_{\min}$ and $A_{\max}$. We found that input weights, $h_{\mathcal{F}}$ and $h_{\mathcal{R}}$, are small and positive, suggesting that these inputs may provide modest but steady excitation to the system (*Figure 6A*). The self-connections $W_{\mathcal{FF}}$ and $W_{\mathcal{RR}}$ are also mainly positive, indicating that the ON states may be stabilized by intrinsic or extrinsic positive feedback. The cross-connections $W_{\mathcal{FR}}$ and $W_{\mathcal{RF}}$ are negative, indicating reciprocal inhibition, as expected for neurons that activate opposing behavioral states. Furthermore, the magnitude of $W_{\mathcal{FR}}$ is greater than the magnitude of $W_{\mathcal{RF}}$, suggesting that the animal spends more time in the forward state than the reverse state in part because the forward neurons inhibit the reverse neurons more strongly than the reverse neurons inhibit the forward neurons.

Synaptic weights in an abstract network model such as this one, where neuronal state is activation rather than voltage, are not generally interpretable as synaptic conductances. Rather, they represent the functional effects of one neuron on another, such as the degree of excitation or inhibition produced by a unit change in activation. Thus, synaptic weights in the Stochastic Switch model cannot be said to predict the magnitude of synaptic conductances, but they can be said to predict aspects

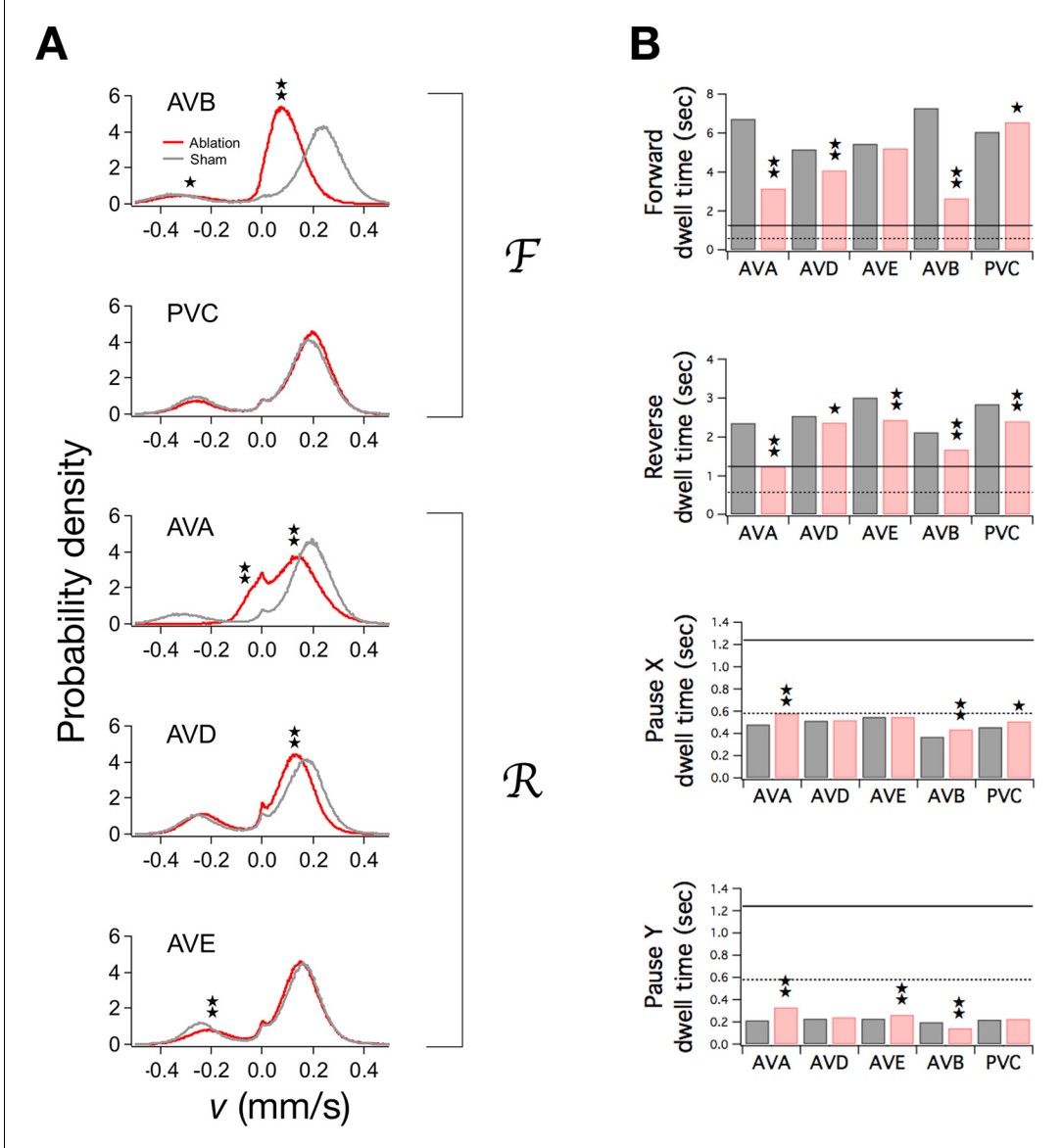

**Figure 5.** Ablation of command neurons. (**A**) Velocity distribution of ablated cohorts (red) compared to sham operated controls (grey) when the indicated command neuron was killed. Stars indicate significant reduction in velocity for the indicated peak ($p < 0.05$ without (★) or with (★★) correction for multiple comparisons; *Table 3*). (**B**) Dwell times in F, R, and P in ablated (red) and sham operated animals (grey). Stars indicate significant differences from sham (as defined in *Table 4*). Horizontal lines indicate the estimated range of $d0$, the dwell time in the uncoupled state. Each group of ablated animals was tested in parallel with a distinct set of sham operated controls to minimize the effects of variation between populations. Error bars for dwell times are not shown because statistical significance was calculated using the likelihood ratio test (see *Table 4* legend), which does not generate sem estimates, and calculation of confidence intervals would have required an excessive amount of computation time. Stars indicate $p < 0.05$ without (★) or with (★★) correction for multiple comparisons (*Table 4*).

DOI: https://doi.org/10.7554/eLife.12572.017

of functional connectivity in certain cases. For example, as command neurons AVA and AVB are behaviorally much more important than the others (*Chalfie et al., 1985*) (see also *Figure 5A,B*), it is reasonable to assume that the signs of their functional synaptic connections match the signs of the net functional connections in the biological network. Thus, the model predicts reciprocal inhibition (*Qi et al., 2012*) between AVA and AVB under this assumption. We tested this prediction by photo-activating either AVA or AVB with channelRhodopsin-2 and recording electrophysiologically from AVB or AVA, respectively (*Figure 6B,C*). We found that the reversal potential of optically induced synaptic currents in AVA and AVB was more negative than the zero-current potential in these

**Table 2.** Synaptic weights derived from the transition rate constants.
The rate constants were taken from **Table 1**, column A. Two values of the fundamental switching time, $A$, corresponding to the minimum (0.40 Hz) and maximum (0.86 Hz) values consistent with the ablation results were used in Materials and methods, **Equations 36–38** to calculate the corresponding synaptic weights.

| | $A = 0.4$ Hz<br>mean ± sem ($n = 5$) | $A = 0.86$ Hz<br>mean ± sem ($n = 5$) |
|---|---|---|
| $h_{\mathcal{F}}$ | 1.01 ± 0.08 | 0.25 ± 0.08 |
| $h_{\mathcal{R}}$ | 1.09 ± 0.08 | 0.32 ± 0.08 |
| $wFR$ | -5.40 ± 0.43 | -5.40 ± 0.43 |
| $W_{\mathcal{RF}}$ | -0.81 ± 0.06 | -0.81 ± 0.06 |
| $W_{\mathcal{FF}}$ | -0.22 ± 0.06 | 1.31 ± 0.06 |
| $W_{\mathcal{FR}}$ | 1.90 ± 0.33 | 3.43 ± 0.33 |

DOI: https://doi.org/10.7554/eLife.12572.018

neurons (**Figure 6B,C,D**), indicating synaptic inhibition as predicted by the model. This inhibition is likely to be monosynaptic as *C. elegans* command neurons are cholinergic and express inhibitory postsynaptic receptors that respond to acetylcholine (**Pereira et al., 2015**). Additionally, the connection from AVB to AVA appeared to be stronger thaCroninn the connection from AVA to AVB (**Figure 6E**), measured in terms of the amplitude of the synaptic current at a holding potential approximately equal to the membrane potential when command neurons are in their depolarized state (**Figure 2B**). However, we do not exclude the possibility that AVB was more strongly activated than AVA as a result of differential expression of the photoprobe. These findings demonstrate the feasibility of using the worm's velocity, $v(t)$, a simple behavioral measure, to predict functional synaptic connections between populations of neurons in a biological neural network, at least under certain assumptions concerning the relationship between model network weights and physiological synaptic strengths.

## Genetic effects on command neuron function

Two classes of ion channel mutants that affect membrane conductances in the command neurons are also known to alter locomotory behavior in systematic ways, thus providing key insights into command neuron function (**Zheng et al., 1999**). The hyperpolarizing class ("HYP") comprises three genotypes in which release of the excitatory neurotransmitter glutamate, presumed to be tonic, is disrupted by mutations that affect either presynaptic (*eat-4(ad572)*, *eat-4(ky5)*) or postsynaptic mechanisms (*glr-1(n2461)*). These mutations are hypothesized to cause chronic hyperpolarization of the command neurons by reducing depolarizing currents. The depolarizing class ("DEP") comprises two genotypes in which a constitutively activated glutamate receptor is expressed in the command neurons (*glr-1::glr-1(A/T)*, *nmr-1::glr-1(A/T)*). These mutants are hypothesized to chronically depolarize the command neurons.

**Table 3.** Effects of command neuron ablations on undulation frequency, forward velocity and reverse velocity.
Values were computed separately for each worm and are shown as mean ± sem ($n = 19$–$29$). Undulation frequency was estimated as one-half of the reciprocal of the time of the first local minimum in the heading autocorrelation function. All *p*-values are from two-tailed *t*-tests and are shown without correction for multiple comparisons. Blue denotes significance at p<0.05. Red denotes significance at p<0.05 after Bonferroni correction for 15 comparisons.

| Neuron | Class | Undulation frequency (Hz)<br>Sham | Ablated | *p* < | Forward velocity (µm/s)<br>Sham | Ablated | *p* < | Reverse velocity (µm/s)<br>Sham | Ablated | *p* < |
|---|---|---|---|---|---|---|---|---|---|---|
| AVB | Forward | 0.355 ± 0.009 | 0.230 ± 0.007 | $7 \times 10^{-11}$ | 236 ± 6 | 109 ± 4 | $5 \times 10^{-20}$ | -327 ± 7 | -302 ± 8 | 0.04 |
| PVC | Forward | 0.283 ± 0.011 | 0.290 ± 0.010 | 0.5 | 187 ± 7 | 192 ± 7 | 0.7 | -253 ± 8 | -248 ± 6 | 0.7 |
| AVD | Reverse | 0.270 ± 0.008 | 0.236 ± 0.008 | 0.009 | 173 ± 6 | 141 ± 5 | 0.0002 | -243 ± 4 | -229 ± 5 | 0.06 |
| AVA | Reverse | 0.302 ± 0.005 | 0.254 ± 0.009 | $4 \times 10^{-5}$ | 195 ± 5 | 155 ± 7 | $4 \times 10^{-5}$ | -293 ± 7 | -69 ± 3 | $3 \times 10^{-22}$ |
| AVE | Reverse | 0.264 ± 0.007 | 0.256 ± 0.008 | 0.6 | 165 ± 4 | 160 ± 5 | 0.5 | -235 ± 4 | -211 ± 6 | 0.003 |

DOI: https://doi.org/10.7554/eLife.12572.019

**Table 4.** Effects of command neuron ablations on model parameters.

The sign of the change (Δ) caused by the ablation is shown as "+" if the value moved away from 0, "−" if the value moved towards 0. Significance was determined using the likelihood ratio test (Weisstein, Eric W. "Likelihood Ratio." From MathWorld–A Wolfram Web Resource. http://mathworld.wolfram. com/LikelihoodRatio. html), which is based on the reduction in likelihood caused by constraining one of the parameters to have the same value in both the ablated cohort and the corresponding sham cohort. The unconstrained fit thus had 12 free parameters (6 for each of the 2 cohorts being compared), while the constrained fit had 11 free parameters. For example, to test the significance of the change in the mean dwell time in the pause state ($dP = (pXdX + pYdY)/(pX + pY)d_p = (p_xd_x + p_Yd_Y)/(p_x + p_y)$) caused by ablation of the AVA neuron pair, two cohorts (ablated and sham) were grown and tested under identical conditions. The ln likelihood with 12 free parameters was found to be 894794.075. When $d_p$ was constrained to be the same for both cohorts, the ln likelihood for the 11 parameter fit was found to be 894784.676. The test statistic $D = 2 \times (894794.075 − 894784.676) = 18.798$ was assumed to come from a chi-squared distribution with one degree of freedom, which yielded $p = 1.45 \times 10 − 5$ (shown in the table as $p < 10 − 4$). The constrained fitting process was repeated in turn for each ablation/sham pair for each of the 9 rows shown in the table. All p-values are shown without correction for multiple comparisons. Blue denotes significance at p<0.05. Red denotes significance at p<0.05 after Bonferroni correction for 27 comparisons

| | REVERSE | | | | | | | | | | | | FORWARD | | | | | | | |
| | AVE | | | | AVD | | | | AVA | | | | AVB | | | | PVC | | | |
| | Sham | Ablate | Δ | p< | Sham | Ablate | Δ | p< | Sham | Ablate | Δ | p< | Sham | Ablate | Δ | p< | Sham | Ablate | Δ | p< |
|---|---|---|---|---|---|---|---|---|---|---|---|---|---|---|---|---|---|---|---|---|
| $d_F$ (s) | 5.455 | 5.221 | − | 0.2 | 5.158 | 4.081 | − | $10^{-15}$ | 6.730 | 3.143 | − | $10^{-99}$ | 7.289 | 2.642 | − | $10^{-99}$ | 6.058 | 6.558 | + | 0.02 |
| $d_R$ (s) | 3.019 | 2.436 | − | $10^{-6}$ | 2.540 | 2.367 | − | 0.05 | 2.359 | 1.243 | − | $10^{-41}$ | 2.127 | 1.681 | − | $10^{-6}$ | 2.842 | 2.396 | − | 0.0005 |
| $d_X$ (s) | 0.548 | 0.548 | − | 1 | 0.514 | 0.520 | + | 0.6 | 0.480 | 0.582 | + | $10^{-7}$ | 0.370 | 0.437 | + | $10^{-6}$ | 0.457 | 0.508 | + | 0.004 |
| $d_Y$ (s) | 0.229 | 0.263 | + | 0.002 | 0.229 | 0.241 | + | 0.07 | 0.214 | 0.331 | + | $10^{-7}$ | 0.197 | 0.144 | − | $10^{-7}$ | 0.220 | 0.226 | + | 0.5 |
| $d_p$ (s) | 0.495 | 0.496 | + | 1 | 0.460 | 0.468 | + | 0.5 | 0.437 | 0.510 | + | $10^{-4}$ | 0.331 | 0.416 | + | $10^{-11}$ | 0.410 | 0.457 | + | 0.003 |
| $p_F$ | 0.720 | 0.747 | + | 0.004 | 0.723 | 0.689 | − | 0.0005 | 0.809 | 0.704 | − | $10^{-24}$ | 0.818 | 0.745 | − | $10^{-15}$ | 0.749 | 0.787 | + | 0.0002 |
| $p_R$ | 0.192 | 0.158 | − | $10^{-4}$ | 0.188 | 0.203 | + | 0.05 | 0.122 | 0.137 | + | 0.04 | 0.129 | 0.120 | − | 0.2 | 0.181 | 0.139 | − | $10^{-9}$ |
| $p_X$ | 0.073 | 0.078 | + | 0.07 | 0.072 | 0.088 | + | $10^{-7}$ | 0.058 | 0.113 | + | $10^{-9}$ | 0.041 | 0.125 | + | $10^{-99}$ | 0.056 | 0.061 | + | 0.2 |
| $p_Y$ (s) | 0.014 | 0.017 | + | 0.02 | 0.017 | 0.020 | + | 0.005 | 0.011 | 0.046 | + | $10^{-23}$ | 0.012 | 0.010 | − | 0.01 | 0.014 | 0.013 | − | 0.5 |

DOI: https://doi.org/10.7554/eLife.12572.020

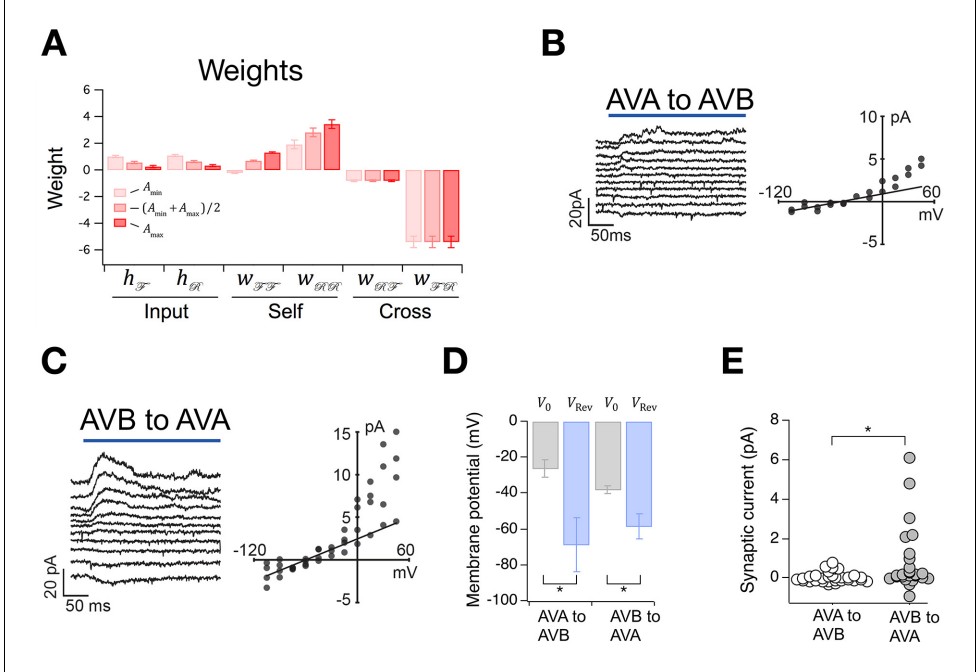

**Figure 6.** The Stochastic Switch Model correctly predicts the sign and strength of synaptic connections. (A) Synaptic weights (mean ± sem, n = 5 cohorts) from maximum likelihood fits to velocity data from wild type worms. (B, C) *Left*, synaptic current in AVB or AVA when the indicated presynaptic neuron was photoactivated (blue line). *Right*, mean synaptic current during the first 100 ms of the stimulus plotted against holding potential in the postsynaptic neuron (I-V curve). Lines show linear fits to the data at negative holding potentials which were used to estimate *vRev*. (D), Zero-current holding potential and reversal potential of synaptic currents (mean ± sem) in the indicated postsynaptic neuron (paired *t*-tests: AVA to AVB, *p*= 0.043, n = 9; AVB to AVA, *p*= 0.019, n = 17). (E), Scatter plot of synaptic currents recorded at a holding potential of -15 mV (unpaired *t*-test: *p*= 0.010, n ≥ 25).
DOI: https://doi.org/10.7554/eLife.12572.021

We found that the frequency of locomotory undulations was decreased in HYP mutants and increased in DEP mutants compared to wild-type controls (*Table 5*), consistent with the likely effects of respectively increasing and decreasing tonic drive to the presumptive pattern generator for locomotion. Importantly, however, it is possible that both classes of mutation also alter the input resistance of the command neurons. The closure or removal of glutamate receptors in HYP mutants should increase input resistance whereas the introduction of constitutively active glutamate receptors in DEP mutants should decrease it. Thus, the previously observed effects of these mutations on locomotory state transitions (*Zheng et al., 1999*) could be the result of changes in membrane potential ($\Delta V$), input resistance ($\Delta r$), or both.

Changes in membrane potential and input resistance can both be represented in the stochastic switch model by changes in synaptic weights. We modeled the effects of $\Delta V$ by adding an increment $\Delta h$ ($-1 \leq \Delta h \leq 1$) to wild type $h$ values, with negative $\Delta h$ for HYP mutations and positive $\Delta h$ for DEP mutations. We modeled the effect of $\Delta r$ as a change in the magnitude of synaptic weights ($h$ and $w$ quantities). This representation of $\Delta r$ is appropriate because changes in input resistance alter the magnitude of the voltage change that would be produced by a fixed postsynaptic current. All weights were scaled by a common factor $z$ ($1 < z < 2$ for HYP mutants; $0 < z < 1$ for DEP mutants).

Here we consider the effects of $\Delta V$ and $\Delta r$ on dwell times in the stochastic switch model to enable direct comparison with the original study of HYP and DEP strains (*Zheng et al., 1999*). Dwell times can be written as functions of weights:

$$d_\text{X} = (a_\text{XF} + a_\text{XR})^{-1} = [A \exp(h_\mathscr{F}) + A \exp(h_\mathscr{R})]^{-1} \tag{1}$$

$$d_\text{F} = (a_\text{FX} + a_\text{FY})^{-1} = [A \exp(-h_\mathscr{F} - w_{\mathscr{F}\mathscr{F}}) + A \exp(h_\mathscr{R} + w_{\mathscr{F}\mathscr{R}})]^{-1} \tag{2}$$

$$d_\text{R} = (a_\text{RX} + a_\text{RY})^{-1} = [A \exp(-h_\mathscr{R} - w_{\mathscr{R}\mathscr{R}}) + A \exp(h_\mathscr{F} + w_{\mathscr{R}\mathscr{F}})]^{-1} \tag{3}$$

$$d_{\mathrm{Y}} = (a_{\mathrm{YR}} + a_{\mathrm{YF}})^{-1} = [A\exp(-h_{\mathscr{F}} - w_{\mathscr{FF}} - w_{\mathscr{RF}}) + A\exp(-h_{\mathscr{R}} - w_{\mathscr{RR}} - w_{\mathscr{FR}})]^{-1} \qquad (4)$$

These equations show that the $\Delta V$ and $\Delta r$ hypotheses make qualitatively distinct predictions. The simplest case is dwell $d_X$, which depends only on $h_{\mathscr{F}}$ and $h_{\mathscr{R}}$. *Equation 1* shows that $d_X$ rises and falls as $h$ terms are made more negative or positive, respectively. Thus, under the $\Delta V$ hypothesis, $d_X$ should rise in HYP mutants and fall in DEP mutants (*Figure 7A*, row 4). In contrast, under the $\Delta r$ hypothesis, in which weight magnitudes ($|W|$ and $|h|$) decrease in DEP mutants and increase in HYP mutants, $d_X$ should rise in DEP mutants and fall in HYP. To distinguish between these hypotheses, we measured dwell times in mutants and wild type animals during local search. The pattern of observed changes in $d_X$ matched the pattern predicted by the $\Delta V$ hypothesis but not the $\Delta r$ hypothesis (*Figure 7C*, row 4). Thus, the effects of membrane potential appear to dominate the effects of changes in synaptic strength in the case of mutant values of $d_X$.

In contrast to $d_X$, $d_F$ and $d_R$ depend on $w$ terms as well as $h$ terms. Under the $\Delta V$ hypothesis, the $h$ terms but not the $w$ terms would be affected by the mutations. Positive and negative increments in $h$ have the effects shown in *Figure 7A*, rows 1 and 2; $d_F$ and $d_R$ are predicted to shift in opposite directions. Changes in $d_F$ are dominated by the effects of $h_{\mathscr{F}}$ on the first term in *Equation 2* (which represents $a_{FX}$) because the second term in the equation (which represents $a_{FY}$) remains close to zero in the mutants. Analogously, changes in $d_R$ are dominated by the effects of $h_{\mathscr{F}}$ on the second term in *Equation 3* ($a_{RY}$) because the first term in the equation ($a_{RX}$) remains close to zero in the mutants.

The $\Delta r$ hypothesis makes a distinctly different prediction. In this version of the model, $w$ terms and $h$ terms would both be affected by the mutations. Now, the predicted pattern of dwell time changes across both $d_F$ and $d_R$ is such the both dwell times shift in the same direction (*Figure 7B*, rows 1 and 2); specifically, dwell times in DEP and HYP mutants move toward or away from their uncoupled dwell times, respectively. Taken together, the pattern of observed changes in $d_F$ and $d_R$ matched the pattern predicted by the $\Delta r$ hypothesis (*Figure 7C*, rows 1 and 2) but not the $\Delta V$ hypothesis. We conclude that changes in synaptic strength may dominate the effects of changes in membrane potential on mutant values of $d_F$ and $d_R$.

Neither hypothesis predicts the observed changes in $d_Y$ (*Figure 7C*, row 5) which resembled the pattern of changes in $d_X$ (*Figure 7C*, row 4). However, the $\Delta V$ hypothesis correctly predicts observed dwell times in the overall pause state $d_p$ (*Figure 7C* row 3). This is because $d_p$ is dominated by $d_X$ and changes in $d_X$ are well-predicted by the $\Delta V$ model as noted above. Overall, our analysis of the effects of HYP and DEP mutations in terms of the Stochastic Switch Model points to a role for changes in both membrane potential and input resistance in regulating dwell times.

## Regulation of search scale

The Stochastic Switch Model immediately suggests a family of models for the regulation of the spatial scale of random search in response to the availability of food and the worm's physiological state. The scale of random search is determined primarily by $m_F$, the mean distance traveled during a forward run. In *C. elegans*, a run begins with a transition from state R (via P) into state F and continues until the next transition into state R. Any run may include one or more visits to state P, but FPF

**Table 5.** Effects of mutations on mean undulation frequency, mean forward velocity and mean reverse velocity.
Values were computed separately for each worm and are shown as mean ± sem ($n = 25$–$31$). Undulation frequency was estimated as one-half of the reciprocal of the time of the first local minimum in the heading autocorrelation function. All $p$-values are from two-tailed $t$-tests and are shown without correction for multiple comparisons. Blue denotes significance at $p<0.05$. Red denotes significance at $p<0.05$ after Bonferroni correction for 15 comparisons.

| Genotype | Class | Undulation frequency (Hz) Wild type | Mutant | $p <$ | Forward velocity (µm/s) Wild type | Mutant | $p <$ | Reverse velocity (µm/s) Wild type | Mutant | $p <$ |
|---|---|---|---|---|---|---|---|---|---|---|
| *eat-4(ad572)* | HYP A | 0.272 ± 0.011 | 0.222 ± 0.007 | $4\times10^{-4}$ | 156 ± 5 | 122 ± 4 | $1\times10^{-5}$ | -228 ± 5 | -236 ± 9 | 0.5 |
| *eat-4(ky5)* | HYP B | 0.317 ± 0.011 | 0.256 ± 0.009 | $2\times10^{-4}$ | 184 ± 7 | 143 ± 6 | $5\times10^{-5}$ | -262 ± 10 | -271 ± 7 | 0.5 |
| *glr-1(n2461)* | HYP C | 0.294 ± 0.008 | 0.291 ± 0.010 | 0.9 | 158 ± 5 | 166 ± 6 | 0.3 | -236 ± 6 | -236 ± 5 | 1 |
| *glr-1::glr-1(A/T)* | DEP A | 0.272 ± 0.011 | 0.642 ± 0.029 | $6\times10^{-13}$ | 156 ± 5 | 112 ± 5 | $3\times10^{-7}$ | -228 ± 5 | -143 ± 5 | $2\times10^{-15}$ |
| *nmr-1::glr-1(A/T)* | DEP B | 0.294 ± 0.008 | 0.695 ± 0.037 | $2\times10^{-12}$ | 158 ± 5 | 138 ± 5 | 0.011 | -236 ± 6 | -144 ± 5 | $7\times10^{-15}$ |

DOI: https://doi.org/10.7554/eLife.12572.022

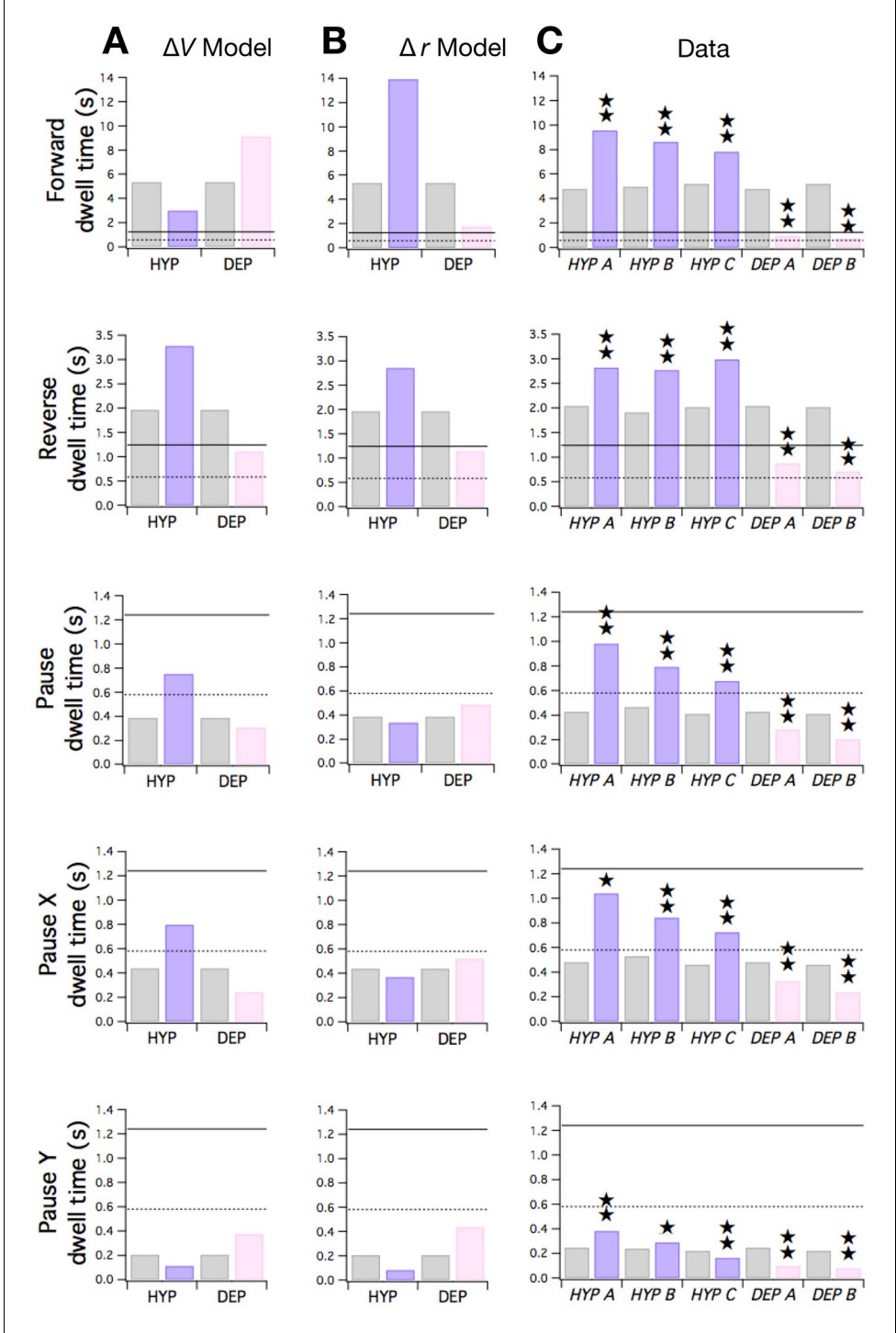

**Figure 7.** Predicted and observed effects of HYP and DEP mutations on dwell times. (A) Predicted effects of changes in membrane potential. (B) Predicted effects of changes in input resistance. (C) Dwell times in F, R, and P for cohorts of HYP mutants, DEP mutants, and wild type animals. Stars indicate significant change in dwell time ($p<0.05$ without (★) or with (★★) correction for multiple comparisons; *Table 6*). In A-C wild type dwell times are indicated by gray bars. Horizontal lines indicate the estimated range of $d_0$, the dwell time in the uncoupled state. In the $\Delta V$ model, $h$ terms were made more negative to model HYP mutants and more positive to model DEP mutants by subtracting or adding a constant $\Delta h = 0.6$; qualitatively similar results were obtained for $0 < \Delta h \leq 0.8$. In the $\Delta r$ model, $h$ and $w$ terms were scaled by $(1 + f)$ to model HYP mutants and by $(1 - f)$ to model DEP mutants, with $f = 0.6$; qualitatively similar results were obtained for $0 < f \leq 1$. Strains, *HYP A*: DA572 *eat-4(ad572)*; *HYP B*: MT6308 *eat-4(ky5)*; *HYP C*: KP4 *glr-1(n2461)*; *DEP A*: VM1136 *lin-15(n765); akIs9 [lin-15(+), Pglr-1::GLR-1(A/T)]*; *DEP B*: VM188 *lin-15(n765); akEx52[lin-15(+), Pnmr-1::GLR-1(A/T)]*.
DOI: https://doi.org/10.7554/eLife.12572.023

**Table 6.** Effects of mutations on model parameters.

Significance was determined using the likelihood ratio test as described in *Table 4*. The sign of the change (Δ) caused by the mutation is shown as "+" if the value moved away from 0, "−" if the value moved towards 0. All *p*-values are shown without correction for multiple comparisons. Blue denotes significance at p<0.05. Red denotes significance at p<0.05 after Bonferroni correction for 27 comparisons.

|  | HYP | | | | | | | | | | | | DEP | | | | | | | |
|  | HYP A: *eat-4(ad572)* | | | | HYP B: *eat-4(ky5)* | | | | HYP C: *glr-1(n2461)* | | | | DEP A: *glr-1::glr-1(A/T)* | | | | DEP B: *nmr-1::glr-1(A/T)* | | | |
|  | Control | Mutant | Δ | p< | Control | Mutant | Δ | p< | Control | Mutant | Δ | p< | Control | Mutant | Δ | p< | Control | Mutant | Δ | p< |
|---|---|---|---|---|---|---|---|---|---|---|---|---|---|---|---|---|---|---|---|---|
| $d_F$ (s) | 4.771 | 9.564 | + | $10^{-87}$ | 4.956 | 8.643 | + | $10^{-64}$ | 5.181 | 7.871 | + | $10^{-33}$ | 4.771 | 0.940 | − | $10^{-99}$ | 5.181 | 0.742 | − | $10^{-99}$ |
| $d_R$ (s) | 2.043 | 2.821 | + | $10^{-7}$ | 1.910 | 2.769 | + | $10^{-12}$ | 2.018 | 3.004 | + | $10^{-16}$ | 2.045 | 0.875 | − | $10^{-99}$ | 2.018 | 0.709 | − | $10^{-9\,9}$ |
| $d_X$ (s) | 0.481 | 1.040 | + | 0.005 | 0.529 | 0.844 | + | $10^{-43}$ | 0.459 | 0.727 | + | $10^{-39}$ | 0.482 | 0.328 | − | $10^{-49}$ | 0.460 | 0.235 | − | $10^{-99}$ |
| $d_Y$ (s) | 0.247 | 0.382 | + | $10^{-5}$ | 0.238 | 0.290 | + | 0.005 | 0.221 | 0.164 | − | $10^{-5}$ | 0.247 | 0.097 | − | $10^{-93}$ | 0.221 | 0.079 | − | $10^{-99}$ |
| $d_P$ (s) | 0.428 | 0.982 | + | $10^{-99}$ | 0.466 | 0.793 | + | $10^{-52}$ | 0.409 | 0.677 | + | $10^{-43}$ | 0.428 | 0.286 | − | $10^{-61}$ | 0.409 | 0.204 | − | $10^{-99}$ |
| $p_F$ | 0.729 | 0.839 | + | $10^{-27}$ | 0.734 | 0.832 | + | $10^{-26}$ | 0.755 | 0.785 | + | 0.003 | 0.728 | 0.410 | − | $10^{-99}$ | 0.755 | 0.407 | − | $10^{-99}$ |
| $p_R$ | 0.177 | 0.062 | − | $10^{-37}$ | 0.167 | 0.079 | − | $10^{-29}$ | 0.161 | 0.135 | − | $10^{-4}$ | 0.177 | 0.389 | + | $10^{-99}$ | 0.161 | 0.404 | + | $10^{-99}$ |
| $p_X$ | 0.073 | 0.090 | + | $10^{-5}$ | 0.077 | 0.081 | + | 0.3 | 0.067 | 0.073 | + | 0.03 | 0.073 | 0.164 | + | $10^{-99}$ | 0.067 | 0.151 | + | $10^{-99}$ |
| $p_Y$ | 0.022 | 0.009 | − | $10^{-13}$ | 0.021 | 0.008 | − | $10^{-25}$ | 0.018 | 0.007 | − | $10^{-27}$ | 0.022 | 0.037 | + | $10^{-21}$ | 0.018 | 0.037 | + | $10^{-41}$ |

DOI: https://doi.org/10.7554/eLife.12572.024

transitions are not usually associated with changes in heading. In terms of the Stochastic Switch Model, $m_F = \overline{V}_F p_F / f_{RPF}$, where $\overline{V}_F$ is the average velocity in state F, $p_F$ is the probability of being in state F, and $f_{RPF}$ is the frequency of RPF transitions (Materials and methods, *Equation 39*), which coincide with random reorientations. Importantly, under the approximation $a_{FY} \cong 0$ (*Table 1*, column A), $m_F$ is can be expressed as a function of just three of the six weights in the network:

$$m_{\mathrm{F}} \cong \frac{\overline{v_{\mathrm{F}}}}{A} \cdot \frac{\exp(h_{\mathscr{F}}) + \exp(h_{\mathscr{R}})}{\exp(h_{\mathscr{R}} - h_{\mathscr{F}} - w_{\mathscr{F}\mathscr{F}})} \tag{5}$$

We refer to these weights as potential control points in the network. In a minimal model of search scale regulation, $m_F$ could be controlled by sensory inputs represented by $h_{\mathscr{F}}$ and $h_{\mathscr{R}}$ (*Figure 8A*).

Search scale ($m_F$) together with the frequency of reversals (FPR transitions), have been used to define the three search modes commonly recognized in *C. elegans*: cropping, local search, and ranging. To find minimal models for regulation of search mode, we performed exhaustive searches of subregions of network's six-dimensional weight space. Subspaces, defined by one, two, or three weights, were scanned across a wide range of values ($-6 \leq w \leq 6$) while other weights remained fixed at their wild type levels (*Figure 7B–H*). The performance of each configuration of the network was scored according to whether it matched the range of $m_F$ magnitudes and reversal frequencies characteristic of each mode (see Materials and methods). Another consideration was the number of distinct search modes available; accordingly, we also noted the density with which the plane defined by reversal frequency and $m_F$ was covered in the scan (Figure. 8B-H, gray symbols).

All three search modes were available in the subspace defined by the control points ($h_{\mathscr{F}}$, $h_{\mathscr{R}}$, $W_{\mathscr{F}\mathscr{F}}$) (*Figure 8B*, *Figure 8—figure supplement 1*). However, only cropping and local search were available in the complementary subspace ($W_{\mathscr{R}\mathscr{R}}$, $W_{\mathscr{R}\mathscr{F}}$, $W_{\mathscr{F}\mathscr{R}}$) (*Figure 8C*); thus, to achieve the full set of search modes, at least one of the weights in *equation 5* must be free to change. None of the control-point weights was sufficient on its own to produce all three search modes (*Figure 8D–F*). Scanning the subspaces ($h_{\mathscr{F}}$, $W_{\mathscr{F}\mathscr{F}}$) and ($h_{\mathscr{R}}$, $W_{\mathscr{F}\mathscr{R}}$) showed these pairs of weights to be sufficient for all modes (*Figure 8G,H*), but a three-dimensional subspace containing at least one of the control-point weights was a necessary condition for both dense coverage of this plane and the presence of all three search modes (*Table 7*). We suggest that these three-weight subspaces constitute the most likely minimal models for the regulation of search in *C. elegans*. They could be tested by chronic manipulation of control-point weights utilizing a variety of approaches, such as chemical or optical probes that alter tonic inputs to the command network from sensory neurons and interneurons represented by the parameters $h_{\mathscr{F}}$ and $h_{\mathscr{R}}$.

## Biased random walks

Mean forward run length is also modulated during biased random walks, increasing or decreasing when the animal is moving in a favorable or unfavorable direction, respectively (*Pierce-Shimomura et al., 1999*; *Block et al., 1982*; *Iino and Yoshida, 2009*; *Luo et al., 2014*). When *C. elegans* is engaged in chemotaxis toward an attractive substance, the direction of motion relative to the gradient is represented by specialized chemosensory neurons that respond either to increases (ON cells) or decreases in concentration (OFF cells) (*Thiele et al., 2009*) ; moreover, interventions that activate ON cells or OFF cells promote runs and pirouettes, respectively (*Suzuki et al., 2008*). Thus, in one simple model of random-walk chemotaxis, ON cells increase $h_{\mathscr{F}}$ and decrease $h_{\mathscr{R}}$, whereas OFF cells do the opposite. Simulations show that this model is sufficient to generate realistic chemotaxis in a point model of search behavior in *C. elegans* (*Figure 8—figure supplement 2*) when the worm is below the target concentration of attractant. Similar circuitry can explain biased random walks in response to other physical gradients (*Lockery, 2011*).

## The Stochastic Switch Model and deterministic behaviors

In addition to random search, the command neurons in *C. elegans* are required for a variety of escape responses (*Hart et al., 1995*) that are deterministic in that $p_R$ closely approaches unity for strong stimuli (*Wittenburg and Baumeister, 1999*; *Tobin et al., 2002*; *Liu et al., 2012*; *Mohammadi et al., 2013*). *C. elegans* escape responses can be produced by two pathways, one that requires the reverse command neurons (*Chalfie et al., 1985*) and one that does not (*Piggott et al., 2011*). Three distinct circuit motifs for the functional connectivity underlying escape

responses requiring reverse command neurons are conceivable (*Figure 9A*). In the Push motif, nociceptive neurons excite reverse command neurons via $h_\mathcal{R}$ thereby increasing the rate constants for transitions in which $\mathcal{R}$ turns ON ($a_{XR}$ and $a_{FY}$), and decreasing the rate constants for transitions in which $\mathcal{R}$ turns OFF ($a_{XR}$ and $a_{FY}$). In the limit where $hR \to \infty$, both $a_{XR}$ and $a_{FY} \to \infty$, whereas $a_{RX}$ and $a_{YF} \to 0$ (*Figure 9B*). The system now inhabits only states R and Y, and $p_R = a_{YR}/(a_{YR} + a_{RY})$. In the Pull motif, nociceptive neurons inhibit the forward command neurons via $hF$. In the limit where $hF \to -\infty$, the system switches only between states R and X and $p_R = a_{XR}/(a_{XR} + a_{RX})$. In the third motif, in which Push and Pull are combined, R becomes an absorbing state ($p_R = 1$). Using the rate constants shown in column A of *Table 1* to compute limiting values of $p_R$ in each motif, we found that the Pull and Push-Pull motifs are sufficient for deterministic escape, whereas the Push motif is not (*Figure 9B*). Thus, inhibition of forward command neurons is required for deterministic escape, predicting that nociceptive neurons functionally inhibit these neurons.

To test this prediction we examined the ASH neurons, a pair of nociceptive sensory neurons required for the majority of escape responses in *C. elegans*. ASH neurons have anatomically defined monosynaptic and polysynaptic connections to both the behaviorally dominant command neurons AVB and AVA (*Chalfie et al., 1985*; *White et al., 1986*). We have previously shown that the functional connection from ASH to AVA is excitatory (*Lindsay et al., 2011*). To test whether the functional connection from ASH to AVB is inhibitory, we photoactivated ASH neurons while recording from AVB (*Figure 9C,D*). The reversal potential of this connection was more negative than the zero current potential, indicating inhibition as predicted by the model. Thus, ASH-mediated escape may be controlled by a Push-Pull motif, further demonstrating the feasibility of using behavioral data to

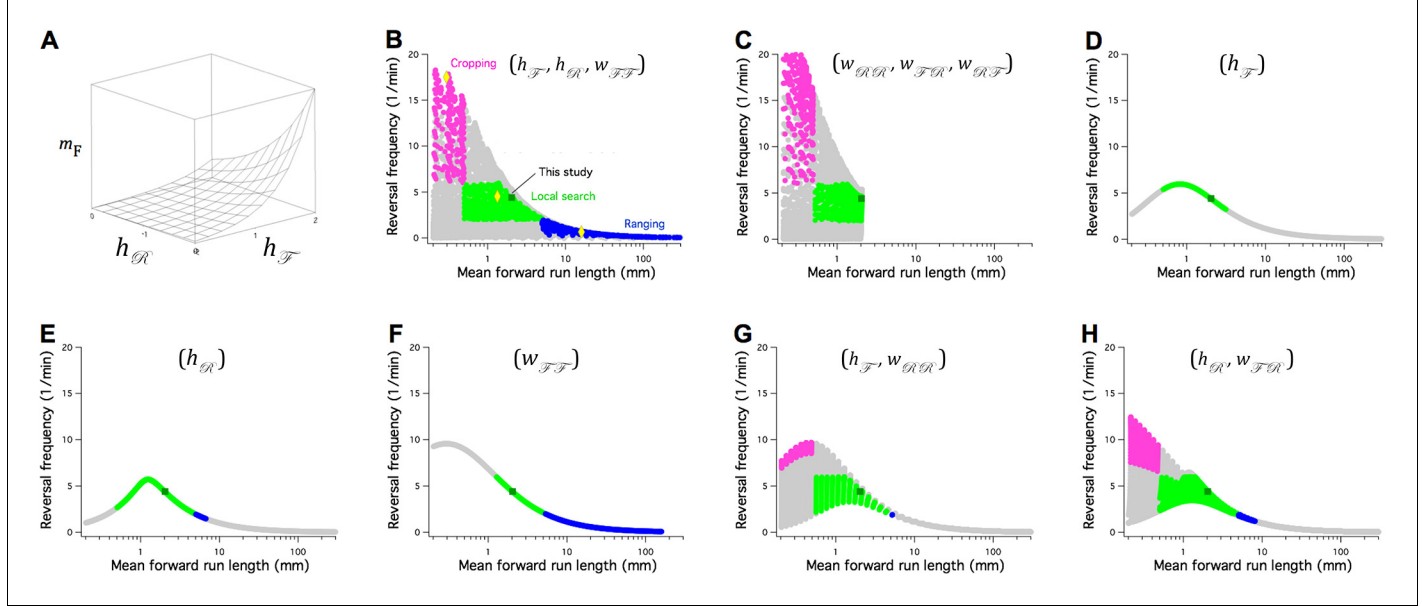

**Figure 8.** The Stochastic Switch Model accounts for the three main modes of random search in *C. elegans*. (A) Plot of mean forward run length versus the weights $h_\mathcal{F}$ and $h_\mathcal{R}$, illustrating a minimal model of search-scale regulation. (B-H) Calculated effects on search mode of the weights indicated in parentheses. The frequency of reversals ($f_{FPR}$) is plotted against $m_F$ while these three weights are scanned from -6 to 6 weight units in steps of 0.4. Each point was categorized as cropping (magenta), local search (green), ranging (blue), or indeterminate (grey) according to value of $f_{FPR}$ and $m_F$, and whether or not the associated value of $m_R$ (not shown) indicated a short or long reversal; see Materials and methods for definitions of search modes. Yellow diamonds mark the scanned points modeled in *Figure 8—figure supplement 1*. $A$ = 1 Hz; similar results were obtained for $A = A_{max}$ and $A = A_{min}$ (*Table 7*).

DOI: https://doi.org/10.7554/eLife.12572.025

The following figure supplements are available for figure 8:

**Figure supplement 1.** Simulated worm tracks illustrating cropping, local search, and ranging as defined in the model.
DOI: https://doi.org/10.7554/eLife.12572.026

**Figure supplement 2.** Extension of the Stochastic Switch Model to chemotaxis.
DOI: https://doi.org/10.7554/eLife.12572.027

**Table 7.** Regulation of search mode.

The weights in each subspace were scanned from -6 to 6 weight units in steps of 0.4 with $A = A_{\min}$ or $A = A_{\max}$. The letter x means that the indicated search mode was present for at least one point in the subspace when $A = A_{\min}$ and when $A = A_{\max}$; the letters y and z mean that the mode was present only when $A = A_{\min}$ or $A = A_{\max}$, respectively. See Materials and methods for definitions of search modes. Control-point weights as defined by the theoretical relationship between weights and search scale (*Equation 5*) are shown in bold. Only the three-weight subspaces are sufficient for producing all three search modes and full coverage of the plane defined by reversal frequency and $m_F$ plane as shown in *Figure 8*.

| Subspace | Cropping | Dwelling | Ranging | Coverage |
|---|---|---|---|---|
| $hFh_{\mathcal{F}}$ | | x | | . . . |
| $hR$ | | x | x | . . . |
| $wFF$ | | x | x | . . . |
| $wRR$ | | x | | . . . |
| $wRF$ | | x | | . . . |
| $wFR$ | | x | | . . . |
| $hF, wRR$ | y | x | x | . . . |
| $hF, wFR$ | | x | x | . . . |
| $hF, wRF$ | | x | x | . . . |
| $hR, wRR$ | | x | x | . . . |
| $hR, wFR$ | x | x | x | . . . |
| $hR, wRF$ | | x | x | . . . |
| $wFF, wRF$ | | x | x | . . . |
| $wFF, wFR$ | | x | x | . . . |
| $wFF, wRR$ | | x | x | . . . |
| $hR, wFF$ | | x | x | . . . |
| $hF, hR$ | | x | x | . . . |
| $hF, wFF$ | | x | x | . . . |
| $wRR, wFR$ | x | x | | . . . . . . . |
| $wRR, wRF$ | | y | | . . . |
| $wFR, wRF$ | | x | | . . . . . . . |
| $hF, wRF, wFR$ | x | x | x | . . . . . . . |
| $hR, wRF, wFR$ | x | x | x | . . . . . . . |
| $wFF, wRF, wFR$ | x | x | x | . . . . . . . |
| $hF, wRR, wFR$ | x | x | x | . . . . . . . |
| $hR, wRR, wFR$ | x | x | x | . . . . . . . |
| $wFF, wRR, wFR$ | x | x | x | . . . . . . . |
| $hF, wRR, wRF$ | x | x | x | . . . . . . . |
| $hR, wRR, wRF$ | z | x | x | . . . . . . . |
| $wFF, wRR, wRF$ | x | x | x | . . . . . . . |
| $hF, hR, wFR$ | x | x | x | . . . . . . . |
| $hF, hR, wRR$ | x | x | x | . . . . . . . |
| $hF, hR, wRF$ | x | x | x | . . . . . . . |
| $hF, wFF, wRR$ | x | x | x | . . . . . . . |
| $hF, wFF, wFR$ | | x | x | . . . . . . . |
| $hF, wFF, wRF$ | | x | x | . . . . . . . |
| $hR, wFF, wFR$ | x | x | x | . . . . . . . |
| $hR, wFF, wRF$ | x | x | x | . . . . . . . |
| $hR, wFF, wRR$ | y | x | x | . . . . . . . |
| $hF, hR, wFF$ | x | x | x | . . . . . . . |
| $wRR, wRF, wFR$ | x | x | | . . . . . . . |

DOI: https://doi.org/10.7554/eLife.12572.028

predict population-level synaptic connectivity. The source of the AVB inhibition could be the inhibitory connection from AVA, polysynaptic pathways from ASH to AVB, or both.

Notably, the Pull and Push-Pull motifs are equally effective in driving $p_R$ to 1.0 (*Figure 9B*). Nevertheless, computation of the expected latency to the first reversal event when a forward moving animal suddenly encounters a strong nociceptive stimulus indicates a 2.3-fold reduction in latency for

the Push-Pull motif (*Figure 9B*, parenthetical values). We conclude that the ASH mediated escape circuit in *C. elegans* may be specialized for short latency escape responses.

## Discussion

The Stochastic Switch Model is cast at a level of biological detail that is minimally sufficient to capture the stochastic dynamics of *C. elegans* locomotion in neuronal terms. Despite its simplicity, the model predicts the unexpected effects of neuronal ablations and genetic manipulations. It also predicts the sign and strengths of key synaptic connections, which were confirmed by combining optogenetics with electrophysiology. The model is immediately extensible to random search at a variety of spatial scales, biased random walks such as chemotaxis, and deterministic escape behaviors. The predictive success of the model indicates that random search in *C. elegans* can be understood in terms of a neuronal flip-flop circuit involving reciprocal inhibition between two populations of stochastic neurons. Two likely sources of stochastic state transitions are quantal synaptic transmission and ion channel gating. Both of these sources derive their randomness from thermal fluctuations at the molecular level, a phenomenon that is common to all nervous systems. The stochasticity underlying search behavior in *C. elegans* could be intrinsic to the command neurons, their presynaptic neurons (*Gordus et al., 2015*), or both.

The simplifying assumptions of the model introduce several limitations worth noting. (i) By representing the ten command neurons as only two functional units, the model ignores possible functional differences between individual neurons within each group. (ii) By design, the model predicts exponentially distributed dwell times, but *Figure 2—figure supplement 3* shows that this relationship is only approximate. (iii) The model also has no provision to explain the strong correlation between locomotory phase and entry into state Y (*Figure 4*), although this could be added by modeling feedback from the pattern generator as a time-varying component of $h_\mathcal{F}$ and $h_\mathcal{R}$. (iv) The model does not take into account temporal correlations in velocity, but instead uses only the present velocity, along with the present state, to compute transition probabilities. For example, the fact that locomotion gradually slows before the worm enters the pause state (*Figure 1G,H*) suggests that transition probabilities might be more reliably calculated by including the recent velocity history, rather than just the present velocity. (v) Finally, the model does not attempt to explain the observation that the number of command neurons that are present and the degree of command neuron activation has an effect on velocity and undulation frequency (*Figure 5A*, *Table 3*, *Table 5*). Velocity modulation could be incorporated by relaxing the assumptions that command neurons within pools are co-active and have a single non-zero level of activation.

Although the model correctly predicts several unexpected and even paradoxical observations at the behavioral and electrophysiological levels, it would be premature to conclude that the biological system functions as assumed. This caution extends to all of the model's assumptions, including the mapping relationship between pause states X and Y and their behavioral correlates. We view the pause states as theoretical constructs having an epistemological status akin to theoretical constructs in many widely accepted models, such as the gating particles that were proposed in the Hodgkin-Huxley model of the squid action potential to explain the voltage sensitivity of ion channels.

An altogether different method for analyzing locomotory states in *C. elegans* also proposed the existence of two pause states (*Stephens et al., 2008*). In that work, each pause state was associated with a particular locomotory phase. In contrast, we found that only state Y occurred in association with a particular posture (a ventral bend in the middle of the body), whereas state X occurred with essentially no postural preference. The reason for this discrepancy may be that pauses are identified in different ways in the two studies. Here pauses are identified in terms of tangential velocity. In Stephens et al. (*Stephens et al., 2008*; 2011; 2010), however, pauses are identified in the phase space defined by the amplitudes of first two principle components of the worm's instantaneous shape. For the two approaches to yield the same result, minima in the magnitudes of tangential and phase velocity would have to be coincident at all times. We believe this outcome is unlikely because the third and fourth principle shape components, which account for approximately 30% of the shape variance (*Stephens et al., 2008*), meet the necessary and sufficient conditions for generating tangential thrust: a gradient of curvature along the worm's centerline (*Gray, 1946*; *Gray, 1953*; *Gray and Lissmann, 1964*); this is one way thrust is believed to be generated during omega turns (*Stephens et al., 2008*). Thus, the worm can be moving with respect to the substrate even when

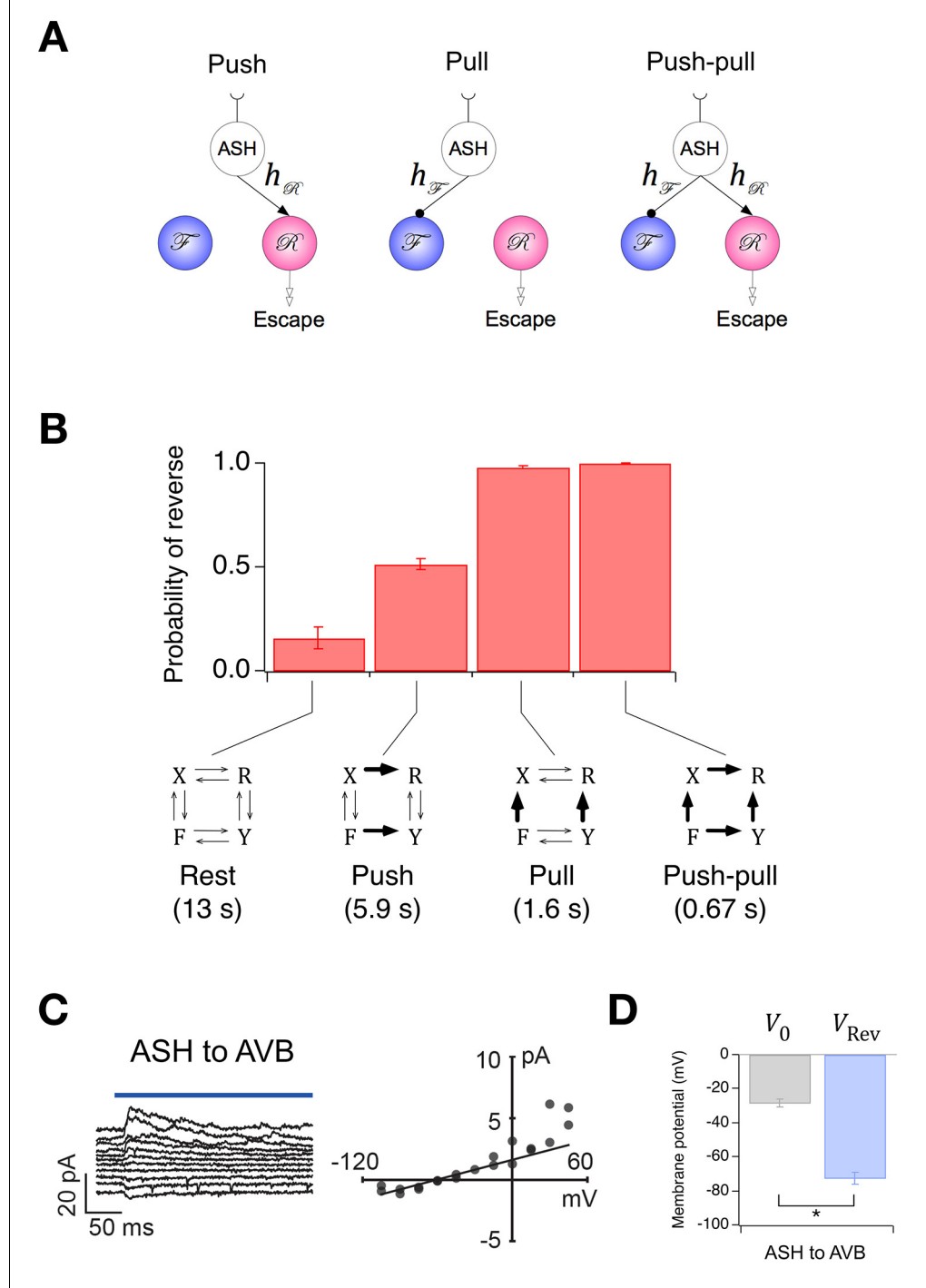

**Figure 9.** Extension of the Stochastic Switch Model to deterministic behaviors. (**A**) Three functional circuit motifs for deterministic escape behavior initiated by the nociceptive neuron ASH. (**B**) Predicted steady-state probability of reversal behavior in the resting state and the activated state of the three motifs shown in **A**. Plotted values are means across the five wild type cohorts shown in *Figure 1F*. Error bars are ± sem. Numbers in parenthesis are predicted mean first latency to a reversal response. (**C**) *Left*, synaptic current in AVB when ASH was photoactivated (blue line). *Right*, mean synaptic current during the first 100 ms of the stimulus plotted against holding potential in AVB. The line is fit to the data at negative holding potentials. (**D**) Mean zero-current holding potential and mean reversal potential of synaptic currents (± sem) in AVB (paired *t*-test: *p* = 0.013, *n* = 4).
DOI: https://doi.org/10.7554/eLife.12572.029

phase velocity is zero. Overall, we speculate that pauses in phase velocity are a subset of pauses in tangential velocity. The extent to which this is true could be determined by performing spot tracking and shape analysis on the same individual worms.

It will be interesting to test several additional predictions of Stochastic Switch Model:

1. The sign of the input weights, $h_{\mathcal{F}}$ and $h_{\mathcal{R}}$ predicts tonic excitation of the network. This could be the result of constitutive excitatory synaptic inputs, or depolarizing leakage currents in individual command neurons as has been proposed (*Gao et al., 2015*).

2. The sign of the self-connections $W_{\mathcal{FF}}$ and $W_{\mathcal{RR}}$ predicts one or more mechanisms of self-excitation within command neuron pools. These might include excitatory connections between command neurons, or intrinsic membrane currents capable of producing plateau potentials (*Mellem et al., 2008*).

3. The fate diagram (*Figure 3*) predicts that forward commands neurons generally lead the changes in direction during spontaneous locomotion. For example, transitions from F to R almost always begin with the $\mathcal{F}$ unit turning off, whereas transitions from R to F almost always begin with the $\mathcal{F}$ unit turning on. This prediction could be tested by calcium imaging in command neurons in freely moving animals (*Nguyen et al., 2015*; *Venkatachalam et al., 2016*).

4. Finally, the prediction that forward command neurons lead the changes in direction, coupled with the observation that transitions from R to Y occur at a particular phase, predicts that the forward command neurons are the predominant site at which phasic feedback from the locomotion pattern generator influences the network. Direct observation of neuronal activity in freely moving animals would be the ideal experiment to confirm the existence of the two pause states proposed in the Stochastic Switch Model (*Nguyen et al., 2015*; *Venkatachalam et al., 2016*). In particular, it will be necessary to show that whenever all command neurons are OFF, or all are ON, tangential velocity goes to zero. These experiments will be challenging because they must be done by imaging neuronal activity in freely moving animals at a temporal resolution that exceeds what can be obtained with the current generation of calcium probes. In fact, it may be necessary to use voltage probes rather than calcium indicators because even a very fast calcium probe will be limited by the dynamics of calcium accumulation, which is slow on the time scale of the pause dwell times predicted by the model. Another potential complication is that velocity may not change instantaneously with changes in the state of the command network, but with a delay imposed by time constants in the motor system. A less direct approach, although one with much higher temporal resolution, would be to make whole cell current clamp recordings from command neurons or motor neurons in restrained animals, which cycle through global brain states analogous to forward and reverse locomotion (*Kato et al., 2015*) even though they cannot move. Instances in which both motor systems are OFF or ON would provide evidence for states X and Y, respectively.

Like the Stochastic Switch Model, a previous model of the command neuron circuit by Rakowski et al. (*Rakowski et al., 2013*) predicts reciprocal inhibition between command neurons. Although the two models analyze locomotion behavior in terms of the same three behavioral states – forward, reverse, and pause – the models have essentially no points of mathematical contact. In the Rakowski model, neurons are deterministic electrical compartments and only the long-term average state probabilities of the network are computed. In the Stochastic Switch Model, by contrast, neurons are inherently stochastic and instantaneous state is computed. These disparities are significant because only the Stochastic Switch Model can predict temporal phenomena including such fundamental quantities as transition rates and mean dwell times. The fact that the both models predict reciprocal inhibition may reflect that fact that the behavioral signal of reciprocal inhibition is strong enough to transcend large differences between models.

Mammalian sleep, like *C. elegans* locomotion, is composed of numerous abrupt alternations between opposing behavioral states. Sleep is punctuated frequently by brief periods of wakefulness, and dwell time distributions in sleep and wake states indicate that switching between them is a stochastic process (*Lo et al., 2004*). Sleep and wakefulness are controlled by mutually inhibitory brainstem nuclei, implying a reciprocal inhibition motif. In a significant parallel to the effects of command neuron ablations on dwell times in *C. elegans* locomotion (*Figure 5B*), lesions of sleep-related nuclei simultaneously reduce the dwell times in both sleep and wake states, as do lesions of wakefulness nuclei (*Saper et al., 2010*). Thus the relationship between synaptic uncoupling of the circuit and changes in dwell times may be a general principle of reciprocal inhibition in stochastic neuronal

networks. Further study of invertebrate models of this circuit motif would be a productive means of identifying the genetic and physiological underpinnings of such circuits.

The debut of the essentially complete wiring diagram of the *C. elegans* nervous system raised the prospect of the first account of the entire behavioral repertoire of an organism at single-neuron resolution (*White et al., 1986*; *Varshney et al., 2011*). To date, the repertoire of behaviors commonly recognized in *C. elegans* can be divided into three main functional categories, subsuming 23 different elementary actions (*Faumont et al., 2012*). Because the command neurons considered here are required for almost half of this repertoire, the Stochastic Switch Model is a significant step toward a comprehensive understanding of the neuronal basis of behavior in this animal, bringing us closer to the goal of computing the behavior of an entire organism. Though abstract by design in its representation of individual neurons and synapses, the model accommodates not only random search at multiple spatial scales (*Figure 8*), but also biased random walks (*Figure 8—figure supplement 1*) and deterministic escape behaviors (*Figure 9*). We propose, therefore, that the Stochastic Switch Model could serve as a multipurpose module for computing *C. elegans* behavior. Combining this mathematically tractable module with others representing sensory inputs, modulatory states, and the presumptive pattern generators for forward and reverse locomotion, could lead to essentially complete models of the *C. elegans* nervous system that are at once predictive and intuitively comprehensible (*Abbott, 2008*).

## Materials and methods

### Strains

All strains were cultivated at 22.5°C on low-density NGM (nematode growth medium) agar plates seeded with the *E. coli* bacteria (OP50) as described by Brenner (*Brenner, 1974*). Transgenic lines were made using standard protocols (*Mello and Fire, 1995*).

| Experiment | Figure | Strains and genotypes |
|---|---|---|
| Wild type | 1-8 | N2 |
| AVA → AVB synaptic current | 5B | XL238 *ntIs[Prig-3::ChR2, Punc-122::dsRed]; ntIs35[Psra-11::tdTomato]; lite-1(ce314)* |
| AVB → AVA synaptic current | 5C | XL237 *kyEx3801[Psra-11::ChR2::GFP, Punc-122::dsRed]; ntIs29[Pnmr-1::tdTomato]; lite-1(ce314)* |
| AVA ablation | 4 | N2 |
| AVD ablation | 4 | XL59 *akIs [lin-15(+); Pnmr-1::GFP]* |
| AVE ablation | 4 | XL59 *akIs [lin-15(+); Pnmr-1::GFP]* |
| AVB ablation | 4 | N2 |
| PVC ablation | 4 | XL59 *akIs [lin-15(+); Pnmr-1::GFP]* |
| HYP A[†] | 6 | DA572 *eat-4(ad572)* |
| HYP B[†] | 6 | MT6308 *eat-4(ky5)* |
| HYP C[†] | 6 | KP4 *glr-1(n2461)* |
| DEP A[†] | 6 | VM1136 *lin-15(n765); akIs9 [lin-15(+), Pglr-1::GLR-1(A/T)]* |
| DEP B[†] | 6 | VM188 *lin-15(n765); akEx52[lin-15(+), Pnmr-1::GLR-1(A/T) ]* |
| ASH → AVB synaptic current | 8 | XL194 *ntIs27 [ Psra-6::ChR2::YFP, Punc-122::dsRed]; ntIs35 [Psra-11::tdTomato]; lite-1(ce314)* |

[†]Internal reference HYP A = *HYP16*, HYP B = *HYP 56*, HYP C = *HYP20*, DEP A = *DEP14*, DEP B = *DEP19*.

### Physiological solutions

External saline for electrophysiology (mM): 5 KCl, 10 HEPES, 8 CaCl2, 143 NaCl, 30 glucose, pH 7.2 (NaOH); internal saline for electrophysiology (mM): 125K-gluconate, 1 CaCl2, 18 KCl, 4 NaCl, 1 MgCl2, 10 HEPES, 10 EGTA, pH 7.2 (KOH). Medium for behavioral assays (mM): $NH_4Cl$ 2, $CaCl_2$ 1,

$MgSO_4$ 1, and $KPO_4$ 25, pH 6.5; M9 Buffer (grams): 3 $KH_2PO_4$, 6 $Na_2HPO_4$, 5 NaCl, 1 ml 1 M $MgSO_4$, $H_2O$ to 1 liter.

## Behavior and tracking system

Prior to each assay, an individual adult hermaphrodite was picked to a bacteria-free agar transfer plate by means of a platinum-wire pick. The worm was then washed in M9 to remove excess bacteria, then transferred in a pipette filled with assay medium to a 10 cm petri plate containing 1.7% agarose in assay medium. A black dot approximately 40 microns in diameter was applied to the center of the body as shown in *Figure 1A* (see Spotting procedure). The worm was allowed to recover from transfer and handling for 2 min., then recorded for 10 min. The assay plate rested on a motorized microscope stage (Applied Scientific Instrumentation MS-2000, Eugene, OR USA) fitted with position encoders (Gurely Precision Instruments LE-1800, Troy, NY USA) having a resolution of 0.5 µm. Behavior was recorded using an analog video camera (CCD Sony XC-ST70, 29.97 frames per second) fitted with a 12× zoom lens (Navitar 50486D, Rochester, NY USA). For tracking purposes, video was analyzed in real time by custom software to calculate the eccentricity of the ink spot relative to the center of the field of view, and to compute the stage movements required to re-center the spot. Motion blur was minimized by making stage speed during corrective movements an increasing exponential function of target eccentricity such that small corrections were made more slowly than large corrections. Position encoders were read in synchrony with the video stream and this information was stored for off-line analysis. The overall trajectory of the worm was computed by combining the location of the spot in the field of view with stage position in each video frame. The direction of movement (forward or reverse) at the start of each recording was keyed by the observer and subsequent assignments were made automatically by computer. Each recording was spot-checked for correct assignments at four or more points during the recording. In experiments involving neuronal ablations or genetic mutations, recordings of sham operated controls or wild type worms, respectively, were interleaved with worms in each treatment group.

## Spotting procedure

The animal was immobilized by a stream of humidified $CO_2$ emitted by a 1.5 mm diameter glass pipette positioned near the worm. The spotting ink was comprised of petroleum jelly (1 ml), mineral oil (1 ml), and black iron oxide (3 g). Ink was applied by means of 1.5 mm diameter glass rod that had been pulled to a fine point, fire polished to produce a bulbous tip, and dipped in the ink. The rod was positioned by means of a micromanipulator. To control for the effects of the spotting procedure, we compared the speed of locomotion of worms that had been immobilized, or immobilized and spotted, to untreated worms. There were no significant differences between these three groups.

## Electrophysiology

Worms were glued to an agarose coated coverslip using cyanoacrylate adhesive as previously described (*Lindsay et al., 2011*). The coverslip formed the bottom of the recording chamber, which was filled with external saline. The cell body of the neuron to be recorded was exposed by making a small slit in the cuticle using a finely drawn glass rod. Recording pipettes had resistances of 10–20 MΩ when filled with internal saline. Voltage- and current-clamp recordings were made with a modified Axopatch 200A amplifier (*Lockery and Goodman, 1998*). In reversal potential measurements, recordings of photostimulation-evoked synaptic currents were filtered at 2 kHz and sampled at 10 kHz. Postsynaptic neurons (AVA, AVB) were identified using a combination of fluorescent markers and distinctive voltage clamp currents as described (*Lindsay et al., 2011*). Presynaptic neurons (AVA, AVB, and ASH) were activated by expression of ChannelRhodopsin-2 expressed under the control of neuron-specific promoters as described (see "Strains"). Worms were photostimulated in electrophysiological experiments using the blue channel (470 nm) of a dual-wavelength LED module (Rapp OptoElectronic, Wedel, Germany) that was focused by a 63×, 1.4 NA oil immersion objective lens (Zeiss, part number 440762–9904). Irradiance (12.5 mW/mm$^2$) was determined by measuring the power emitted from the objective using an optical power meter placed above the front lens of the objective and dividing by the area of the field of illumination at the focal plane of the preparation.

## Ablations

Neurons were ablated using a laser as described previously (*Bargmann and Avery, 1995*). L1 larvae were mounted on 2.5% agarose pads containing 5–7 mM of the immobilizing agent NaN$_3$. AVA and AVB neurons were ablated in N2 animals and identified by position. AVD, AVE and PVC were ablated in animals expressing *nmr-1*::GFP and identified by a combination of position and GFP expression. To limit potential behavioral differences in the two strains, we outcrossed (4×) the *nmr-1*::GFP strain to the N2 strain used for AVA and AVB ablations. All animals were remounted 1–3 hr after surgery to confirm the ablation; those with collateral damage were discarded. Sham-operated animals were treated in the same manner except that the laser was not fired.

## Statistical tests

Statistical significance for the results shown in *Figures 4B* and *6C*, and in *Tables 4* and *6* were obtained using the likelihood ratio test (see *Table 1* and *4* legends). Otherwise, two-tailed *t*-tests or 2-tailed Mann-Whitney U tests were used.

## Descriptive statistics

The worm's position in video frame *k* is represented as the row vector:

$$\boldsymbol{R}(t_k) = [x(t_k),\, y(t_k)] \qquad (k = 1,\, 2,\, \ldots,\, N) \tag{6}$$

where $X(t_k)$ and $Y(t_k)$ are the coordinates of centroid of the tracking spot in the frame of reference of the agar plate, $t_k = k\Delta t$, $\Delta t =$ 33 ms, and $N \cong$ 18000 is the number of video frames analyzed in a continuous recording of one worm. We made the following definitions:

### Row vectors
Velocity:

$$\boldsymbol{V}(t_k) = \frac{\boldsymbol{R}(t_{k+1}) - \boldsymbol{R}(t_k)}{\Delta t} \tag{7}$$

Heading:

$$\boldsymbol{H}(t_k) = \frac{\boldsymbol{V}(t_k)}{s(t_k)} \tag{8}$$

### Scalar quantities
Speed:

$$s(t_k) = \|\boldsymbol{V}(t_k)\| = \sqrt{\boldsymbol{V}(t_k) \cdot \boldsymbol{V}^{\mathbf{t}}(t_k)} \qquad \left(\boldsymbol{V}^{\mathbf{t}} \equiv transpose\ of\ \boldsymbol{V}\right) \tag{9}$$

Mean speed:

$$\dot{s} = \frac{1}{N-1}\sum_{k=1}^{N-1} s(t_k) \tag{10}$$

Instantaneous turn rate:

$$\frac{\Delta\varphi}{\Delta t}(t_k) = \frac{cos^{-1}\left(\boldsymbol{H}(t_{k-1}) \cdot \boldsymbol{H}^{\mathbf{t}}(t_k)\right)}{\Delta t} \qquad (0 \leq \Delta\varphi \leq \pi) \tag{11}$$

Mean heading change:

$$\frac{\left(t_j\right) = \frac{1}{N-j-1}\sum_{k=1}^{N-j-1} cos^{-1}\left(\boldsymbol{H}\left(t_k + t_j\right) \cdot \boldsymbol{H}^{\mathbf{t}}(t_k)\right) \qquad (0 \leq \Delta\varphi \leq \pi)}{\Delta\varphi} \tag{12}$$

Speed autocovariance:

$$A_s(t_j) = \frac{1}{N-j-1} \sum_{K=1}^{N-j-1} \left( s(t_k + t_j) - \bar{s} \right) \cdot \left( s(t_k) - \bar{s} \right) \tag{13}$$

Velocity autocorrelation:

$$A_V(t_j) = \frac{1}{N-j-1} \sum_{K=1}^{N-j-1} \mathbf{V}(t_k + t_j) \cdot \mathbf{V}^{\mathrm{t}}(t_k) \tag{14}$$

Heading autocorrelation:

$$A_H(t_j) = \frac{1}{N-j-1} \sum_{K=1}^{N-j-1} \mathbf{H}(t_k + t_j) \cdot \mathbf{H}^{\mathrm{t}}(t_k) \tag{15}$$

Mean squared displacement:

$$\overline{r^2(t_j)} = \frac{1}{N-j-1} \sum_{k=1}^{N-j-1} ||\mathbf{R}(t_k + t_j) - \mathbf{R}(t_k)||^2 \tag{16}$$

## Maximum likelihood estimation of state transition rates in a hidden Markov model

To analyze locomotory states we converted the velocity vector, $V(t)$, into a signed scalar quantity $v(t)$ that represents the component of velocity in the direction of the worm's track, with positive values indicating forward movement. We first smoothed $x(t)$ and $y(t)$ using an 11 frame window, assigned a direction to the smoothed track with respect to the head/tail orientation of the worm, and projected $V(t)$ onto the smoothed track to obtain $v(t)$. For each cohort of worms we collected all $v(t)$ values into a single velocity distribution $g(v)$. The central peak of $g(v)$ was fit by a Cauchy distribution with median 0 and half-width $b$ = 18 μm/s (*Figure 2—figure supplement 2*), which we used to approximate the pause velocity distribution for states X and Y for all worms:

$$g_{\mathrm{X}}(v) = g_{\mathrm{Y}}(v) = g_{\mathrm{P}}(v) = \frac{b}{\pi(b^2 + v^2)} \tag{17}$$

We used a Cauchy distribution because it has long tails that describe the pause velocity distribution better than a Gaussian distribution (i.e., the worm does not stop instantaneously when it switches from forward or reverse locomotion into one of the pause states). We estimated the forward and reverse velocity distributions $g_F(v)$ and $g_R(v)$ by scaling $g_P(v)$ to fit the peak at $v = 0$, subtracting it from the overall distribution and splitting the remaining distribution into $g_F(v)$ for $v > 0$ and $g_R(v)$ for $v < 0$. Velocity distributions were scaled to be probability densities (area =1) and collected into a row vector:

$$\mathbf{G}(v) = [g_{\mathrm{F}}(v), \ g_{\mathrm{R}}(v), \ g_{\mathrm{X}}(v), \ g_{\mathrm{Y}}(v)] \tag{18}$$

where $g_i(v)$ is the estimated probability density that worms move at velocity $v$ when in state $i$.

The goal of the maximum likelihood fitting procedure is to find the set of state transition rates $\{a_{XF}, a_{FX}, a_{XR}, a_{RX}, a_{FY}, a_{YF}, a_{RY}, a_{YR}\}$ that maximize the probability of the observed velocity time series $v(t)$ given the velocity distribution $G(v)$. All transition rates were constrained to be $\geq 0$, and usually were additionally constrained to correspond to valid synaptic weights as described below. The likelihood is most conveniently calculated using matrix notation as follows; see *Colquhoun and Hawkes, 1995* for a more complete explanation of these computations. Let:

$$\mathbf{Q} = \begin{bmatrix} -(a_{\mathrm{FX}} + a_{\mathrm{FY}}) & 0 & a_{\mathrm{FX}} & a_{\mathrm{FY}} \\ 0 & -(a_{\mathrm{RX}} + a_{\mathrm{RY}}) & a_{\mathrm{RX}} & a_{\mathrm{RY}} \\ a_{\mathrm{XF}} & a_{\mathrm{XR}} & -(a_{XF} + a_{XR}) & 0 \\ a_{\mathrm{YF}} & a_{\mathrm{YR}} & 0 & -(a_{\mathrm{YF}} + a_{\mathrm{YR}}) \end{bmatrix} \tag{19}$$

Element $q_{ij}$ ($i \neq j$) of matrix $Q$ is the transition rate from state $i$ to state $j$ (i.e., the instantaneous probability per unit time that the system in state $i$ will make a transition to state $j$, and element $q_{ii}$ is the negative of the total transition rate out of state $i$, which is related to the mean dwell time in state $i$ by:

$$d_i = -1/q_{ii} \tag{20}$$

Matrix $Q$ is composed of instantaneous transition rates, which can be converted into the matrix of transition probabilities during a brief time interval of duration $\varepsilon$ by multiplying $Q$ by $\varepsilon$ and adding 1 to each diagonal element (i.e., by calculating $\varepsilon \cdot Q + I$, where $I$ is the 4×4 identity matrix). If $\varepsilon$ is sufficiently small that multiple state transitions can be ignored, then element $ij$ of matrix $\varepsilon \cdot Q + I$ is the probability that the system is in state $j$ at the end of a time interval of duration $\varepsilon$ given that it was in state $i$ at the beginning of the interval. For longer time intervals during which multiple state transitions may occur, transition probabilities can be calculated by repeatedly multiplying matrix $\varepsilon \cdot Q + I$ by itself. Thus, if

$$M = (\varepsilon \cdot Q + I)^K \tag{21}$$

then $M$ is the matrix of transition probabilities during a time interval of duration $K\varepsilon$. If $K$ and $\varepsilon$ are chosen such that $\Delta t = K\varepsilon$, then element $ij$ of matrix $M$ is the transition probability from state $i$ to state $j$ during one video frame of duration $\Delta t$. We chose $K = 230$ and let $\varepsilon = \Delta t / K = 30.7$ picoseconds, a time interval during which multiple state transitions can safely be ignored. Since $K$ was chosen to be a power of 2, $M$ could be rapidly and accurately calculated by 30 serial multiplications using 64-bit floating point arithmetic.

Let $P(t)$ be the row vector of history-dependent state probabilities:

$$P(t) = [\, p_F(t),\ p_R(t),\ p_X(t),\ p_Y(t)\,] \tag{22}$$

where $pi(t)$ is the probability of being in state $i$ at time $t$ given $v(u)$ for all $u$ up to and including the present time ($u \leq t$). The matrix product $P(t) \cdot M$ is the state probability vector at time $t + \Delta t$ prior to accounting for the observed velocity at time $t + \Delta t$. To account for $v(t + \Delta t)$ we used the information contained in $G(v(t + \Delta t))$ and applied Bayes theorem:

$$P(t + \Delta t) = l \cdot P(t) \cdot M \cdot diag\Big[ G\Big( v(t + \Delta t) \Big) \Big] \tag{23}$$

where $diag G(v(t + \Delta t))$ is the 4×4 matrix with the elements of $G(v(t + \Delta t))$ along the diagonal, and $l$ is the scalar multiplicative factor required for the sum of the four elements of $P(t + \Delta t)$ to equal 1 (i. e., $P(t + \Delta t)$ is a vector of probabilities). Initially ($t = 0$) we set $P(0)$ equal to the steady-state probability vector $P_\infty$, which is given by:

$$P_\infty \cdot Q = 0 \qquad \Rightarrow \qquad P_\infty = U_4 \cdot (Q_a \cdot Q_a^t)^{-1} \tag{24}$$

where $U4$ is the $1 \times 4$ row vector of ones and $Qa$ is the $4 \times 5$ matrix constructed by appending a column of ones to $Q$. To break the symmetry between the behaviorally indistinguishable states X and Y, we identified X as the state with higher steady-state probability.

We then calculated the log-likelihood, summed over all worms in the cohort:

$$ln(L) = \sum_{t,w} ln\Big( P_w(t) \cdot G^t\Big( v_w(t) \Big) \Big) \tag{25}$$

where $V_W(t)$ is the velocity and $P_W(t)$ is the history-dependent state probability vector of worm $w$ at time $t$.

We used a random optimization algorithm to find the set of transition rates that maximized $lnL$. Initial guesses for 6 of the 8 rates were chosen independently from log uniform distribution between 0.01 Hz and 10 Hz. The remaining 2 rates were calculated to satisfy the constraints needed to generate valid synaptic weights (see below). At each iteration, each of the 6 independently chosen rates was altered by adding a random number chosen from a Cauchy distribution with median 0 and width $b_{random}$ (initially $b_{random} = 0.01$ Hz), and the remaining 2 rates were recalculated. To avoid getting trapped in local likelihood maxima, the new rates were rejected and another set was calculated if any of the new rates were <0.01 Hz. If the new rates generated an increased likelihood, the new rates were accepted and $b_{random}$ was increased by 3%. Otherwise the old rates were retained and $brandom$ was decreased by 0.5%. The procedure was iterated until $b_{random} < 0.001$ Hz. The random optimization procedure was replicated 10 times for each cohort using different randomly chosen initial guesses. In 71% of the replicates the procedure converged on a set of transition rates in which none of the transition rates differed from the best set by more than 5%. The best set of transition

rates was then refined by applying the optimization procedure using a success criterion of $b_{random} < 10^{-5}$ and constraining transition rates to be $\geq 10^{-4}$ Hz.

The likelihood calculations described above use only past and present velocity observations to calculate $P(t)$, but once the optimal transition rates were determined, the Forward-Backward algorithm (**Rabiner and Juang, 1986**) can be used to yield a better estimate of the state probabilities based on past, present and future velocity observations, and the Viterbi Algorithm can be used to find the sequence of states with the highest probability of producing the observed velocities (**Figure 2E**).

## Stochastic model neurons

We expressed the effect of synaptic inputs to command units $F$ and $R$ by equations of the form:

$$a_{\mathrm{ON}} = A \cdot e^S \tag{26}$$

$$a_{\mathrm{OFF}} = A \cdot e^{-S} \tag{27}$$

where $a_{OFF}$ is the transition rate from ON to OFF, $a_{ON}$ is the transition rate from OFF to ON, and $S$ is the total synaptic input to the unit. We do not attach any mechanistic significance to these equations, but note that they are analogous to the Arrhenius Equation (**Stiller, 1989**) an approximation commonly used to describe the rates of chemical reactions in terms of an activation energy, $E$:

$$a = A \cdot e^{-\frac{E}{k_B T}} \tag{28}$$

where $a$ is the reaction rate constant, $A$ is an empirically determined constant, $K_B$ is the Boltzmann constant, and $T$ is the absolute temperature. Under this interpretation, $S$ is analogous to activation energy expressed in units of $K_B T$. Thus, $\mathcal{F}$ and $\mathcal{R}$ are assumed to be symmetrical bi-stable units that change state at rate $A$ when $S = 0$. Deviations from this baseline condition are modelled as external synaptic inputs $h_{\mathcal{F}}$ and $h_{\mathcal{R}}$.

We represented the total synaptic input onto units $\mathcal{F}$ and $R$, respectively, by:

$$S_{\mathcal{F}} = h_{\mathcal{F}} + b_{\mathcal{F}}(t)w_{\mathcal{FF}} + b_{\mathcal{R}}(t)w_{\mathcal{RF}} \tag{29}$$

$$S_{\mathcal{R}} = h_{\mathcal{R}} + b_{\mathcal{R}}(t)w_{\mathcal{RR}} + b_{\mathcal{F}}(t)w_{\mathcal{FR}} \tag{30}$$

where $b_{\mathcal{F}}(t)$ and $b_{\mathcal{R}}(t)$ are the states of $\mathcal{F}$ and $\mathcal{R}$ (1 = ON, 0 = OFF), $W_{\mathcal{RF}}$ and $W_{\mathcal{FR}}$ are the synaptic weights from $\mathcal{R}$ onto $\mathcal{F}$ and from $\mathcal{F}$ onto $\mathcal{R}$, respectively, and $W_{\mathcal{FF}}$ and $W_{\mathcal{RR}}$ represent synaptic interactions among command neurons of the same class, plus any intrinsic membrane properties that may promote bistability. Applying these definitions to the rate constants in **Figure 2C** gives:

$$a_{\mathrm{XF}} = A \exp(h_{\mathcal{F}}) \qquad\qquad a_{\mathrm{XR}} = A \exp(h_{\mathcal{R}}) \tag{31}$$

$$a_{\mathrm{FX}} = A \exp(-h_{\mathcal{F}} - w_{\mathcal{FF}}) \qquad\qquad a_{\mathrm{RX}} = A \exp(-h_{\mathcal{R}} - w_{\mathcal{RR}}) \tag{32}$$

$$a_{\mathrm{RY}} = A \exp(h_{\mathcal{F}} + w_{\mathcal{RF}}) \qquad\qquad a_{\mathrm{FY}} = A \exp(h_{\mathcal{R}} + w_{\mathcal{FR}}) \tag{33}$$

$$a_{\mathrm{YR}} = A \exp(-h_{\mathcal{F}} - w_{\mathcal{FF}} - w_{\mathcal{RF}}) \qquad a_{\mathrm{YF}} = A \exp(-h_{\mathcal{R}} - w_{\mathcal{RR}} - w_{\mathcal{FR}}) \tag{34}$$

In these experiments the sensory environment was kept constant (e.g., no chemical or temperature gradients). Therefore $h_{\mathcal{F}}$ and $h_{\mathcal{R}}$ were assumed to be constant. For simulations of chemotaxis $h_{\mathcal{F}}$ and $h_{\mathcal{R}}$ varied with position in the chemical gradient.

**Equations 31–34** express the 8 transition rates in terms of 6 parameters and yield the following two constraints on the transition rates:

$$a_{\mathrm{FX}}\, a_{\mathrm{XF}} = a_{\mathrm{RY}}\, a_{\mathrm{YR}} \qquad\qquad a_{\mathrm{FY}}\, a_{\mathrm{YF}} = a_{\mathrm{RX}}\, a_{\mathrm{XR}} \tag{35}$$

The inverse relations between transition rates and synaptic parameters are:

$$h_{\mathcal{F}} = ln(a_{\mathrm{XF}}) - ln(A) \qquad\qquad h_{\mathcal{R}} = ln(a_{\mathrm{XR}}) - ln(A) \tag{36}$$

$$w_{\mathcal{RF}} = ln(a_{\mathrm{RY}}/a_{\mathrm{XF}}) \qquad\qquad w_{\mathcal{FR}} = ln(a_{\mathrm{FY}}/a_{\mathrm{XR}}) \tag{37}$$

$$w_{\mathcal{FF}} = -ln(a_{\mathrm{XF}}\, a_{\mathrm{FX}}) + 2 \cdot ln(A) \qquad\qquad w_{\mathcal{RR}} = -ln(a_{\mathrm{XR}}\, a_{\mathrm{RX}}) + 2 \cdot ln(A) \tag{38}$$

## Derivation of the mean distance traveled during a forward run

The time series of the worm's locomotory states can be divided into forward runs, during which the worm is in either the F or P state, and reverse runs, during which the worm is in either the R or P state. Forward runs always begin with an RPF transition and end with the next FPR transition, which marks the beginning of a reverse run. Thus forward runs and reverse runs occur in strict alternation, such that the number of forward runs equals the number of reverse runs.

Let $m_F$ denote the mean distance traveled during a single forward run, assuming that forward runs are straight. The value of $m_F$ is most easily calculated by dividing time into non-overlapping epochs, each of which begins with an RPF transition and ends immediately before the next RPF transition. Each epoch thus contains exactly one forward run, which includes all visits to state F during the epoch. Therefore, $m_F$ is also equal to the mean distance travelled while in the forward state during a single epoch:

$$m_{\mathrm{F}} = \frac{\overline{v_{\mathrm{F}}} \, p_{\mathrm{F}}}{f_{\mathrm{RPF}}} \tag{39}$$

where $\overline{V}_F$ is the mean velocity in the forward state and $f_{RPF}$ is the frequency of RPF transitions. Since FPR and RPF transitions occur in strict alternation they must occur in equal numbers: $f_{RPF} = f_{FPR}$. Thus, eq. 39 can also be written with $f_{FPR}$ in the denominator, which is more useful for the calculation that follows, although the form shown above is more directly interpreted in terms of the frequency of random reorientations, which occur at the RPF transitions. It is straightforward to calculate $f_{FPR}$ given $p_F$, $a_{FX}$, $a_{FY}$, and the probabilities that the transitions out of states X and Y will be into state R:

$$prob(\mathrm{X} \longrightarrow \mathrm{R}) = a_{\mathrm{XR}}/(a_{\mathrm{XF}} + a_{\mathrm{XR}}) \tag{40}$$

$$prob(Y \longrightarrow R) = a_{\mathrm{YR}}/(a_{\mathrm{YF}} + a_{\mathrm{YR}}) \tag{41}$$

$$f_{\mathrm{FPR}} = p_{\mathrm{F}}\left( a_{\mathrm{FX}} \frac{a_{\mathrm{XR}}}{a_{\mathrm{XF}} + a_{\mathrm{XR}}} + a_{\mathrm{FY}} \frac{a_{\mathrm{YR}}}{a_{\mathrm{YF}} + a_{\mathrm{YR}}} \right) \tag{42}$$

Combining eqns. 39 and 42 yields:

$$m_{\mathrm{F}} = \overline{v_{\mathrm{F}}} \left( \frac{(a_{\mathrm{XF}} + a_{\mathrm{XR}})(a_{\mathrm{YF}} + a_{\mathrm{YR}})}{a_{\mathrm{FX}} a_{\mathrm{XR}}(a_{\mathrm{YF}} + a_{\mathrm{YR}}) + a_{\mathrm{FY}} a_{\mathrm{YR}}(a_{\mathrm{XF}} + a_{\mathrm{XR}})} \right) \tag{43}$$

An approximation to $m_F$ in terms of synaptic weights is obtained by noting that transitions from F to Y were extremely rare ($a_{FY}$ = 0.007 s$^{-1}$; **Table 1**). Setting $a_{FY} \cong 0$ yields:

$$m_{\mathrm{F}} \cong \left( \frac{a_{\mathrm{XF}} + a_{\mathrm{XR}}}{a_{\mathrm{FX}} a_{\mathrm{XR}}} \right) = \frac{v_F}{A} \left( \frac{\exp(h_{\mathscr{F}}) + \exp(h_{\mathscr{R}})}{\exp(h_{\mathscr{R}} - h_{\mathscr{F}} - w_{\mathscr{F}\mathscr{F}})} \right) \frac{v_F}{v_F} \tag{44}$$

## Simulations of worm behavior

In **Figure 8—figure supplement 1** and **2**, the worm was represented as a point that moved forward or backward at speeds of 0.2 and 0.3 mm/sec, respectively, and was stationary during the pause state. Rate constants were calculated according to **equations 31–34** based on the weights that pertain under random search or chemotaxis, using either $A_{\min} =$ or $A_{\max} =$. Weights were used to compute the state transition matrix $M$. At each time step ($\Delta t$ = 33 ms), the next state was selected randomly according to the state probabilities given by $M$. When an RPF transition occurred, a new direction of movement (heading) was selected from a uniform distribution. The random component of the heading was modeled as Gaussian noise having a standard deviation of 0.001 degrees. In the case of chemotaxis simulations, the values of $h_{\mathscr{F}}$ and $h_{\mathscr{R}}$ were updated at every time step according to the direction in which the worm was heading, leading to an updated set of weights and a new $M$ matrix.

## Definition of modes of random search in *C. elegans*

To date, these behaviors have been defined mainly in operational terms. Following the terminology of *Jander (1975)*: (i) cropping is the locomotory behavior exhibited by well-fed worms on plates with densely populated patches of bacteria; (ii) local search (also "area restricted search"

(*Hills et al., 2004*) or "pivoting" (*Wakabayashi et al., 2004*) is exhibited by well-fed worms within about 10 min after being transferred to a foodless plate; and (iii) ranging ("dispersal" (*Gray et al., 2005*) or "traveling" (*Wakabayashi et al., 2004*) is exhibited by well-fed worms tens of minutes after being transferred to a foodless plate. Each mode can be associated with approximate ranges of three parameters: mean forward run length ($m_F$), mean frequency of reversals ($f_{FPR}$), and mean reverse run length ($m_R$). Local search serves as a useful reference point. During cropping, $m_F$ is greatly reduced, $f_{FPR}$ is greatly increased, and $m_R$ is also reduced, being limited to "short reversals" (the distance traveled in one or two head sweeps, or about 0.5 mm (*Gray et al., 2005*); during local search, reverse runs are almost always "long" (the distance traveled in at least three head sweeps). During ranging, $m_F$ is greatly increased, $f_{RPF}$ is reduced, and reversals are long. Cutoff values for search modes, inferred from behavioral data (*Wakabayashi et al., 2004*; *Gray et al., 2005*; *Fujiwara et al., 2002*; *Hills et al., 2004*) were: *Dwelling* – short forward run length ($m_F < 0.5$ mm), high reversal frequency ($f_{FPR} > 6.0$/min), short reversals ($m_R < 0.5$ mm); *Local search* – moderate forward run length (0.5 mm $\leq m_F < 5.0$ mm), moderate reversal frequency (2.0/min $\leq f_{FPR} < 6.0$/min), non-short reversals ($m_R \geq 0.5$ mm); R*anging* – long forward run length ($m_F \geq 5.0$ mm), low reversal frequency ($f_{FPR} < 2$/min), non-short reversals ($m_R \geq 0.5$ mm).

## Data archive
All data and the analysis program are publicly available at doi:10.5061/dryad.35qv6.

## Acknowledgements
This research was supported by a grant MH051383 from the National Institute of Mental Health to SRL.

## Additional information

### Funding

| Funder | Grant reference number | Author |
| --- | --- | --- |
| National Institutes of Health | RO1 MH051383 | Steven B Augustine<br>Kristy J Lawton<br>Theodore H Lindsay<br>Tod R Thiele<br>Serge Faumont<br>Rebecca A Lindsay<br>Matthew Cale Britton |

The funders had no role in study design, data collection and interpretation, or the decision to submit the work for publication.

### Author contributions
William M Roberts, Modeling, Conception and design, Analysis and interpretation of data, Drafting or revising the article; Steven B Augustine, Design and construction of tracking system, Acquisition of data; Kristy J Lawton, Rebecca A Lindsay, Matthew Cale Britton, Behavior, Acquisition of data; Theodore H Lindsay, Electrophysiology, Acquisition of data; Tod R Thiele, Neuronal ablations, Acquisition of data; Eduardo J Izquierdo, Chemotaxis simulations, Analysis and interpretation of data; Serge Faumont, Spotted worm tracking method, Acquisition of data; Navin Pokala, Cornelia I Bargmann, Molecular biology, Contributed unpublished essential data or reagents; Shawn R Lockery, Modeling, Conception and design, Acquisition of data, Analysis and interpretation of data, Drafting or revising the article

### Author ORCIDs
Cornelia I Bargmann [ID] http://orcid.org/0000-0002-8484-0618
Shawn R Lockery [ID] http://orcid.org/0000-0001-8535-7989

## Additional files

### Data availability

The following dataset was generated:

| Author(s) | Year | Dataset title | Dataset URL | Database and Identifier |
|---|---|---|---|---|
| Roberts WM, Augustine SB, Lawton KJ, Lindsay TH, Thiele TR, Izquierdo E, Faumont S, Lindsay RA, Britton MC, Pokala N, Bargmann CI, Lockery SR | 2016 | Data from: A stochastic neuronal model predicts random search behaviors at multiple spatial scales in C. elegans | http://dx.doi.org/10.5061/dryad.35qv6 | Dryad Digital Repository, 10.5061/dryad.35qv6 |

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
