## [Decision Letter]

Thank you for submitting your work entitled "A stochastic neuronal model predicts random search behaviors at multiple spatial scales in *C. elegans*" for consideration by *eLife*. Your article has been reviewed by favorably evaluated by Eve Marder (Senior Editor) and three reviewers, one of whom, Ronald L. Calabrese, is a member of our Board of Reviewing Editors.

The reviewers have discussed the reviews with one another and the Reviewing Editor has drafted this decision to help you prepare a revised submission.

Summary:

The authors present a kinematic analysis of random local search behavior in *C. elegans* at high resolution. This analysis leads to a Markov model of underlying neuronal network based on the *C. elegans* connectome. This inherently stochastic model not only accounts for the data but also predicts the sign and strengths of key connections which are then confirmed by electrophysiology/optogenetics. The model can be adjusted easily to account for other types of random searches by adjusting parameters compatible with sensory input or modulation. Further the model can be expanded to incorporate directed searches as in chemotaxis and can also account for 'deterministic' behavior such as escape movements. The model also accounts for counter intuitive results on dwell times associated with genetic manipulation and laser ablation of key command neurons. These findings conceptually inform experiments in the mammalian sleep network. Altogether this work is an impressive display of the power of modeling when combined with detailed experimental manipulation. The authors make a strong effort to cast their results in the context of other efforts to measure search motion and define pause states. They also explicitly address more neuronal-based determinist models of the network underlying random searches.

Essential revisions:

1) Results, second paragraph: The claim that this study represents "a 10-fold improvement over previously published tracking systems" seems to be a stretch. In some sense, centerline-tracking experiments operate at a much finer resolution than tracking a single dot on a worm, with papers such as Brown, et al., PNAS (2013) tracking many more worms with more detail and higher resolution. I agree that the authors appear to have done a fine job at spot tracking, but we don't believe that they have a claim to novelty here.

2) The authors should address the Stephens et al. (2011) paper in a different manner than they do here (particularly in the Discussion section, fifth paragraph). To begin with, the statement that the "mathematical relationship between tangential velocity and phase velocity (so defined) has not been delineated, but is likely to be complex" largely due to slippage between the worm and the substrate seems an overstatement. If the paper is interested in modeling the animal's neural control of motion, then shouldn't the model be more concerned about the dynamics actually being controlled – in this case, the postural dynamics? And while it is indeed theoretically possible to produce thrust without advancing the phase, the collection of papers from Stephens et al. show that the worms' dynamics lie almost entirely on a manifold of remarkably small dimension, showing that these types of potential postural changes occur only rarely. Moreover, the authors themselves admit (in the subsection “Wild type locomotion”, fifth paragraph) that "on an agar surface, the worm moves without slipping."

In addition, the authors state that the Stephens et al. (2010) paper shows that postural modeling does not accurately model the worm when its center of mass trajectory follows an arc. In fact, the cited paper shows the necessity of looking at arcing trajectories between reorientation events and not just using a run-and-tumble analogy from bacteria, showing that shape-space dynamics form a predictive relationship with foraging trajectories.

All this being said, we are not disputing the authors' modeling choice of only using the midpoint of the worm's body. We have no arguments that there is utility in using a simplified description of a system to gain quantitative insight, but we want to see the authors distinguish themselves from the Stephens (2011) paper differently, focusing instead on the fundamental difference of modeling via a hidden Markov model instead of fitting parameters in a set of deterministic/stochastic differential equations. The choice of measuring Euclidean velocity instead of phase velocity is a modeling choice. Put another way, if someone with comparable amounts of shape-space data were to fit a hidden Markov model of your form to their data using phase velocity instead of velocity, would they likely obtain the same results?

3) In Figure 4, could the result that there is no postural stereotypy entering into state X be the result of an over-zealous "pause" caller. Specifically, this model calls many more things pauses than other approaches because of the nature of the sub-frame-resolution hidden Markov fit. What do these plots look like if only the longest visits to the X states are included?

4) Is there any evidence that the time scales of neural activity in *C. elegans*' motor neurons can be as short as the pause states being predicted? Although the model is supposed to be an abstraction of what's occurring in the worm's nervous system, one should expect to predict reasonable numbers for this nonetheless.

5) In the first paragraph of the subsection “Wild type locomotion” the authors claim that their "tracking system is capable of revealing briefer visits" to the pause state and that this is the reason why their measured dwell time is much shorter than previously measured. Again, we are not yet convinced that this method is much more accurate than the posture tracking used in Stephens et al. (2011) to measure the forward dwell time. The Stephens paper, however, uses a slightly different definition of the dwell time, using cessation of the forward phase velocity instead. Although the paper here does not measure phase velocity, they could measure the dwell time between visits to the reversal state (i.e. ignoring F->P->F type transitions). What happens if this definition is used? Does the same result emerge?

6) A more general comment is that we would like to understand more what type of neural activity the authors would predict based on their model, connectome data, and polarity results from papers such as Rakowski (2013) despite imaging in a freely-behaving worm being well outside of the scope of this paper (although not nearly as impossible as the authors seem to suggest in the Discussion section). If this experiment were performed, what is the most likely neural instantiation of this model that is consistent with the current literature? The strength of this model is that it makes an attempt at getting at how this circuit may actually function. Accordingly, if possible, a concrete prediction or predictions to this end would greatly increase the value of this paper to researchers and could guide experimentalists performing imaging in freely-behaving animals in their measurements and analysis.

[Editors’ note: a previous version of this study was rejected after peer review, but the authors submitted for reconsideration. The previous decision letter after peer review is shown below.]

Thank you for choosing to send your work entitled "A stochastic neuronal flip-flop circuit regulates random search during foraging behavior" for consideration at *eLife*. Your full submission has been evaluated by Eve Marder (Senior editor) and three peer reviewers, one of whom, Ronald L. Calabrese, is a member of our Board of Reviewing Editors, and the decision was reached after discussions between the reviewers. Based on our discussions and the individual reviews below, we regret to inform you that your submission in its present form is not suitable for publication in *eLife*. That said, if you feel that you can adequately rebut, answer the reviews with a combination of rewriting or new work, we would be willing to consider a new submission of this material, as it clearly asks an important and interesting set of questions.

Consensus review:

The authors present a kinematic analysis of random local search behavior in *C. elegans*. The then use a Hidden Markov Model to quantitatively model such searches based their kinematic analyses. They then identify this HMM with known aspects of the worm connectome to understand the neural implementation of switches between different behavioral states. In particular, the model contains two populations of neurons, each controlling either forward or reverse locomotion, with the overall network resulting in four possible behaviors: forward, reverse, and two pause states. They perform a maximum likelihood fit of the model to measured time series, making predictions about the effect of neural ablations, the sign and strength of synaptic connections, effects of perturbations to membrane potentials. Moreover, they phenomenologically model a potential underlying neural mechanism for different search modalities.

The subject matter of the paper is definitely one of broad interest, and if the model is correct it would provide us with a substantive understanding into the neural implementation of the forward/pause/reversal/pause(?) dynamics it would be a major contribution. However, the reviewers are not fully convinced that the model is the simplest possible explanation of the animal's behavior and that the authors have shown that a clear connection can be drawn to neural correlates.

Enthusiasm for the subject matter and the potential neural correlates vs. concerns about the appropriateness of the approach and its rigor were weighed differently by the reviewers initially, but in consultation the concerns predominated. The detailed reviews of the expert reviewers are appended but their concerns are summarized below.

1) The model is not adequately placed within the context of previous work in the field. There have been other papers attempting to understand both exploitation/exploration behavioral generation (e.g. Flavell et al., Cell, 2013) and forward/reversing dynamics (e.g. Stephens, PNAS, 2011).

Moreover, a direct comparison to (Rakowski et al., 2013) is needed. This work, which is cited as Ref. 12, also develops a model of the command neuron network and uses this model to predict behavioral states and the effects of ablations.

2) The existence of two different pause states X and Y is not well supported in the data and the HMM does not have the power to confirm their existence.

3) The mapping of the HMM onto the underlying neuronal network is not convincing. In particular there little data in this paper or in any of the citations to support the contention that pause state X = all command neurons off and pause state Y = all command neurons on. Mapping synaptic weights onto coupling in the HMM is not rigorous.

For the paper to be suitable for *eLife* the following extensive changes would have to be made.

1) Relax the claims about the model's prediction of synaptic weights.

2) The authors need to establish firmly that tracking a single point provides an unbiased estimate of trajectory in light of Stephens, PLoS One (2010). Given their extensive data set, we presume that converting their point imaging into centerline tracking would be extremely difficult and time consuming, but the authors should at least place their analysis in this light. They should also bolster their arguments that there are two behaviorally distinct pause states and discuss how they are related to the two pause states of Stephens.

3) The authors should differentiate their work with respect to Rakowski (2013). Does their model provide different predictions? Can they be disambiguated? It would be great if they could show this, but minimally, they should suggest what experiments should be performed in the future.

4) Similarly – what measurements should be made to distinguish X and Y pause states map onto the electrical activity of the command neuron network at least at some coarse-grained level?

*Reviewer #1:*

The authors present a kinematic analysis of random local search behavior in *C. elegans* at high resolution. This analysis leads to a Markov model of underlying neuronal network based on the *C. elegans* connectome. This inherently stochastic model not only accounts for the data but also predicts the sign and strengths of key connections which are then confirmed by electrophysiology/optogenetics. The model can be adjusted easily to account for other types of random searches by adjusting parameters compatible with sensory input or modulation. Further the model can be expanded to incorporate directed searches as in chemotaxis and can also account for 'deterministic' behavior such as escape movements. The model also accounts for counter intuitive results on dwell times associated with genetic manipulation and laser ablation of key command neurons. These findings conceptually inform experiments in the mammalian sleep network.

Altogether this work is an impressive display of the power of modeling when combined with detailed experimental manipulation. Moreover the approach complements and expands other efforts using unbiased approaches that make no assumptions about whether and how behavioral categories can be mapped to specific states of the nervous system. This approach allows mapping of states on to the neuronal network. The paper is very clearly and crisply written. The figures are clear and contain useful data well organized. All essential data is presented.

This reviewer is not an expert in Markov models and the associated mathematics, so I would defer to the experts on any potential flaws in this analysis, but the behavioral analyses are very straightforward and carefully done, the electrophysiology is state of the art for *C. elegans*, and the mutant analysis and cell ablation experiments seem carefully controlled. The statistical analysis seems appropriate but I defer to the other more expert reviewers. Methods are a model of completeness. There are no major concerns noted by this reviewer. I really enjoyed reading this paper and I learned a lot.

*Reviewer #2:*

Roberts et al. present a study combining behavioral measurements of *C. elegans* locomotion, modeling of a two-state stochastic system, and measurements of functional connectivity between two command interneurons. The authors claim that using the stochastic model, they can make nontrivial predictions about the state of the underlying neural network, solely based on behavioral measurements. They also claim that using behavioral data and their two-state model they can predict the strengths of synaptic connections between interneurons. If these claims were valid, this would be a significant achievement and worthy of publication in *eLife*, although the novelty of this work is somewhat compromised by another recent publication (Rakowski et al., 2013).

My major concerns are:

1) The claim that there are two distinct pause states resulting from different levels of activity in the command interneurons is not supported by evidence presented in this paper or cited work.

2) The stochastic switch model is not grounded enough in the biology of the system to justify comparing model fit parameters to measurements made on individual neurons. In particular the comparison of the model fit parameters labeled "synaptic weights" to actual synaptic connections is inappropriate.

Behavioral Measurements and Hidden Markov Model:

In Figure 1, the authors present data obtained by tracking a fiducial marker painted onto the back of a freely moving worm. They show a distribution of "velocities" (1D) that appear to be the sum of 3 distributions, which they label "reverse, pause," and "forward." In Figure 1E, they then show a velocity vs. time graph that has been segmented into these states by thresholding the velocity at +/- 0.05 mm/s. In Figure 2E, they show a similar velocity vs. time graph that has been similarly segmented using a Hidden Markov Model. The claim is that this model does a much better job of describing the behavioral state than the process used in Figure 1E, although I cannot find any quantitative measures in the paper to support this assertion (more on this later).

The HMM differs from simple thresholding by (1) assigning different prior probabilities to transitions between states and staying in the same state and (2) including 2 pause states (called X and Y). The states of the HMM are related to behavior and to the stochastic flip flop model as follows.

F: worm is moving forward. "F" unit is active and "R" unit is inactive. What F and R represent is unclear in the paper? They are described as "binary stochastic elements," (subsection “The Stochastic Switch Model”, third paragraph), "single binary neuron-like units" (Figure 2 caption), and as a collection of neurons as in "self connections represent synaptic connections between neurons comprising a given unit." (Figure 2 caption). I will come back to this under Stochastic Switch Model, but for discussion of the hidden Markov model, the distinctions are not important.

R: worm is moving backward. "R" unit is active and "F" unit is inactive.

X: worm is not moving/moving slowly. Both units are inactive.

Y: worm is not moving/moving slowly. Both units are active.

A great deal of the argument in this paper relies on the existence of two pause states and of the identification of these two behaviorally identical pause states with specific patterns of activity in command interneurons, so I will go over in some detail the arguments the manuscript advances:

1) Using a speed threshold, pauses during forward->reverse transitions were found to be longer than pauses during reverse->forward transitions.

2) "When both F and R are OFF (state X) it is assumed that movement ceases, consistent with studies showing that genetic ablation or silencing of all command interneurons induces prolonged pauses (Kawano et al., 2011; Zheng et al., 1999)." Zheng et. al (Zheng et al., 1999) report that glr-1::ICE (klys36) "moved significantly slower than wild type worms and also had long pauses during which no movement occurred." Kawano et al. (Kawano et al., 2011) report that silencing all of AVA, AVB, AVE, AVD, and PVC led to prolonged pausing with the body in a straight position and that using a combination of genetic and laser ablation to destroy these same methods resulted in worms that were "kinked" with odd body postures.

3) "In the event that F and R are simultaneously ON (state Y), the resulting motor commands are assumed to conflict at the level of the motor neurons of the body wall muscles, resulting in a second motionless state, as has been observed (Kawano et al., 2011)." The way this sentence is structured may leave a mistaken impression of what is reported in Kawano et al., 2011. Kawano et al., 2011 does not show that when F&R command interneurons are simultaneously active, worms stop moving. Instead it shows that in innexin mutants (unc-7 and unc-9), the "kink" state is correlated with the A and B type motorneurons having similar levels of calcium.

4) In the second paragraph of the subsection “Wild type locomotion“. Removing state Y (i.e., setting aFY=aRY=0) caused a large and highly significant reduction in the summed log-likelihood of the 5 wild type cohorts (𝑝 < 10^-100^) demonstrating that the second pause state greatly improves the fit.

Taken individually or together, these arguments fail to support the existence of multiple pause states or that these states can be reached by co-activation or inactivation of command neurons.

Argument 1 merely shows that the speed of a dot on a worm's midline recovers more quickly in the reverse to forward transition than the forward to reverse. This could be due to measurement artifacts, biomechanical differences, differences in the response latencies of A and B motorneurons, or differences elsewhere in the neural network.

Argument 2 shows that permanent inactivation of all command motor neurons leads to a worm with severe motion defects. It's quite a stretch to go from there to saying that transient decreases in the activities of these neurons will lead to immediate cessation of movement. It does seem at least plausible that if both forward and reverse command neurons are "off," the worm stops moving, although I question whether this cessation would really be immediate, especially since forward motion results from the propagation of a wave down the body due to proprioreceptive feedback (Wen et al., 2012)

Argument 3 does not show that co-activation of command neurons leads to pausing in wild type animals.

Argument 4 suffers from several flaws. First, the hidden Markov model used with the sole observable being the speed of a dot on the worm's midline may not be an appropriate description of the system (discussed in other comments). Second, the fact that one model produces a better fit than another does not show that either model is true. IE: that adding a state to the HMM improves the model's fit does not prove that state exists.

There are other ways we might improve the model fit that are incompatible with the stochastic switch model: Allowing the HMM to have 8 degrees of freedom rather than 6; adding a third pause state or a second forward state with a different velocity distribution; allowing direct transitions between X and Y or F and R. If any of these produce a significant improvement in the likelihood of the data given the model, what would that imply for the stochastic switch model? I would also like to see the likelihood ratio test applied to compare a model where the state is determined by simple thresholding (as in Figure 1E) to the 4-state HMM.

In summary, the authors present only weak evidence that there even are two distinct pause states. There is no data presented here or in the cited literature that show that simultaneous activation or deactivation of forward and reverse command neurons causes pausing. There is no evidence given for the correspondence of the two pause states in the HMM with two distinct states of the command neuron network. Finally, beyond a very thin justification about metabolic efficiency, there is nothing to support the assignment of state X to the R->F transition and state Y to the F->R.

I think the idea that the same behavioral state (pausing) can be the result of opposite patterns of activity in the command neurons is intriguing and worth pursuing.

The authors write that "The currently available set of optical probes of neuronal activity do not have sufficient temporal resolution to allow direct observation of the underlying states of the command network in intact, freely moving animals." The recovery time of GCaMP3 is ~300 ms and faster dynamics can be inferred by deconvolving the observed signal with the known GCaMP3 filter (Kato et al., 2014). And faster indicators, like GCaMP6f, are now available. The mean duration of inferred state X pauses is 480 ms and 20% of inferred state Y pauses are over 300 ms (Figure 2—figure supplement 3). The authors assert that in state X the activity in all command neurons will be much lower than in state Y. Surely in these longer pauses it would be possible to make a measurement that verifies this prediction.

A second approach would be to use optogenetic tools to determine the pause state. In state X (all command neurons off), ChR2 activation of AVB should induce forward locomotion by hyperpolarization of AVA via NpHR/halo should not. In state Y (all command neurons on), hyperpolarization of AVA should lead to forward locomotion.

The "Stochastic Switch Model":

The stochastic switch model uses a nonstandard description of neural state. Instead of parameterizing the state of a neuron by a continuous variable, e.g. membrane potential, the neuron is treated as a binary element, existing in one of two discrete states. Synaptic connections are also represented/modeled in a non-standard way. Normally, if A is presynaptic to B, activity in A causes a change in the membrane potential of B by an amount set by the weight of the synaptic connection from A to B. In this model, if A is active, the probability that B will spontaneously switch states is modified by an amount that depends exponentially on the "weight" of the connection. As a model of coupled two state systems, there is nothing inherently wrong with this approach, but comparing the resulting model fit parameters to biological parameters is confusing and potentially misleading.

Figure 5, "the stochastic switch model correctly predicts the sign and strength of synaptic connections" shows the confusion generated by the choice of terminology. Figure 5A shows model fit parameters that have no obvious biological interpretation, but the figure caption identifies these fit parameters as "synaptic weights." Then Figure 5B and C show "synaptic currents" (currents that result from functional, but not necessarily direct synaptic, connections) measured by recording the amount of current received by neuron B when neuron A is activated. The figure invites the reader to compare 5A to the 5B-5E even though 5A shows fit parameters from a model that does not even have synaptic currents. This presentation is at best confusing and at worst misleading.

In their description of the stochastic switch model, the authors should only use the term "binary stochastic elements" to describe F and R, and choose a term other than "synaptic weights" to describe the model fit parameters (hF, wff etc.), which are not synaptic weights as the term is commonly used. The chosen terminology will lead all but the most careful readers to misunderstand the work that has been done.

Measurements of functional connectivity, effect of ablations, and extensions to chemotaxis:

The measurements of functional connectivity appear to be well done; I have no major concerns with these, beyond that already noted in the text: different levels of ChR2 expression in AVA and AVB could complicate the interpretation of these results. Perhaps the current evoked by ChR2 activation could be directly measured and calibrated. In most of the literature about the *C. elegans* motor circuit, there is assumed to be reciprocal inhibition between AVA and AVB, but I think Figure 5 shows the first direct measurement, a significant contribution.

The rest of the paper (Figures 4, 6-8) relies on the results of the modeling and behavioral experiments.

*Reviewer #3:*

The submission by Roberts, et al., uses a Hidden Markov Model to quantitatively model the locomotory behavior of *C. elegans*, with a particular emphasis on understanding the neural implementation of switches between different behavioral states. In particular, the model contains two populations of neurons, each controlling either forward or reverse locomotion, with the overall network resulting in four possible behaviors: forward, reverse, and two pause states. They perform a maximum likelihood fit of the model to measured time series, making predictions about the effect of neural ablations, the sign and strength of synaptic connections, effects of perturbations to membrane potentials. Moreover, the phenomenologically model a potential underlying neural mechanism for different search modalities.

In general, I found the work to be careful done, and understanding the mechanisms of behavioral control is an important question in the field. My major concerns have to do with the novelty of the work.

As far as novelty is concerned, there have been other papers attempting to understand both exploitation/exploration behavioral generation (e.g. Flavell, et al., Cell, 2013) and forward/reversing dynamics (e.g. Stephens, PNAS, 2011), with both of the cited examples leading to the generation of long behavioral time scales. It would be useful for the authors to place their results in the context of these papers. In particular, the Stephens (2011) paper uses an analogous (but non-identical approach) of fitting a dynamical system to the data and makes (highly accurate) predictions about forward/reversal switching rates using Arrhenius-like stochastic dynamics. In addition, an earlier paper from the same group (Stephens, PLoS Comp Bio, 2008) shows that a 4-state system emerges naturally from a data set. Here, this set-up is assumed a priori.

Specific comments:

1) In Figure 4B, I find the excuse for omitting error bars to be relatively weak. Minimally, bootstrapped confidence intervals could be found, which would be significantly better than the current lack of any measure of variability.

2) In the fifth paragraph of the subsection “Genetic effects on command neuron function” and Figure 6. The claim is made that "changes in synaptic strength dominate the effects of changes in membrane potential on ΔF and ΔR." This certainly seems true for ΔF, but both models appear to give more-or-less identical predictions here for ΔR.

3) In the first paragraph of the subsection “The Stochastic Switch Model and other random search behaviors“. It is claimed that "changes are required in at least three weights to produce the full range of search behaviors." This is not actually shown anywhere in the paper, just that three weights are sufficient to induce the full range. A supplementary figure to this effect would be useful.

4) The predicted value for the dwell time in the forward state is not comparable to the rest of the literature, as a different definition is used here. What would happen if a standard definition was used? Does the same rate result?

---

## [Author Response]

Essential revisions: 1) Results, second paragraph: The claim that this study represents "a 10-fold improvement over previously published tracking systems" seems to be a stretch. In some sense, centerline-tracking experiments operate at a much finer resolution than tracking a single dot on a worm, with papers such as Brown, et al.,

*PNAS (2013) tracking many more worms with more detail and higher resolution. I agree that the authors appear to have done a fine job at spot tracking, but we don't believe that they have a claim to novelty here.*

None of the published centerline tracking papers cite the spatial resolution of the method, so the argument against our claim to novelty seems to be unfounded. However, we are willing to limit this claim to measurement oftangential velocityand have adjusted the manuscript accordingly (Introduction, last paragraph). Note that our demonstration of higher velocity resolution than other methods is significant because it provides with an explanation for why we obtained a lower value of mean dwell time in the forward state than previous studies (subsection “Wild type locomotion”, first paragraph).

In preparing this manuscript, we reviewed all *C. elegans* centerline tracking papers, including Brown et al., PNAS (2013) cited above, and the complete set of the Stephens/Bialek papers. As far as we can tell, none of them presents an explicit statement of spatial resolution or the information needed to compute spatial resolution (pixel size on the camera chip, and overall optical magnification), nor do any of them show data from motionless worms (Figure 1—figure supplement 1), which would at least yield an estimate of resolution in the limiting case of no worm movement. Thus, we find no basis in the literature for the referee’s statement that centerline methods have higher spatial resolution.

Whereas we agree that centerline tracking is superior when the worm’s *posture* is the key datum, here our focus is on *tangential velocity*, for which centerline tracking is not the most direct measure possible. As we noted in the main text of the paper (Results, end of first paragraph), the centerline method computes tangential velocity by following virtual points rather than physical marks on the body. This is done by fitting a spline to the worm’s boundary, breaking the spline into a fixed number of segments, and computing the velocity of segment endpoints. The spline method is inaccurate because the tail of the worm, being faint, is differentially resolved from frame to frame. As a result, the length of the spline, and thus the distance between segment endpoints varies significantly (e.g. Figure 1B of Karbowski et al., 2008; Figure 2 of Cronin et al., 2005 where centerline head and tail are chopped off and misaligned in comparison to real worm; Figure 1 panel B of Karbowski et al. 2006). Thus the spline method adds segmentation noise to the velocity measurements. In contrast, the spot tracking method follows a sharply defined, high-contrast mark that is essentially unaffected by image segmentation noise. Our literature review revealed only one other *C. elegans* tracking paper that cites the accuracy of spot of tracking. Here the tracking object is a fluorescent neuron and the resolution is given as 5 µm (Guo, 2009). This paper is the basis of our statement that our method affords a 10-fold improvement, as the stage encoders we used had a resolution of 0.5 µm. Our dead worm experiments verify that encoder precision is the limiting factor in the overall spatial resolution.

*2) The authors should address the Stephens* et al. *(2011) paper in a different manner than they do here (particularly in the Discussion section, fifth paragraph).*

We agree with the reviewers’ main points (see details below): (i) Worm's do not slip under our conditions, nor under those of Stephens et al. (ii) Thrust can be produced when phase velocity is zero. (iii) Midpoint tracking is a useful simplified description of the system. The main force of this critique is thus the request to distinguish our work from Stephens' in a different manner – by using the thought experiment in which phase velocity is substituted for tangential velocity when fitting and analyzing our Markov model. Would the two definitions of velocity yield the same results? We have now adopted this excellent suggestion in the Discussion (fourth paragraph).

*To begin with, the statement that the "mathematical relationship between tangential velocity and phase velocity (so defined) has not been delineated, but is likely to be complex" largely due to slippage between the worm and the substrate seems an overstatement.*

We have removed the quoted statement, and the discussion of slippage. (As point of fact, however, we did not say that the complexity of the relationship between tangential and phase velocities is due to the possibility of slippage. Rather, this complexity is due the fact that tangential velocity in the worm is fully determined by thrust, and the higher order components of the worm’s shape, which are capable of generating thrust, are ignored by definition in computing phase velocity.)

*If the paper is interested in modeling the animal's neural control of motion, then shouldn't the model be more concerned about the dynamics actually being controlled* – *in this case, the postural dynamics?*

We could just as easily argue that from the perspective of the command neurons, what is being controlled is the presence or absence of thrust, and its direction.

*And while it is indeed theoretically possible to produce thrust without advancing the phase, the collection of papers from Stephens et al.show that the worms' dynamics lie almost entirely on a manifold of remarkably small dimension, showing that these types of potential postural changes occur only rarely.*

Shape components 3 and 4 account for approximately 30% of the shape variance (Stephens et al., 2008, Figure 2B), so it is hard to see how they can be characterized as “rare.” Moreover, these two components meet necessary and sufficient conditions for generating thrust: a gradient of curvature along the worm’s centerline (Gray 1946; Gray 1953; Gray 1964); Thus there are non-rare components that are generating thrust.

Moreover, the authors themselves admit (in the subsection “

*Wild type locomotion”, fifth paragraph) that "on an agar surface, the worm moves without slipping."*

Yes, we do believe that slip is generally minimal. Now, we no longer use slippage as a means of illustrating the differences between the two methods of analyzing behavior.

*In addition, the authors state that the Stephens et al.(2010) paper shows that postural modeling does not accurately model the worm when its center of mass trajectory follows an arc.*

We are not saying that postural modeling – defined in terms of *all* shape components – is inaccurate; we are actually making a much more restricted assertion. We are merely saying that phase velocity, because it is based on components 1 and 2 alone, is not necessarily an accurate measure of tangential velocity. In particular, this mismatch is accentuated in arcs, including omega turns, where other thrust producing shape components are strongly engaged.

*In fact, the cited paper shows the necessity of looking at arcing trajectories between reorientation events and not just using a run-and-tumble analogy from bacteria, showing that shape-space dynamics form a predictive relationship with foraging trajectories.*

We agree with this point. The work by Iino and colleagues (Iino, 2009) has shown that spatial orientation behavior is the product of two mechanisms: run-and-tumble (klinokinesis) and weathervaning (klinotaxis). However, klinotaxis, being deterministic rather than random, is outside the scope of the paper. We now make it clear that the type of chemotaxis modeled here is “random-walk chemotaxis” (Discussion, first paragraph).

*All this being said, we are not disputing the authors' modeling choice of only using the midpoint of the worm's body. We have no arguments that there is utility in using a simplified description of a system to gain quantitative insight, but we want to see the authors distinguish themselves from the Stephens (2011) paper differently, focusing instead on the fundamental difference of modeling via a hidden Markov model instead of fitting parameters in a set of deterministic/stochastic differential equations. The choice of measuring Euclidean velocity instead of phase velocity is a modeling choice. Put another way, if someone with comparable amounts of shape-space data were to fit a hidden Markov model of your form to their data using phase velocity instead of velocity, would they likely obtain the same results?*

We would not get the same results (rate constants, weights, etc.) because thrust (hence movement) can be generated even when phase velocity is zero (see point (ii) of agreement above). Thus some of the pauses identified phase velocity, would not be identified as pauses by tangential velocity because the worm is still moving. Would all pauses in tangential velocity be detected by the phase velocity method? We speculate that this is mainly true, because when tangential velocity is zero, net thrust is zero, so none of the shape components are generating thrust, including components 1 and 2.

3) In Figure 4, could the result that there is no postural stereotypy entering into state X be the result of an over-zealous "pause" caller. Specifically, this model calls many more things pauses than other approaches because of the nature of the sub-frame-resolution hidden Markov fit. What do these plots look like if only the longest visits to the X states are included?

We tested for this possibility by reanalyzing the average posture at FX transitions after eliminating all but long pauses, and found a similar lack of postural preference at FX transitions. This result held for any minimum pause duration up to 2 s; longer dwells in state X were too rare to analyze. We have added this result to the text (subsection “Wild type locomotion”, last paragraph).

*4) Is there any evidence that the time scales of neural activity in C. elegans' motor neurons can be as short as the pause states being predicted? Although the model is supposed to be an abstraction of what's occurring in the worm's nervous system, one should expect to predict reasonable numbers for this nonetheless.*

To our knowledge, there is only one report of voltage recordings from body wall muscle motor neurons in *C. elegans* (Liu, 2014). Interestingly, these motor neurons, which happen to be in the pool of reverse motor neurons, exhibit distinct up and down states. Theoretically, the down states could be pauses of the state-X type in our model when they coincide with down states in forward motor neurons, whereas the up states could be pauses of the state-Y type in our model when they coincide with up states in forward motor neurons. Unfortunately, no simultaneous recordings of reverse and forward neurons were made so it is impossible to determine the duration of possible pause states. The Liu paper is now referenced in the third paragraph of the subsection “The Stochastic Switch Model”.

5) In the first paragraph of the subsection “Wild type locomotion” the authors claim that their "tracking system is capable of revealing briefer visits" to the pause state and that this is the reason why their measured dwell time is much shorter than previously measured. Again, we are not yet convinced that this method is much more accurate than the posture tracking used in Stephens et al.

*(2011) to measure the forward dwell time. The Stephens paper, however, uses a slightly different definition of the dwell time, using cessation of the forward phase velocity instead. Although the paper here does not measure phase velocity, they could measure the dwell time between visits to the reversal state (i.e. ignoring*
F*->P->*F
*type transitions). What happens if this definition is used? Does the same result emerge?*

We have added this calculation to the text. The resulting mean dwell time in the forward state using the standard fixed velocity threshold of 0.5 mm/s and ignoring FPF transitions was 9.13 ± 0.15 s. which matches the value obtained by others using the same threshold (8.98 ± 0.57 s) (Rakowski, 2013). This result has been added to the text (subsection “Wild type locomotion”, first paragraph).

*6) A more general comment is that we would like to understand more what type of neural activity the authors would predict based on their model, connectome data, and polarity results from papers such as Rakowski (2013) despite imaging in a freely-behaving worm being well outside of the scope of this paper (although not nearly as impossible as the authors seem to suggest in the Discussion section). If this experiment were performed, what is the most likely neural instantiation of this model that is consistent with the current literature? The strength of this model is that it makes an attempt at getting at how this circuit may actually function. Accordingly, if possible, a concrete prediction or predictions to this end would greatly increase the value of this paper to researchers and could guide experimentalists performing imaging in freely-behaving animals in their measurements and analysis.*

This is a great idea! We have now included a paragraph of predictions in the Discussion (fifth paragraph).

[Editors’ note: the author responses to the previous round of peer review follow.]

*Consensus review:*

*The authors present a kinematic analysis of random local search behavior in C. elegans. The then use a Hidden Markov Model to quantitatively model such searches based their kinematic analyses. They then identify this HMM with known aspects of the worm connectome to understand the neural implementation of switches between different behavioral states. In particular, the model contains two populations of neurons, each controlling either forward or reverse locomotion, with the overall network resulting in four possible behaviors: forward, reverse, and two pause states. They perform a maximum likelihood fit of the model to measured time series, making predictions about the effect of neural ablations, the sign and strength of synaptic connections, effects of perturbations to membrane potentials. Moreover, they phenomenologically model a potential underlying neural mechanism for different search modalities. The subject matter of the paper is definitely one of broad interest, and if the model is correct it would provide us with a substantive understanding into the neural implementation of the forward/pause/reversal/pause(?) dynamics it would be a major contribution. However, the reviewers are not fully convinced that the model is the simplest possible explanation of the animal's behavior and that the*

*authors have shown that a clear connection can be drawn to neural correlates. Enthusiasm for the subject matter and the potential neural correlates vs. concerns about the appropriateness of the approach and its rigor were weighed differently by the reviewers initially, but in consultation the concerns predominated. The detailed reviews of the expert reviewers are appended but their concerns are summarized below. 1) The model is not adequately placed within the context of previous work in the field. There have been other papers attempting to understand both exploitation/exploration behavioral generation (e.g. Flavell et al., Cell, 2013) and forward/reversing dynamics (e.g. Stephens, PNAS, 2011).*

The requested context is now provided in the Introduction.

*Moreover, a direct comparison to (Rakowski* et al.*, 2013) is needed. This work, which is cited as Ref. 12, also develops a model of the command neuron network and uses this model to predict behavioral states and the effects of ablations.*

The Rakowski paper is difficult to decipher at several key respects, but as we understand the text, reviewers’ points (i) and (ii) are almost certainly over-interpretations of what the Rakowski model (RM) achieves.

(i) Prediction of behavioral states. We agree that the RM enables one to predict/compute from a given weight matrix the quantity R = Tf/(Tf + Tb), where Tf and Tb are, respectively, the total time the animal spends in the forward and backward states during the observation period. Note, however, that R is not a state. Rather, it is the *steady-state* probability of forward locomotion (conditional on the animal not being in the pause state). The individual behavioral states themselves – forward, backward, and stopped, in their terminology – are *not* represented in the mathematics. It would therefore be impossible to derive mathematical expressions for mean dwell times in these states, nor could one simulate the network in time to produce a predicted sequence of behavioral states. Thus, it is simply not true that the RM predicts behavioral states.

(ii) Prediction of ablation effects. We agree that the RM computes the effects of ablations on the quantity *R*, but we don’t agree it predicts them. This is for the simple reason that the model is fitted to the ablation data, so it can’t be said to predict them. It should have been tested on data to which is was not fitted.

With respect to the question of ablation effects, we must call the reviewers' attention to the fact that the ablation data in Rakowski were not analyzed for statistical significance, and the Sternberg lab has confirmed this in personal communications. The absence of statistics means that the question of which purported "effects" are real, and which are false positives of false negatives simply cannot be addressed. Rakowski’s claims related to behavioral effects of ablations should therefore be put on hold until they have been subjected to the field’s common standard of validity.

In revising our manuscript, we nevertheless made the requested direct comparison be between the two models. We added a paragraph in Discussion that is dedicated to the comparison (sixth paragraph). We chose not to mention the missing statistics.

*2) The existence of two different pause states X and Y is not well supported in the data and the HMM does not have the power to confirm their existence.*

This comment raises two separate questions: (a) Does the addition of a second pause state significantly improve the ability of the model to fit the data? (b) Do the two pause states correspond to “real’” physiological states? These two questions raise fundamentally different issues, which we consider separately below:

a) Does the addition of a second pause state significantly improve the ability of the model to fit the data? Yes, the second pause state improves the fit well beyond what can be attributed to the addition of two free parameters in the model (p<10^-100^ by likelihood ratio test (see subsection “Wild type locomotion“, fourth paragraph and Table 1). For this test we constrained the rates into state Y to be aFY = aRY = 10^-10^ s^-1^, which effectively eliminates state Y (pY < 10^-18^); model B in Table 1). Because the model with one pause state (model B in Table 1) is a special case of the model with two pause states (model A in Table 1), we could apply the likelihood ratio test. The likelihood ratio test is designed specifically for the purpose of comparing two models, one of which is a constrained version of the other, by comparing the reduction in likelihood in the constrained model with the reduction that can be attributed to having fewer free parameters. We further showed that there is almost no benefit in allowing direct transitions between F and R states (model C in Table 1; equivalent to the most general model with one pause state and all 6 transition rates left unconstrained). We conclude that our synaptic model with two pause states, which has 6 free parameters, fits the data far better than any similar model with only one pause state.

b) Do the two pause states correspond to “real’” physiological states? This is subtle question because “real” physiological state is not easily defined. Therefore, for the purpose of this rebuttal we will consider the strongest version of this question, which we infer to be the question that the reviewer is asking: Does the model have the power to confirm the hypothesis that state X corresponds to both pools of command neurons being “OFF”, whereas state Y corresponds to both pools being “ON”? Our answer is no, and we do not make this claim. Instead, we propose this as a working hypothesis that provides a useful framework in which to interpret results but which will need to be tested directly by physiological experiments when and if such experiments become technically feasible. In support of the working hypothesis, it is important to note that it is consistent with experimental observations beyond the well-known result that the forward command neurons promote forward locomotion and reverse command neurons promote reverse locomotion. It has also been shown that genetic ablation or silencing of both pools of command neurons induces long pauses (subsection “The Stochastic Switch Model, seventh paragraph), as expected for state X, and that worms pause when forward and reverse motor neurons simultaneously enter their activated state (in the eighth paragraph of the aforementioned subsection). Although the latter results are for the motor neurons, not the command neurons that drive them, the gap junction synapses known to exist between the command neurons and their motor neuron targets makes it reasonable to hypothesize that both pools of command neurons are also co-activated during these pauses, as expected for state Y. Thus, there is sufficient experimental evidence to justify proposing our working hypothesis. Additional evidence that states X and Y are “real” by physiological criteria comes from our results showing that the two states have different properties. State Y is intimately associated with the termination of reverse locomotion, whereas state X is associated with the end of forward locomotion (Figure 3). Because worms typically end reverse locomotion with a ventral bend, state Y is associated with this posture (Figure 4), whereas state X has no strong postural preference. The two states also have different mean dwell times (Figure 2—figure supplement 3), and state Y almost always leads into state F, whereas state X can exit to either state F or R (Figure 3). Thus, pauses that follow reverse locomotion are different from pauses that follow forward locomotion, and can therefore be regarded as two different states of the system. It remains to be shown experimentally how closely the states of the model correspond to activity states of the command neurons

We have made three changes to the manuscript which we hope will limit the risk that other readers will over-interpret what we are claiming with respect to the possible association of states of the model with activity states of the command neurons:

A) We have rewritten the paragraphs that explain the model assumptions and the possible relationships between the four states of the model and the activity states of the command neurons (subsection “The Stochastic Switch Model”, third, seventh and eighth paragraphs).

B) The title has been changed from “A stochastic neuronal flip-flop circuit regulates random search during foraging behavior,” to “A stochastic neuronal model predicts random search behaviors at multiple spatial scales in *C. elegans*.”

C) We have re-stated the primary conclusion of the paper (Discussion, fourth paragraph): “Nevertheless, the predictive and explanatory successes of the model demonstrate the utility of present conceptualization of the network.”

*3) The mapping of the HMM onto the underlying neuronal network is not convincing. In particular there little data in this paper or in any of the citations to support the contention that pause state X = all command neurons off and pause state Y = all command neurons on.*

As noted in response to critique 2, this is an assumption, not a contention, and we have taken steps to clarify this point in the Discussion.

*Mapping synaptic weights onto coupling in the HMM is not rigorous.*

We understand this statement to mean that we have not provided an equation for computing synaptic strength, as would be measured electrophysiologically in pairwise recordings quantifying synaptic currents, in terms of synaptic weights (*w*) as defined in the Stochastic Switch Model. We agree. As requested below, we have relaxed claims to have demonstrated prediction of synaptic weights in the electrophysiological sense of the term.

*For the paper to be suitable for eLife the following extensive changes would have to be made.*

*1) Relax the claims about the model's prediction of synaptic weights.*

We have relaxed these claims by pointing out that the term “synaptic weight” has different meanings in the neural-network versus the electrophysiological literature (neither field, of course, can be said to own the term). We now relax our claims by explicitly stating that the Stochastic Switch Model does not predict the magnitude of synaptic conductances (subsection “Synaptic weights in the stochastic switch model”, second paragraph). But it does predict aspects of functional connectivity, such as excitation or inhibition, because weights in the model are signed quantities. We also relax our claims of correspondence between the model and the electrophysiology by noting that our method of testing the model carries with it the assumption that functional connectivity between populations of neurons reflects the connectivity between the neurons in those pools that have the greatest effects on behavior (e.g. in ablations).

*2) The authors need to establish firmly that tracking a single point provides an unbiased estimate of trajectory in light of Stephens, PLoS One (2010). Given their extensive data set, we presume that converting their point imaging into centerline tracking would be extremely difficult and time consuming, but the authors should at least place their analysis in this light.*

The field recognizes two definitions of trajectory. One is path of the centroid (Pierce- Shimomura et al., 1999), defined as the midpoint of the smallest rectangle that can enclose the worm. The other is the path taken by one or more points on the worm’s centerline (Cronin et al., 2005); spot tracking is an instance of the latter.

Centroid trajectory. Stephens et al. are mainly concerned with the time series of postural measurements, not trajectory. The 2010 paper is an exception, but it considers only centroid trajectory. The main point of that paper is to show that the centroid trajectories of real worms can be recovered from the time series of their posture. The authors report, however, that their “eigenworm model” can be expected to do this only when the trajectory of the real worm is relatively straight. The degree of fit between real and computed trajectories is shown in the supplemental material; the fits may or may not be convincing, depending upon one’s point of view. No attempt is made by Stephens et al. to compute centerline trajectories.

Centerline trajectory. According to the theory of thrust generation in sinusoidal locomotion, established by Gray (1946, 1953) and extended to nematodes by (Gray and Lissman, 1964), the spot tracking method should provide an accurate estimate of centerline trajectory, an estimate that is probably more accurate than previous centerline methods (Cronin et al., 2005). In the case of ideal sinusoidal locomotion (no lateral slip), each segment of the body recapitulates the path of the segment ahead of it, like train cars on a sinuous track; thus the trajectory is just the path taken by the segments. On agar plates, *C. elegans* plows a furrow as it crawls; surface tension provides the necessary downward pressure. The walls of this furrow, which provide the lateral resistance for thrust generation, are what causes each segment to follow the one ahead of it. It is general practice in the field to dry agar plates slightly before use to reduce slip, ensuring that the body stays within the furrow, and locomotion approaches the ideal case. Under these conditions, a radially symmetric tracking perfectly centered on the body axis introduces no bias into trajectory. Bias occurs only when the center of mass of the tracking spot is eccentric to the centerline. This offset shifts the trajectory by the extent of the eccentricity and can increase/decrease the curvature maxima/minima. But in practice eccentricities were small, probably less than 1/10 of the worm’s diameter, because spots with greater eccentricities were quickly rubbed off by friction with the substrate; such recordings were discarded. Another potential problem is deformation of the spot as the worm’s body undulates, but this was almost certainly not an issue in our experiments because the long axis of the spot was small with respect to the radius of curvature of the worm. Note: The first step in Stephens’ image analysis was to compute the worm’s centerline (Stephens et al., 2008), but centerlines were used only to compute posture. They were not used to compute the centerline velocity, although this would have been possible as previously shown (Cronin et al., 2005).

Conversion of spot-tracking to centerline tracking. Centerlines can be recovered easily from spot tracking data. We have now done this for the entire wild type data set and included it in the paper (Figure 4). Our goal was to ask whether the worm adopts a particular posture as it enters the pause state, as Stephens et al. report. In this analysis we were somewhat limited by the fact that in the spot tracking method, pause posture can be delineated only up to the anterior-posterior level of the spot (approximately the midpoint). This limitation occurs because the spot’s future trajectory after the pause has not yet occurred, but it is mitigated by the fact that postures in which curvature is discontinuous do not occur in live, wild type worms. We find that pauses following reversals occur at a particular posture whereas pauses following forward locomotion do not. In contrast, Figure 5D of Stephens et al. (2008) indicates pauses occur a particular postures regardless of the direction of movement. One obvious reason for the discrepancy between the two studies is the difference in definitions of velocity used to identify the pause state. This topic is addressed in detail in the Discussion (fifth paragraph).

In summary, the eigenworm model is based on postures, not trajectories. Centroid trajectory can be recovered in the eigenworm model, but it is inaccurate unless the worm is moving comparatively straight. In contrast, the spot tracking method provides an accurate measure of centerline trajectory without significant bias.

*They should also bolster their arguments that there are two behaviorally distinct pause states and discuss how they are related to the two pause states of Stephens.*

(i) On the existence of two pause states. As stated above, we do not claim that the model has the power to confirm the existence of the two pause states. However, the model does provide reasons to believe such states might exist. As requested, we have reinforced these arguments, now devoting a whole new paragraph to the issue (subsection “Wild type locomotion“, fourth paragraph).

(ii) Discuss relationship to pause states of Stephens. Done. See fifth paragraph of the Discussion.

*3) The authors should differentiate their work with respect to Rakowski, 2013. Does their model provide different predictions? Can they be disambiguated? It would be great if they could show this, but minimally, they should suggest what experiments should be performed in the future.*

See Discussion, sixth paragraph.

*4) Similarly* –

*what measurements should be made to distinguish X and Y pause states map onto the electrical activity of the command neuron network at least at some coarse-grained level?*

See Discussion, third and fourth paragraphs.

Reviewer #3: 1) In Figure 4B, I find the excuse for omitting error bars to be relatively weak. Minimally, bootstrapped confidence intervals could be found, which would be significantly better than the current lack of any measure of variability.

Note: Figure 4 is now Figure 5.

Calculating confidence intervals would be nice for graphical representation, but was unnecessary to compute statistical significance of the differences shown in the figure, which was done using the likelihood ratio test, results of which are shown in Table 4. Computing the results shown in Table 4 was extremely time-consuming; >95% of the computation time for the entire data analysis presented in the paper was for computing the p-values in this table. Computing confidence intervals for Figure 5B (formerly Figure 4B) would require another ~10x increase in computation time, thousands of hours. Given that the confidence intervals would not be used to for the key statistical tests, which we already have done using the appropriate test, it would be an unnecessary burden to compute them.

*2) In the fifth paragraph of the subsection “Genetic effects on command neuron function “*and Figure 6. The claim is made that "changes in synaptic strength dominate the effects of changes in membrane potential on ΔF and

*ΔR." This certainly seems true for ΔF, but both models appear to give more-or-less identical predictions here for ΔR.*

We understand the concern here to be that the ΔV and ΔR models make the same prediction for changes in ΔR, meaning that the ΔR data cannot be used to distinguish between the two models. However, taking a step back and looking across the ΔR and ΔF data simultaneously, it can be seen that the ΔR model gets both patterns right, whereas the ΔV model gets only one of them right. It is this fact that we take as evidence in support of the ΔR model for ΔF and ΔR. The text has been modified to make it more clear that we are looking across ΔF*and* ΔR when comparing the models (subsection “Genetic effects on command neuron function”, sixth paragraph).

*3) In the first paragraph of the subsection “The Stochastic Switch Model and other random search behaviors“. It is claimed that "changes are required in at least three weights to produce the full range of search behaviors." This is not actually shown anywhere in the paper, just that three weights are sufficient to induce the full range. A supplementary figure to this effect would be useful.*

The revised text and figures now carefully document weight changes that are minimally required for the full range of search modes (Figure 8 including supplementals, Table 7 (new), and accompanying text (subsection “Regulation of search scale, last two paragraphs and subsection “Biased random walks”).

4) The predicted value for the dwell time in the forward state is not comparable to the rest of the literature, as a different definition is used here. What would happen if a standard definition was used? Does the same rate result?

We now report the mean dwell times for F, R and X based on fixed velocity thresholds of ± 0.05 mm/s. *dF* = 1.86 ± 0.03; *dR* = 1.23 ± 0.02;dP,0.05= 0.14 ± 0.001. Thus, the short dwell times in the forward state that we observed were not due to the model. We attribute them instead to the higher time resolution of our tracking hardware (subsection “Wild type locomotion”, first paragraph).